# Discrete and Continuous Difference of Submodular Minimization

**George Orfanides** [1]   **Tim Hoheisel** [1]   **Marwa El Halabi** [2]

## Abstract

Submodular functions, defined on continuous or discrete domains, arise in numerous applications. We study the minimization of the difference of two submodular (DS) functions, over both domains, extending prior work restricted to set functions. We show that all functions on discrete domains and all smooth functions on continuous domains are DS. For discrete domains, we observe that DS minimization is equivalent to minimizing the difference of two convex (DC) functions, as in the set function case. We propose a novel variant of the DC Algorithm (DCA) and apply it to the resulting DC Program, obtaining comparable theoretical guarantees as in the set function case. The algorithm can be applied to continuous domains via discretization. Experiments demonstrate that our method outperforms baselines in integer compressive sensing and integer least squares.

## 1. Introduction

Many problems in machine learning require solving a non-convex problem, with potentially mixed discrete and continuous variables. In this paper, we investigate a broad class of such problems that possess a special structure, namely the minimization of the *difference of two submodular (DS)* functions, over both continuous and discrete domains of the form $\mathcal{X} = \prod_{i=1}^{n} \mathcal{X}_i$, where each $\mathcal{X}_i \subseteq \mathbb{R}$ is compact:

$$\min_{x \in \mathcal{X}} F(x) := G(x) - H(x), \tag{1}$$

and where $G$ and $H$ are *normalized submodular* functions.

Submodularity is an important property which naturally occurs in a variety of machine learning applications (Bilmes, 2022; Bach, 2013). Most of the submodular optimization literature focuses on *set functions*, which can be equivalently viewed as functions on $\mathcal{X} = \{0, 1\}^n$. Throughout, we make this identification and refer to them as such. Submodular set functions can be minimized in polynomial time exactly or up to arbitrary accuracy (Axelrod et al., 2020), and maximized approximately in polynomial time (Buchbinder et al., 2015).

Beyond set functions, submodularity extends to general lattices like $\mathcal{X}$ (Topkis, 1978). Bach (2019) extended results from submodular set function minimization to general domains $\mathcal{X}$. In particular, he provided two polynomial-time algorithms for submodular minimization with arbitrary accuracy on discrete domains, which are also applicable to continuous domains via discretization. Axelrod et al. (2020) later developed a faster algorithm for this problem. Bian et al. (2017b); Niazadeh et al. (2020) extended results from submodular set function maximization to continuous domains, with Bian et al. (2017b) giving a deterministic $1/3$-approximation, and Niazadeh et al. (2020) an optimal randomized $1/2$-approximation, both in polynomial-time.

In this paper, we extend this line of inquiry by generalizing DS set function minimization results to general domains $\mathcal{X}$. However, unlike the special case where $F$ is submodular, Problem (1) in general cannot be solved efficiently, neither exactly nor approximately. Indeed, even for set functions, any constant-factor multiplicative approximation requires exponential time, and any positive polynomial-time computable multiplicative approximation is NP-Hard (Iyer & Bilmes, 2012, Theorems 5.1, 5.2). Even obtaining a local minimum (Definition 4.3) is PLS complete (Iyer & Bilmes, 2012, Section 5.3). Prior work (Narasimhan & Bilmes, 2005; Iyer & Bilmes, 2012; El Halabi et al., 2023) developed descent algorithms for Problem (1) in the special case of set functions, which converge to a local minimum of $F$.

Minimizing DS set functions is equivalent to minimizing the *difference of two convex (DC)* functions, obtained by replacing each submodular function by its *Lovász extension* (Lovász, 1983). DC programs capture most well-behaved non-convex problems on continuous domains. The DC algorithm (DCA) is a classical method to solve them (Le Thi & Pham Dinh, 2018). El Halabi et al. (2023) leverage this connection to minimize DS set functions by applying DCA variants to the equivalent DC program. They also show that the submodular-supermodular (SubSup) method of Narasimhan & Bilmes (2005) is a special case of one of their variants.

[1]Department of Mathematics and Statistics, McGill University, Montreal [2]Samsung AI Lab, Montreal. Correspondence to: George Orfanides <george.orfanides@mail.mcgill.ca>, Marwa El Halabi <marwa.elhalabi@gmail.com>.

*Proceedings of the $42^{nd}$ International Conference on Machine Learning*, Vancouver, Canada. PMLR 267, 2025. Copyright 2025 by the author(s).

We adopt a similar approach here to solve Problem (1) for general discrete domains. We use the convex continuous extension introduced in Bach (2019) to reformulate the problem as a DC program, then apply a DCA variant, obtaining theoretical guarantees comparable to the set function case. As in Bach (2019), the algorithm is applicable to continuous domains via discretization. Our key contributions are:

- We show that any function on a discrete domain and any smooth function on a continuous domain can be expressed as a DS function, though finding a *good* DS decomposition can be challenging.

- We highlight applications with natural DS objectives, such as quadratic programming and sparse learning.

- For discrete domains, we introduce an efficient DCA variant for Problem (1), which is a descent method and converges to a local minimum at rate $O(1/T)$.

- We demonstrate that our method empirically outperforms baselines on two applications: integer compressed sensing and integer least squares.

**Additional Related Work** Maehara & Murota (2015) proposed a discrete analogue of DCA for minimizing the difference of two *discrete* convex (*discrete DC*) functions $G, H : \mathbb{Z}^n \to \mathbb{Z} \cup \{+\infty\}$. They also show that any function $F : D \to \mathbb{Z}$ on a finite $D \subset \mathbb{Z}^n$ is the restriction of a discrete DC function to $D$, making discrete DC functions essentially equivalent to *integer valued* DS functions on discrete domains. However, obtaining a DS decomposition is easier, since a discrete DC function is a special case of DS functions. In Section 3.2, we provide examples of applications which naturally have the form of Problem (1), but not of a discrete DC program, even if we ignore the integer valued restriction. For further details, see Appendix B.2.

Several works studied optimizing *DR-submodular* functions, a subclass of submodular functions which are concave along non-negative directions. For set functions, DR-submodularity is equivalent to submodularity, but not for more general domains such as the integer lattice. Ene & Nguyen (2016) reduced DR-submodular functions on a bounded integer lattice $\mathcal{X} = \prod_{i=1}^{n} \{0, \ldots, k_i - 1\}$, to submodular set functions on a ground set of size $O(\log(k))$, enabling the application of set-function results to this setting. Approximation algorithms with provable guarantees for maximizing continuous DR-submodular functions on continuous domains have also been proposed (Bian et al., 2017a;b; Hassani et al., 2017; Mokhtari et al., 2017; Fazel & Sadeghi, 2023; Mualem & Feldman, 2023). Yu & Küçükyavuz (2024) also gave a polynomial-time algorithm for DR-submodular minimization with mixed-integer variables over box and monotonicity constraints, with no dependence on $k$.

## 2. Preliminaries

We introduce here our notation and relevant background. Given $n \in \mathbb{N}$, we let $[n] = \{1, \ldots, n\}$ and $[0 : n] = \{0\} \cup [n]$. We use $\overline{\mathbb{R}} = \mathbb{R} \cup \{+\infty\}$ and $\overline{\mathbb{Z}} = \mathbb{Z} \cup \{+\infty\}$. The standard basis vectors for $\mathbb{R}^n$ and $\mathbb{R}^{n \times m}$ are denoted by $e_i$, $i \in [n]$ and $E_{ij}$, $i \in [n]$, $j \in [m]$, respectively. The vectors of all ones and all zeros in $\mathbb{R}^n$ are denoted by $\mathbb{1}$ and $\mathbf{0}$ respectively. The support of a vector $x \in \mathbb{R}^n$ is defined as $\text{supp}(x) = \{i \mid x_i \neq 0\}$. For $q > 0$, the $\ell_q$-(quasi)-norm is given by $\|x\|_q = (\sum_{i=1}^{n} x_i^q)^{1/q}$ and the $\ell_0$-pseudo-norm by $\|x\|_0 = |\text{supp}(x)|$. We conflate $\|x\|_q^q$ with $\|x\|_0$ for $q = 0$. Given $x, y \in \mathbb{R}^n$, $x \leq y$ means that $x_i \leq y_i$ for all $i \in [n]$.

We denote by $\langle \cdot, \cdot \rangle_F$ and $\|\cdot\|_F$ the Frobenius inner product and norm respectively. We call $X \in \mathbb{R}^{n \times m}$ *row non-increasing* if its rows are all non-increasing, i.e., $X_{i,j} \geq X_{i,j+1}$ for all $i \in [n]$, $j \in [m-1]$. We denote by $\{0, 1\}_\downarrow^{n \times m}$, $[0, 1]_\downarrow^{n \times m}$, and $\mathbb{R}_\downarrow^{n \times m}$ the sets of row non-increasing matrices in $\{0, 1\}^{n \times m}$, $[0, 1]^{n \times m}$, and $\mathbb{R}^{n \times m}$, respectively. Given a vector $x \in \mathbb{R}^m$, a *non-increasing* permutation $p = (p_1, \ldots, p_m)$ of $x$ satisfies $x_{p_i} \geq x_{p_{i+1}}$ for all $i \in [m]$. Similarly, a non-increasing permutation $(p, q) = ((p_1, q_1), \ldots, (p_{nm}, q_{nm}))$ of a matrix $X \in \mathbb{R}^{n \times m}$ satisfies $X_{p_i, q_i} \geq X_{p_{i+1}, q_{i+1}}$ for all $i \in [nm]$. A non-increasing permutation $(p, q)$ of $X \in \mathbb{R}^{n \times m}$ is called *row-stable* if it preserves the original order within rows when breaking ties, i.e., for any $i, j \in [nm]$ with $i < j$, if $p_i = p_j$, then $q_i < q_j$.

Throughout this paper, we consider a function $F : \mathcal{X} \to \mathbb{R}$ defined on $\mathcal{X} = \prod_{i=1}^{n} \mathcal{X}_i$, the product of $n$ compact subsets $\mathcal{X}_i \subset \mathbb{R}$. The set $\mathcal{X}_i$ is typically a finite set such as $\mathcal{X}_i = \{0, \ldots, k_i - 1\}$ (*discrete domain*) or a closed interval (*continuous domain*). We assume access to an *evaluation oracle* of $F$, which returns $F(x)$ for any $x \in \mathcal{X}$ in time $\text{EO}_F$. Whenever we state that $F$ is differentiable on $\mathcal{X}$, we mean that $F$ is differentiable on an open set containing $\mathcal{X}$. We say that $F$ is $L$-Lipschitz continuous on $\mathcal{X}$ with respect to $\|\cdot\|$, if $|F(x) - F(y)| \leq L\|x - y\|$ for all $x, y \in \mathcal{X}$. When not specified, $\|\cdot\|$ is the $\ell_2$-norm.

A function $F$ is *normalized* if $F(x^{\min}) = 0$ where $x^{\min}$ is the smallest element in $\mathcal{X}$, non-decreasing (non-increasing) if $F(x) \leq F(y)$ $(F(x) \geq F(y))$ for all $x \leq y$, and monotone if it is non-decreasing or non-increasing. We assume without loss of generality that $F$ is normalized.

**Submodularity** A function $F$ is said to be *submodular* if

$$F(x) + F(y) \geq F(\min\{x, y\}) + F(\max\{x, y\}), \quad (2)$$

for all $x, y \in \mathcal{X}$, where the min and max are applied component-wise. We call $F$ *modular* if Eq. (2) holds with equality, *strictly submodular* if the inequality is strict whenever $x$ and $y$ are not comparable, and *(strictly) supermodular* if $-F$ is (strictly) submodular. A function $F$ is modular iff it is separable (Topkis, 1978, Theorem 3.3).

An equivalent diminishing-return characterization of submodularity is that $F$ is submodular iff for all $x \in \mathcal{X}$, $i, j \in [n], i \neq j, a_i, a_j > 0$ such that $x + a_i e_i, x + a_j e_j \in \mathcal{X}$,

$$F(x+a_i e_i) - F(x) \geq F(x+a_i e_i + a_j e_j) - F(x + a_j e_j). \quad (3)$$

Strict submodularity then corresponds to a strict inequality in (3) (Topkis, 1978, Theorem 3.1-3.2). Other equivalent characterizations of submodularity hold for differentiable and twice-differentiable $F$ (Topkis, 1978, Section 3).

**Proposition 2.1.** *Given $F : \mathcal{X} \to \mathbb{R}$, where each $\mathcal{X}_i$ is a closed interval, we have:*

a) *If $F$ is differentiable on $\mathcal{X}$, then $F$ is submodular iff for all $x \in \mathcal{X}$, $i, j \in [n]$, $i \neq j$, $a > 0$ such that $x + ae_j \in \mathcal{X}$:*

$$\frac{\partial F}{\partial x_i}(x) \geq \frac{\partial F}{\partial x_i}(x + ae_j). \quad (4)$$

b) *If $F$ is twice-differentiable on $\mathcal{X}$, then $F$ is submodular iff for all $x \in \mathcal{X}, i \neq j$:*

$$\frac{\partial^2 F}{\partial x_i \partial x_j}(x) \leq 0, \ i \neq j. \quad (5)$$

*In both cases, $F$ is also strictly submodular if the inequalities are strict.*

On continuous domains, submodular functions can be convex, concave, or neither (Bach, 2019, Section 2.2). For $n = 1$, any function is modular and strictly submodulars since any $x, y \in \mathbb{R}$ are comparable. (Strict) submodularity is preserved under addition, positive scalar multiplication, and restriction (Bach, 2019, Section 2.1).

**Submodular minimization** Minimizing a submodular *set function $F$* is equivalent to minimizing a convex continuous extension of $F$, called the Lovász extension (Lovász, 1983), on the hypercube $[0,1]^n$. Bach (2019) extended this result to general submodular functions on $\mathcal{X}$. To that end, he introduced a continuous extension of functions defined on $\mathcal{X}$ to the set of Radon probability measures on $\mathcal{X}$. When $\mathcal{X} = \prod_{i=1}^{n}[0 : k_i - 1]$, this extension has a simple, efficiently computable form (Bach, 2019, Section 4.1), given below. For simplicity, we assume all $k_i$'s are equal to $k$.

**Definition 2.2.** Given a normalized function $F : [0 : k - 1]^n \to \mathbb{R}$, its *continuous extension $f_\downarrow : \mathbb{R}^{n \times (k-1)} \to \overline{\mathbb{R}}$* is defined as follows: Given $X \in \mathbb{R}_\downarrow^{n \times (k-1)}$ and $(p, q)$ a non-increasing permutation of $X$, define $y^i \in [0 : k-1]^n$ as $y^0 = \mathbf{0}$ and $y^i = y^{i-1} + e_{p_i}$ for $i \in [(k-1)n]$. Then,

$$f_\downarrow(X) = \sum_{i=1}^{(k-1)n} X_{p_i, q_i} (F(y^i) - F(y^{i-1})) \quad (6)$$

For $X \notin \mathbb{R}_\downarrow^{n \times (k-1)}$, we set $f_\downarrow(X) = +\infty$.

Evaluating $f_\downarrow(x)$ takes $O(r \log r + r \ \mathrm{EO}_F)$ with $r = nk$. The value of $f_\downarrow(X)$ is invariant to the permutation choice.

We can define a bijection map $\Theta : \{0,1\}_\downarrow^{n \times (k-1)} \to [0 : k-1]^n$ and its inverse as

$$\Theta(X) = X\mathbb{1}, \quad (7a)$$

$$\Theta^{-1}(x) = X \text{ where } X_{i,j} = \begin{cases} 1, & \text{if } j \leq x_i, \\ 0, & \text{if } j > x_i \end{cases}. \quad (7b)$$

In the set function case, i.e., when $k = 2$, we have $\{0,1\}_\downarrow^{n \times (k-1)} = \{0,1\}^n$ and $[0,1]_\downarrow^{n \times (k-1)} = [0,1]^n$, so $\Theta$ becomes the identity map, and $f_\downarrow$ reduces to the Lovász extension; see e.g. Bach (2013, Definition 3.1).

We list below some key properties of the extension. Items a-e, Item g (general bound) are from Bach (2019), Item f follows directly from the definition of $f_\downarrow$, and we prove the bound for non-decreasing $F$ in Item g in Appendix D.

**Proposition 2.3.** *For a normalized function $F : [0 : k - 1]^n \to \mathbb{R}$ and its continuous extension $f_\downarrow$, we have:*

a) *For all $X \in \{0,1\}_\downarrow^{n \times (k-1)}, f_\downarrow(X) = F(\Theta(X))$.*

b) *If $F$ is submodular, then $\min_{x \in [0:k-1]^n} F(x) = \min_{x \in [0,1]_\downarrow^{n \times (k-1)}} f_\downarrow(X)$.*

c) *Given $X \in [0,1]_\downarrow^{n \times (k-1)}$ and $\{y^i\}_{i=0}^{(k-1)n}$ as defined in Definition 2.2, let $i^* \in \operatorname{argmin}_{i \in [0:(k-1)n]} F(y^i)$ and $x = y^{i^*}$, then $F(x) \leq f_\downarrow(X)$. We refer to this as rounding and denote it by $x = \mathrm{Round}_F(X)$.*

d) *$f_\downarrow$ is convex iff $F$ is submodular.*

e) *Given $X \in \mathbb{R}_\downarrow^{n \times (k-1)}$, a non-increasing permutation $(p, q)$ of $X$, and $\{y^i\}_{i=1}^{(k-1)n}$ defined as in Definition 2.2, define $Y \in \mathbb{R}^{n \times (k-1)}$ as $Y_{p_i, q_i} = F(y^i) - F(y^{i-1})$. If $F$ is submodular, then $Y$ is a subgradient of $f_\downarrow$ at $X$.*

f) *If $F = G - H$, then $f_\downarrow = g_\downarrow - h_\downarrow$.*

g) *If $F$ is submodular, then $f_\downarrow$ is $L_{f_\downarrow}$-Lipschitz continuous with respect to the Frobenius norm, with $L_{f_\downarrow} \leq \sqrt{(k-1)n} \max_{x,i} |F(x + e_i) - F(x)|$. If $F$ is also non-decreasing, then $L_{f_\downarrow} \leq F((k-1)\mathbb{1})$.*

Bach (2019, Section 5) provided two polynomial-time algorithms for minimizing submodular functions on discrete domains up to arbitrary accuracy $\epsilon \geq 0$; one using projected subgradient method and the other using Frank-Wolfe (FW) method. When $\mathcal{X} = [0 : k-1]^n$, both obtain an $\epsilon$-minimum in $\tilde{O}((nkL_{f_\downarrow}/\epsilon)^2 \ \mathrm{EO}_F)$ time. Though FW is empirically faster; especially the pairwise FW variant of (Lacoste-Julien & Jaggi, 2015a), which we use in our experiments (Section 5). Axelrod et al. (2020, Theorem 9) gave a faster

randomized algorithm with runtime $\tilde{O}(n(kL_{f_\downarrow}/\epsilon)^2\,\mathrm{EO}_F)$, based on projected stochastic subgradient method.

**DC Programming** Given a Euclidean space $\mathbb{E}$, let $\Gamma_0(\mathbb{E})$ be the set of proper lower semi-continuous convex functions from $\mathbb{E}$ into $\overline{\mathbb{R}}$. For a set $C \subseteq \mathbb{E}, \delta_C$ is the indicator function of $C$ taking value $0$ on $C$ and $+\infty$ outside it. Given a function $f : \mathbb{E} \to \overline{\mathbb{R}}$, its domain is defined as $\mathrm{dom}\, f = \{x \in \mathbb{E} \mid f(x) < +\infty\}$. For $f \in \Gamma_0(\mathbb{E})$, $\epsilon \geq 0$, and $x^0 \in \mathrm{dom}\, f$, the $\epsilon$-subdifferential of $f$ at $x^0$ is defined by $\partial_\epsilon f(x^0) = \{y \in \mathbb{E} \mid f(x) \geq f(x^0) + \langle y, x - x^0\rangle - \epsilon, \forall x \in \mathbb{E}\}$, while $\partial f(x^0)$ stands for the exact subdifferential ($\epsilon = 0$). An element of the $\epsilon$-subdifferential is called an $\epsilon$-subgradient, or simply a subgradient when $\epsilon = 0$.

A standard DC program takes the form

$$\min_{x \in \mathbb{E}} f(x) := g(x) - h(x) \tag{8}$$

where $g, h \in \Gamma_0(\mathbb{E})$. We adopt the convention $+\infty - (+\infty) = +\infty$. The function $f$ is called a DC function.

DC-programs are generally non-convex and non-differentiable. A point $x^*$ is a global minimum of Problem (8) iff $\partial_\epsilon h(x) \subseteq \partial_\epsilon g(x)$ for all $\epsilon \geq 0$ (Hiriart-Urruty, 1989, Theorem 4.4). However, checking this condition is generally infeasible (Pham Dinh & Le Thi, 1997). Instead, we are interested in notions of approximate stationarity. In particular, we say that a point $x$ is an $\epsilon$-critical point of $g - h$ if $\partial_\epsilon g(x) \cap \partial h(x) \neq \emptyset$. Criticality depends on the choice of the DC decomposition $g - h$ of $f$ (Le Thi & Pham Dinh, 2018, Section 1.1) and implies local minimality over a restricted set (El Halabi et al., 2023, Proposition 2.7).

**Proposition 2.4.** *Let $\epsilon \geq 0$ and $f = g - h$ with $g, h \in \Gamma_0(\mathbb{E})$. If $x, x' \in \mathbb{E}$ satisfy $\partial_\epsilon g(x) \cap \partial h(x') \neq \emptyset$, then $f(x) \leq f(x') + \epsilon$*

DCA iteratively minimizes a convex majorant of (8), obtained by replacing $h$ at iteration $t$ by its affine minorization $h(x^t) + \langle y^t, x - x^t\rangle$, with $y^t \in \partial h(x^t)$. Starting from $x^0 \in \mathrm{dom}\, \partial g$, DCA iterates are given by

$$y^t \in \partial h(x^t) \tag{9a}$$
$$x^{t+1} \in \underset{x \in \mathbb{E}}{\mathrm{argmin}}\; g(x) - \langle y^t, x\rangle. \tag{9b}$$

We list now some properties of DCA (Pham Dinh & Le Thi, 1997, Theorem 3), (El Halabi et al., 2023, Theorem 3.1).

**Proposition 2.5.** *Given $f = g - h$ with $g, h \in \Gamma_0(\mathbb{E})$ and a finite minimum $f^*$, let $\epsilon, \epsilon_x \geq 0$, and $\{x^t\}, \{y^t\}$ be generated by DCA (9), where subproblem (9b) is solved up to accuracy $\epsilon_x$. Then for all $t, T \in \mathbb{N}$, we have:*

*a) $f(x^{t+1}) \leq f(x^t) + \epsilon_x$.*

*b) If $f(x^t) - f(x^{t+1}) \leq \epsilon$, then $x^t$ is an $\epsilon + \epsilon_x$-critical point of $g - h$ with $y^t \in \partial_{\epsilon + \epsilon_x} g(x^t) \cap \partial h(x^t)$.*

*c) $\min_{t \in [T-1]} f(x^t) - f(x^{t+1}) \leq \frac{f(x^0) - f^*}{T}$*

DCA is thus a descent method (up to $\epsilon_x$) which converges ($f(x^t) - f(x^{t+1}) \leq \epsilon$) to a critical point with rate $O(1/T)$. We note that Item c is not specific to DCA, but instead follows from $f(x^T) \geq f^*$.

## 3. Difference of Submodular functions

We start by identifying which functions are expressible as DS functions, then highlight important applications where they naturally occur.

### 3.1. Representability

The structure assumed in Problem (1) may seem arbitrary, but it is in fact very general. In particular, we prove that any function on a discrete domain and any smooth function on a continuous domain can be expressed as a DS function.

**Proposition 3.1.** *Given any normalized $F : \mathcal{X} \to \mathbb{R}$ where each $\mathcal{X}_i \subset \mathbb{R}$ is finite, there exists a decomposition $F = G - H$, where $G, H : \mathcal{X} \to \mathbb{R}$ are normalized (strictly) submodular functions.*

*Proof Sketch.* We give a constructive proof, similar to the one in (Iyer & Bilmes, 2012, Lemma 3.1) for set functions. We choose a normalized *strictly* submodular function $\tilde{H} : \mathcal{X} \to \mathbb{R}$, then define $G = F + \frac{|\alpha|}{\beta}\tilde{H}$ and $H = \frac{|\alpha|}{\beta}\tilde{H}$, where $\alpha \leq 0$ and $\beta > 0$ are lower bounds on the difference between the left and right hand sides of Ineq. (3) for $F$ and $\tilde{H}$ respectively. We verify that $G$ and $H$ are normalized and submodular, using Ineq. (3). Choosing $\alpha < 0$ as a strict lower bound further guarantees strict submodularity. The full proof is given in Appendix E.1. $\square$

Obtaining *tight* lower bounds $\alpha$ and $\beta$ in the above proof requires exponential time in general, even for set functions (Iyer & Bilmes, 2012). We provide in Example 3.2 a valid choice for $\tilde{H}$, for which a tight $\beta$ can be easily computed. For any $F$, we can use $\alpha = -4\max_{x \in \mathcal{X}} |F(x)|$, which is often easy to lower bound. Though loose bounds $\alpha$ and $\beta$ result in slower optimization, as explained in Appendix A.

**Example 3.2.** *Let $H : \mathbb{R}^n \to \mathbb{R}$ be the quadratic function $H(x) = -\frac{1}{2}x^\top J x$, where $J$ is the matrix of all ones, and define $\tilde{H} : \mathcal{X} \to \mathbb{R}$ as $\tilde{H}(x) = H(x) - H(x^{\min})$. Then, $\tilde{H}$ is a normalized strictly submodular function.*

*Proof Sketch.* By Proposition 2.1-b, $H$ is strictly submodular on any product of closed intervals. Since strict submodularity is preserved by restriction, $\tilde{H}$ is also strictly submodular on $\mathcal{X}$. The full proof is given in Appendix E.1. $\square$

In the above example, we have $\tilde{H}(x + a_i e_i) - \tilde{H}(x) - \tilde{H}(x + a_i e_i + a_j e_j) + \tilde{H}(x + a_j e_j) = a_i a_j$ for any $x \in \mathcal{X}, a_i, a_j > 0$. When $\mathcal{X}$ is discrete, we obtain a tight lower bound $\beta = d_i d_j$ where $d_i d_j$ are the distances between the closest two points in $\mathcal{X}_i, \mathcal{X}_j$ respectively, for some $j \neq i$.

The decomposition given in the proof of Proposition 3.1 cannot be applied to continuous domains. Indeed, if $\tilde{H}$ is a continuous function, taking $a_i \to 0$ or $a_j \to 0$ in inequality (3) yields $\beta = 0$. However, a similar decomposition can be obtained in this case, if $F$ satisfies some smoothness condition.

**Proposition 3.3.** *Given any normalized $F : \mathcal{X} \to \mathbb{R}$ where each $\mathcal{X}_i$ is a closed interval, if $F$ is differentiable and there exist $L_F \geq 0$ such that $\frac{\partial F}{\partial x_i}(x) - \frac{\partial F}{\partial x_i}(x + a e_j) \geq -L_F a$, for all $x \in X, i \neq j, a > 0, x_j + a \in \mathcal{X}_j$, then there exists a decomposition $F = G - H$, where $G, H : \mathcal{X} \to \mathbb{R}$ are normalized (strictly) submodular functions.*

*Proof Sketch.* The proof is similar to that of Proposition 3.1. We choose a normalized *strictly* submodular function $\tilde{H} : \mathcal{X} \to \mathbb{R}$, which is differentiable and satisfies $\frac{\partial \tilde{H}}{\partial x_i}(x) - \frac{\partial \tilde{H}}{\partial x_i}(x + a e_j) \geq L_{\tilde{H}} a$, for some $L_{\tilde{H}} > 0$, then define $G = F + \frac{L_F}{L_{\tilde{H}}} \tilde{H}$ and $H = \frac{L_F}{L_{\tilde{H}}} \tilde{H}$. We verify that $G$ and $H$ are normalized and submodular, using Proposition 2.1-a. Choosing $L_F > 0$ as a strict lower bound further guarantees strict submodularity. The full proof is in Appendix E. $\square$

One sufficient condition for $F$ to satisfy the assumption in Proposition 3.3 is for its gradient to be $L_F$-Lipschitz continuous with respect to the $\ell_1$-norm, i.e., $\|\nabla F(x) - \nabla F(y)\|_\infty \leq L_F \|x - y\|_1$ for all $x, y \in \mathcal{X}$. It follows then that any twice continuously differentiable function on $\mathcal{X}$ is also a DS function, with $L_F = \max_{x \in \mathcal{X}} \|\nabla^2 F(x)\|_{1,\infty}$ (Beck, 2017, Theorem 5.12). In both cases, computing a *tight* Lipschitz constant $L_F$ has exponential complexity in general (Huang et al., 2023, Theorem 2.2). Though often one can easily derive a bound on $L_F$. The function in Example 3.2 is again a valid choice for $\tilde{H}$, where $L_{\tilde{H}} = 1$ is tight.

Like DC functions, DS functions admit infinitely many DS decompositions. For the specific decompositions in Propositions 3.1 and 3.3, the "best" one is arguably the one with the tightest $\alpha, \beta$ and $L_F, L_{\tilde{H}}$, respectively. Finding the "best" DS decomposition in general is even more challenging, as it is unclear how to define "best". This question is explored for set functions in Brandenburg et al. (2024), who study the complexity of decomposing a set function into a difference of submodular set functions such that their Lovász extensions have as few pieces as possible.

In Appendix B, we discuss the connection between DS functions and related non-convex function classes. In particular, we note that DS functions on continuous domains are not necessarily DC, and that the discrete DC functions considered in Maehara & Murota (2015) are essentially equivalent to *integer valued* DS functions on discrete domains.

## 3.2. Applications

As discussed in Section 3.1, Problem (1) covers most well behaved non-convex problems over discrete and continuous domains. Though finding a good DS decomposition can be expensive in general. We give here examples of applications which naturally have the form of Problem (1).

**Quadratic Programming** Quadratic programs (QP) of the form $\min_{x \in \mathcal{X}} \frac{1}{2} x^\top Q x + c^\top x$, arise in numerous applications. This form includes box constrained QP (BCQP), where $\mathcal{X}_i$'s are all closed intervals, as well as integer and mixed-integer BCQP, where some or all $\mathcal{X}_i \subseteq \mathbb{Z}^n$. Such problems are NP-Hard if the objective is non-convex (De Angelis et al., 1997), or if some of the variables are discrete (Dinur et al., 1998).

The objective in these QPs has a natural DS decomposition $F = G - H$, with $G(x) = x^\top Q^- x + c^\top x$ and $H(x) = x^\top (-Q^+) x$, where $Q^- = \min\{Q, 0\}$ and $Q^+ = \max\{Q, 0\}$. By Proposition 2.1-b, $G$ and $H$ are submodular, since $c^\top x$ is modular. When $\mathcal{X}$ is continuous, these QPs can also be written as DC programs, but this requires computing the minimum eigenvalue of $Q$.

**Sparse Learning** Optimization problems of the form $\min_{x \in \mathcal{X}} \ell(x) + \lambda \|x\|_q^q$, where $q \in [0, 1)$, $\lambda \geq 0$, and $\ell$ is a smooth function, arise in sparse learning, where the goal is to learn a *sparse* parameter vector from data. There, the loss $\ell$ is often smooth and convex (e.g., square or logistic loss), and the non-convex regularizer $\|x\|_q^q$ promotes sparsity. The domain is often unbounded so $\mathcal{X}$ can be set to $\|x\|_\infty \leq R$ for some $R \geq 0$, or $\mathcal{X} \subseteq \mathbb{Z}^n$ as in the integer compressed sensing problems we consider in Section 5.2. Using $q < 1$ makes the problem NP-Hard, even for continuous $\mathcal{X}$ (Chen et al., 2017), but can be preferable to the convex $\ell_1$-norm, as it leads to fewer biasing artifacts (Fan & Li, 2001).

Note that the regularizer $\|x\|_q^q$ is modular since it is separable. Hence, these problems are instances of Problem (1), where a DS decomposition of $\ell$ can be obtained as in Proposition 3.3. If $\ell(x) = \|Ax - b\|_2^2$, we can use the same decomposition as in the QPs above, i.e., $G(x) = x^\top Q^- x + c^\top x + \lambda \|x\|_q^q$ and $H(x) = x^\top (-Q^+) x$, with $Q = A^\top A$ and $c = -2A^\top b$. These problems cannot be written as DC programs even when $\mathcal{X}$ is continuous, since $\|x\|_q^q$ is not DC, as we prove in Proposition B.1.

We show in Appendix B.2 that the natural DS decomposition in both applications is not a discrete DC decomposition as defined in Maehara & Murota (2015) and cannot easily be adapted into one for general discrete domains, even when ignoring the integer-valued restriction.

# 4. Difference of Submodular Minimization

In this section, we show that most well behaved instances of problem (1) can be reduced to DS minimization over a bounded integer lattice $\prod_{i=1}^{n}[0 : k_i - 1]$ for some $k_i \in \mathbb{N}$. Then, we address solving this special case.

## 4.1. Reduction to Integer Lattice Domain

We first obverse that submodularity is preserved by any separable monotone reparametrization. A special case of the following proposition is stated in Bach (2019, Section 2.1) with $\mathcal{X}' = \mathcal{X}$ and $m$ strictly increasing.

**Proposition 4.1.** *Given* $\mathcal{X} = \prod_{i=1}^{n} \mathcal{X}_i, \mathcal{X}' = \prod_i \mathcal{X}_i'$, *with compact sets* $\mathcal{X}_i, \mathcal{X}_i' \subset \mathbb{R}$, *let* $F : \mathcal{X} \to \mathbb{R}$ *and* $m : \mathcal{X}' \to \mathcal{X}$ *be a monotone function such that* $[m(x)]_i = m_i(x_i)$. *If* $F$ *is submodular, then the function* $F' : \mathcal{X}' \to \mathbb{R}$ *given by* $F'(x) = F(m(x))$ *is submodular. Moreover, if* $m$ *is strictly monotone, then* $F$ *is submodular iff* $F'$ *is submodular.*

*Proof Sketch.* We observe that $\min\{m(x), m(y)\} + \max\{m(x), m(y)\} = m(\min\{x, y\}) + m(\max\{x, y\})$. The claim then follows by verifying that Eq. (2) holds. The full proof is given in Appendix F.1 $\qquad\square$

**Discrete case** For discrete domains, the desired reduction follows from Proposition 4.1, by noting that in this case elements in $\mathcal{X}$ can be uniquely mapped to elements in $\prod_{i=1}^{n}[0 : k_i - 1]$ with $k_i = |\mathcal{X}_i|$ via a strictly increasing map.

**Corollary 4.2.** *Minimizing any DS function* $F : \mathcal{X} \to \mathbb{R}$, *where each* $\mathcal{X}_i \subseteq \mathbb{R}$ *is a finite set, is equivalent to minimizing a DS function on* $\prod_{i=1}^{n}[0 : k_i - 1]$ *with* $k_i = |\mathcal{X}_i|$.

*Proof.* For each $i \in [n]$, let $\mathcal{X}_i = \{x_0^i, \ldots, x_{k_i-1}^i\}$ where elements are ordered in non-decreasing order, i.e., $x_j^i < x_{j+1}^i$ for all $j \in [0 : k_i - 2]$. Define the map $m_i : [0 : k_i - 1] \to \mathcal{X}_i$ as $m_i(j) = x_j^i$. Then $m_i$ is a strictly increasing bijection. By Proposition 4.1, then the function $F' : \prod_{i=1}^{n}[0 : k_i - 1] \to \mathbb{R}$ defined as $F'(x) = F(m(x))$ with $[m(x)]_i = m_i$ is a DS function. $\qquad\square$

**Continuous case** We can convert continuous domains into discrete ones using discretization. We consider $\mathcal{X} = [0, 1]^n$ wlog, since by Proposition 4.1 any DS function $F$ defined on $\prod_{i=1}^{n}[a_i, b_i]$ can be reduced to a DS function $F'$ on $[0, 1]^n$ by translating and scaling.

As done in Bach (2019, Section 5.1), given a function $F : [0, 1]^n \to \mathbb{R}$ which is $L$-Lipschitz continuous with respect to the $\ell_\infty$-norm and $\epsilon > 0$, we define $k = \lceil L/\epsilon \rceil + 1$ and the function $F' : [0 : k - 1]^n \to \mathbb{R}$ as $F'(x) = F(x/(k-1))$. Then we have

$$\min_{x \in [0:k-1]^n} F'(x) - \epsilon/2 \leq \min_{x \in [0,1]^n} F(x) \leq \min_{x \in [0:k-1]^n} F'(x).$$

Again by Proposition 4.1, if $F$ is DS then $F'$ is also DS. Moreover, given any minimizer $x^*$ of $F'$, $x^*/(k-1)$ is an $\frac{\epsilon}{2}$-minimizer of $F$. It is worth noting that Lipschitz continuity is not necessary for bounding the discretization error. We show in Appendix F.2 how to handle the function $F(x) = \|x\|_q^q$, with $q \in [0, 1)$, on a domain where it is not Lipschitz continuous.

## 4.2. Optimization over Integer Lattice

We now address solving Problem (1) for $\mathcal{X} = \prod_{i=1}^{n}[0 : k_i - 1]$. For simplicity, we assume $k_i = k$ for all $i \in [n]$, for some $k \in \mathbb{N}$, i.e., $\mathcal{X} = [0 : k - 1]^n$. The results can be easily extended to unequal $k_i$'s. Since Problem (1) is inapproximable, we will focus on obtaining approximate local minimizers. Given $x \in \mathcal{X}$, we define the set of neighboring points of $x$ in $\mathcal{X}$ as $N_\mathcal{X}(x) := \{x' \in \mathcal{X} \mid \exists i \in [n], x' = x \pm e_i\}$.

**Definition 4.3.** *Given* $\epsilon \geq 0$ *and* $x \in \mathcal{X}$, *we call* $x$ *an* $\epsilon$-*local minimum of* $F$ *if* $F(x) \leq F(x') + \epsilon$ *for all* $x' \in N_\mathcal{X}(x)$.

If $\mathcal{X} = \{0, 1\}^n$, we recover the definition of a local minimum of a set function.

A natural approach to solve Problem (1) is to reduce it to DS *set* function minimization, using the same reduction as in submodular minimization (Bach, 2019, Section 4.4), and then apply the algorithms of Narasimhan & Bilmes (2005); Iyer & Bilmes (2012); El Halabi et al. (2023). However, this strategy is more expensive than solving the problem directly, even when $F$ is submodular. We discuss this in Appendix C.

**Continuous relaxation** We adopt a more direct approach to solve Problem (1), which generalizes the approach of El Halabi et al. (2023) for set functions. In particular, we relax the DS problem to an equivalent DC program, using the continuous extension (Definition 2.2) introduced in Bach (2019), then apply a variant of DCA to it.

Recall that minimizing a submodular function $F$ is equivalent to minimizing its continuous extension $f_\downarrow$ (Proposition 2.3-b). We now observe that this equivalence continues to hold even when $F$ is not submodular.

**Proposition 4.4.** *For any normalized function* $F : [0 : k - 1]^n \to \mathbb{R}$, *we have*

$$\min_{x \in [0:k-1]^n} F(x) = \min_{X \in [0,1]_\downarrow^{n \times (k-1)}} f_\downarrow(X). \quad (10)$$

*Moreover, if* $x^*$ *is a minimizer of* $F$ *then* $\Theta^{-1}(x^*)$ *is a minimizer of* $f_\downarrow$, *and if* $X^*$ *is a minimizer of* $f_\downarrow$ *then* $\text{Round}_F(X^*)$ *is a minimizer of* $F$.

Recall that $\Theta$ is the bijection between $\{0, 1\}_\downarrow^{n \times (k-1)}$ and $[0 : k - 1]^n$ defined in (7). The proof follows from the two properties of $f_\downarrow$ in Proposition 2.3-a,c, and is given in

Appendix F.3. By Proposition 2.3-f, Problem (1) is then equivalent to

$$\min_{X \in [0,1]_{\downarrow}^{n \times (k-1)}} f_{\downarrow}(X) = g_{\downarrow}(x) - h_{\downarrow}(x), \qquad (11)$$

where $g_{\downarrow}, h_{\downarrow} \in \Gamma_0(\mathbb{R}^{n \times (k-1)})$ by Proposition 2.3-d,g, since $G, H$ are submodular.

---

**Algorithm 1** DCA with local search

---

1: $\epsilon \geq 0, T \in \mathbb{N}, x^0 \in \mathcal{X}, X^0 = \Theta^{-1}(x^0)$
2: **for** $t = 1, \dots, T$ **do**
3:     $\bar{x}^t \in \arg\min_{x \in N_{\mathcal{X}}(x^t)} F(x)$
4:     Choose a common non-increasing permutation $(p, q)$ of $X^t$ and $\Theta^{-1}(\bar{x}^t)$ (preferably row-stable)
5:     Choose $Y^t \in \partial h(X^t)$ corresponding to $(p, q)$
6:     $\tilde{X}^{t+1} \in \arg\min_{X \in [0,1]_{\downarrow}^{n \times (k-1)}} g_{\downarrow}(X) - \langle Y^t, X \rangle_F$
7:     **if** $\tilde{X}^{t+1} \in \{0,1\}_{\downarrow}^{n \times (k-1)}$ **then**
8:         $x^{t+1} = \Theta(\tilde{X}^{t+1}), X^{t+1} = \tilde{X}^{t+1}$
9:     **else**
10:        $x^{t+1} = \text{Round}_F(\tilde{X}^{t+1}), X^{t+1} = \Theta^{-1}(x^{t+1})$
11:     **end if**
12:     **if** $F(x^t) - F(x^{t+1}) \leq \epsilon$ **then**
13:        Stop.
14:     **end if**
15: **end for**

---

**Algorithm** Problem (11) is a DC program on $\mathbb{E} = \mathbb{R}^{n \times (k-1)}$, with $f = f_{\downarrow} + \delta_{[0,1]_{\downarrow}^{n \times (k-1)}} = g - h$, where $g = g_{\downarrow} + \delta_{[0,1]_{\downarrow}^{n \times (k-1)}}$ and $h = h_{\downarrow}$. Applying the standard DCA (9) to it gives a descent method (up to $\epsilon_x$) which converges to a critical point $X^T$ of $g - h$ by Proposition 2.5. A feasible solution $x^T$ to Problem (1) can then be obtained by rounding, i.e., $x^T = \text{Round}_F(X^T) \in [0 : k-1]^n$, which satisfies $F(x^T) \leq f_{\downarrow}(X^T)$, as shown in Proposition 2.3-c.

However, even in the set functions case, $x^T$ is not necessarily an approximate local minimum of $F$, as shown by El Halabi et al. (2023, Example F.1). To address this, the authors proposed two variants of standard DCA that do obtain an approximate local minimum of $F$. One variant, which generalizes the SubSup method of Narasimhan & Bilmes (2005), is too expensive, as it requires trying $O(n)$ subgradients per iteration. While the other variant checks at convergence if the rounded solution is an approximate local minimum, and if not restarts DCA from the best neighboring point. Both can be generalized to our setting. We provide an extension of the second, more efficient, variant and its theoretical guarantees in Appendix I.

We introduce a new efficient variant of DCA, DCA with local search (DCA-LS), in Algorithm 1, which selects a

single subgradient $Y^t$ using a local search step (lines 3-5), ensuring direct convergence to an approximate local minimum of $F$ without restarts. The algorithm maintains a feasible solution $x^t$ to Problem (1) by rounding $\tilde{X}^{t+1}$ or applying the map $\Theta$ if $\tilde{X}^{t+1}$ is integral. Rounding can optionally be applied in the latter case as well. Finding a common permutation on line 4 is always possible. Indeed, the binary matrix $\Theta^{-1}(\bar{x}^t)$ differs from $X^t$ at only one element: $(i, x_i^t + 1)$ if $\bar{x}^t = x^t + e_i$ and $(i, x_i^t - 1)$ if $\bar{x}^t = x^t - e_i$. Thus, we can choose any row-stable non-increasing permutation $(p, q)$ of $X^t$ such that $(p_{d+1}, q_{d+1}) = (i, x_i^t + 1)$ if $\bar{x}^t = x^t + e_i$ and $(p_{d-1}, q_{d-1}) = (i, x_i^t - 1)$ if $\bar{x}^t = x^t - e_i$, where $d = \|X^t\|_0$.

**Computational complexity** The cost of finding the best neighboring point $\bar{x}^t$ is $2n\, \text{EO}_F$. Finding a valid permutation $(p, q)$ on line 4 as discussed above costs $O(nk)$. The corresponding subgradient $Y^t$ can then be computed as described in Proposition 2.3-e in $O(nk\, \text{EO}_H)$. The subproblem on line 6 is a convex problem which can be solved using projected subgradient method or FW methods. Furthermore, like in the set function case, this subproblem is equivalent to a submodular minimization problem. Indeed, we show in Proposition F.4 that the term $\langle Y^t, X \rangle_F$ corresponds to the continuous extension of the normalized modular function $H^t(x) = \sum_{i=1}^{n} \sum_{j=1}^{x_i} Y_{ij}^t$. Hence, by Proposition 2.3-b,f, the subproblem is equivalent to minimizing the submodular function $F^t = G - H^t$. We can thus obtain an integral $\epsilon_x$-solution $\tilde{X}^{t+1} \in \{0,1\}_{\downarrow}^{n \times (k-1)}$, for any $\epsilon_x \geq 0$, in $\tilde{O}(n(kL_{f_{\downarrow}^t}/\epsilon)^2\, \text{EO}_{F^t})$ time using the algorithm of Axelrod et al. (2020). Rounding can be skipped in this case, and $\tilde{X}^{t+1}$ can be directly mapped to $x^{t+1}$ via $\Theta$ in $O(nk)$. So the total cost per iteration of DCA-LS is $\tilde{O}(n(kL_{f_{\downarrow}^t}/\epsilon)^2\, \text{EO}_{F^t} + nk\, \text{EO}_H)$.

The choice of DS decomposition for $F$ affects the runtime of DCA-LS. For instance, looser bounds $\alpha$ and $\beta$ in the generic decomposition from Proposition 3.1 lead to a larger Lipschitz constant $L_{f_{\downarrow}^t}$ and thus a longer runtime (Appendix A).

**Theoretical guarantees** Let $F^*$ be the minimum of Problem (1). Note that the minimum $f^* = F^*$ of the DC program (11) is finite, and that $\{Y^t\}, \{\tilde{X}^{t+1}\}$ are standard DCA iterates. Proposition 2.5-a,b then apply to them. The following theorem relates DCA properties on Problem (11) to ones on Problem (1), showing that DCA-LS is a descent method (up to $\epsilon_x$) which converges to an $(\epsilon + \epsilon_x)$-local minimum of $F$ in $O(1/\epsilon)$ iterations.

**Theorem 4.5.** *Let $\{x^t\}$ be generated by Algorithm 1, where the subproblem on line 6 is solved up to accuracy $\epsilon_x \geq 0$. For all $t \in [T], \epsilon \geq 0$, we have:*

*a)* $F(x^{t+1}) \leq F(x^t) + \epsilon_x.$

b) Let $\{y^i\}_{i=0}^{(k-1)n}$ be the vectors corresponding to the permutation $(p, q)$ from line 4, defined as in Definition 2.2. If $(p, q)$ is row-stable and $F(x^t) - F(x^{t+1}) \leq \epsilon$, then

$$F(x^t) \leq F(y^i) + \epsilon + \epsilon_x \text{ for all } i \in [0 : (k-1)n].$$

c) Algorithm 1 converges to an $(\epsilon + \epsilon_x)$-local minimum of $F$ after at most $(F(x^0) - F^*)/\epsilon$ iterations.

*Proof Sketch.* Item a follows from Proposition 2.5-a and Proposition 2.3-a,c. Item b follows from Proposition 2.5-b and Proposition 2.4, by observing that if $(p, q)$ is row-stable, then it is a common non-increasing permutation for $X^t$ and $\{\Theta^{-1}(y^i)\}_{i=0}^{(k-1)n}$, and thus $Y^t$ is a common subgradient of $h$ at all these points. The choice of $(p, q)$ on line 4 also ensures the same holds for $\Theta^{-1}(\bar{x}^t)$ even when $(p, q)$ is not row-stable. Item c is then obtained by telescoping sums. The full proof is given in Appendix F.5. $\quad\square$

Restricting $(p, q)$ to be row-stable is necessary to ensure that it is also a non-increasing permutation of $\{\Theta^{-1}(y^i)\}_{i=0}^{(k-1)n}$, which is needed for Item b to hold, but not for Items a and c, as discussed in the proof sketch. In the set function case $(k = 2)$, this restriction is unnecessary, since any non-increasing permutation of $X \in \mathbb{R}_{\downarrow}^{n \times (k-1)}$ is trivially row-stable. DCA-LS can be modified to return a solution with a stronger local minimality guarantee, where $F(x^T) \leq F(x) + \epsilon + \epsilon_x$ for all $x \in \mathcal{X}$ such that $\|x - x^T\|_1 \leq c$, for some $c \in \mathbb{N}$, by setting $\bar{x}^t \in \arg\min_{\|x - x^t\|_1 \leq c} F(X)$. This increases the cost of computing $\bar{x}^t$ to $O(n^c \, \mathrm{EO}_F)$.

Theorem 4.5 generalizes the theoretical guarantees of El Halabi et al. (2023) to general discrete domains; recovering the same guarantees in the set function case. In Appendix I, we compare DCA-LS to an extension of the more efficient DCA variant from El Halabi et al. (2023), theoretically and empirically. We show that both variants have similar theoretical guarantees and computational complexity. In practice, however, DCA-LS performs better in some settings, sometimes by a large margin, but is often slower than DCA-Restart.

Other variants (with regularization, acceleration, complete DCA) explored in El Halabi et al. (2023) are also applicable here. When regularization is used, rounding becomes necessary, as explained in Section 3 therein.

**Implications for non-integer domains**  When $\mathcal{X}$ is a general discrete or continuous domain, we can apply DCA-LS to the function $F'$ obtained via the reductions discussed in Section 4.1. Theorem 4.5 then hold for $F'$.

In the discrete case, recall that $F'(x) = F(m(x))$, where $m$ is the map defined in the proof of Corollary 4.2. The guarantee that $x^t$ is an $(\epsilon + \epsilon_x)$-local minimum of $F'$ implies that $m(x^t)$ is an $(\epsilon + \epsilon_x)$-local minimum of $F$, in the sense

that modifying any coordinate $i \in [n]$ to its previous or next value on the grid $\mathcal{X}_i$ does not reduce $F$ by more than $\epsilon + \epsilon_x$. In the continuous case, assuming again $\mathcal{X} = [0, 1]^n$, recall that $F'(x) = F(x/(k-1))$, where $k = \lceil L/\epsilon' \rceil + 1$ for some $\epsilon' > 0$, and $L$ is the Lipschitz constant of $F$ with respect to the $\ell_\infty$-norm. Let $\tilde{x}^t = x^t/(k-1)$. Then the local minimality guarantee on $F'$ implies that:

$$F(\tilde{x}^t) \leq F(\tilde{x}^t \pm \tfrac{e_i}{k-1}) + \epsilon + \epsilon_x.$$

This is only meaningful if $\epsilon' > \epsilon + \epsilon_x$, since the Lipschitz continuity of $F$ already ensures $F(\tilde{x}^t) - F(\tilde{x}^t \pm \tfrac{e_i}{k-1}) \leq L/(k-1) \leq \epsilon'$.

# 5. Experiments

We evaluate our proposed method, DCA-LS (Algorithm 1), on two applications: integer least squares and integer compressive sensing, where $\mathcal{X} \subseteq \mathbb{Z}^n$. We compare it with state-of-the-art baselines. We use Pairwise-FW (Lacoste-Julien & Jaggi, 2015b) to solve the submodular minimization subproblem at line 6. Results averaged over 100 runs are shown in Figure 1, with error bars for standard deviations. Experimental setups for each application are described below, with additional details given in Appendix G. The code is available at https://github.com/SamsungSAILMontreal/cont-diffsubmin.

## 5.1. Integer Least Squares

We consider the problem of recovering an integer-valued vector $x^\natural \in \mathcal{X}$ from noisy linear measurements $b = Ax^\natural + \xi$, where $A \in \mathbb{R}^{m \times n}$ is a given measurement matrix with $m \geq n$, and $\xi \in \mathbb{R}^m$ is a Gaussian noise vector $\xi \sim \mathcal{N}(0, \sigma^2 \mathbf{I}_m)$. One approach for recovering $x^\natural$ is to solve an integer least squares (ILS) problem $\min_{x \in \mathcal{X}} F(x) = \|Ax - b\|_2^2$. This is an integer BCQP with $Q = A^\top A$ and $c = -2A^\top b$. Thus, we use the DS decomposition given in Section 3.2. ILS problems arise in many applications, including wireless communications (Damen et al., 2003), image processing (Blasinski et al., 2012), and neural network quantization (Frantar & Alistarh, 2022).

We sample $x^\natural$ uniformly from $\mathcal{X} = \{-1, 0, 2, 3\}^n$ with $n = 100$, draw the entries of $A$ i.i.d from $\mathcal{N}(0, 1)$, and vary $m$ from $n$ to $2n$. The noise variance $\sigma^2$ is set to achieve a target signal-to-noise ratio (SNR$_{\text{dB}}$) of 20 dB. We include as baselines ADMM (Takapoui et al., 2020), Optimal Brain Quantizer (OBQ) (Frantar & Alistarh, 2022), and the relax-and-round (RAR) heuristic which solves the relaxed problem on $[-1, 3]^n$, then rounds the solution to the nearest point in $\mathcal{X}$. DCA-LS and ADMM are initialized with the RAR solution, and OBQ with the relaxed solution. We obtain an optimal solution $x^*$ using Gurobi (Gurobi Optimization, LLC, 2024) by rewriting the problem as a binary QP. We report in Figure 1 (top) three evaluation metrics; recovery probability

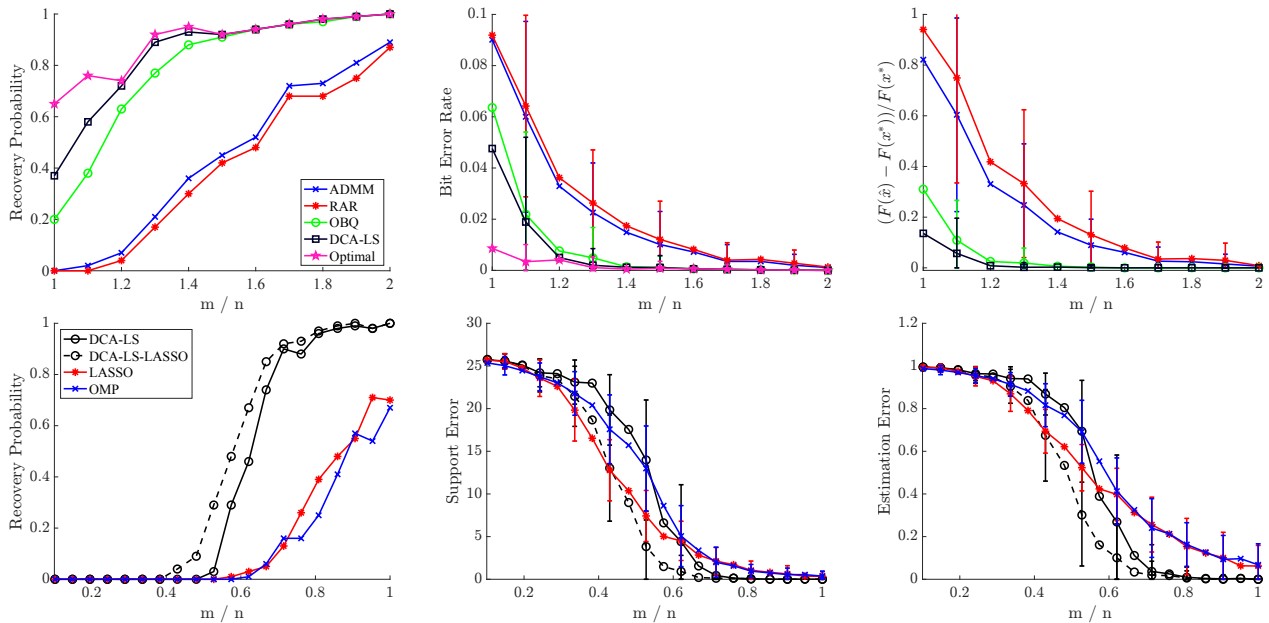

Figure 1: Performance results, averaged over 100 runs, for the integer least squares with $\text{SNR}_{\text{dB}} = 20$ dB, $n = 100$ (top) and for integer compressed sensing with $\text{SNR}_{\text{dB}} = 8$ dB, $n = 256$ and $s = 26 = \lceil 0.1n \rceil$ (bottom).

(ratio of exactly recovered signals over number of runs), bit error rate $\text{BER}(\hat{x}) = \|\hat{x} - x^\natural\|_0 / n$, and relative objective gap with respect to $x^*$. We also compared with Babai's nearest plane algorithm (Babai, 1986, 2nd procedure), but excluded it here as it had worse objective gap and BER than all baselines, and similar recovery probability to ADMM.

We observe that DCA-LS outperforms baselines on all metrics, and performs on par with the optimal solution starting from $m/n = 1.2$. OBQ matches the performance of DCA-LS from $m/n = 1.5$, while ADMM and RAR are both considerably worse across all metrics. In Appendix H.1, we include results using larger $n = 400$, as well as another setup where we fix $m = n$ and vary the $\text{SNR}_{\text{dB}}$. Gurobi cannot be used for $n = 400$ as it is too slow. DCA-LS outperforms other baselines on all metrics in these settings too.

### 5.2. Integer Compressed Sensing

We consider an integer compressed sensing problem, where the goal is to recover an $s$-sparse integer vector $x^\natural$ from noisy linear measurements with $m \leq n$ and $s \ll n$. This problem arises in applications like wireless communications, collaborative filtering, and error correcting codes (Fukshansky et al., 2019). The signal can be recovered by solving $\min_{x \in \mathcal{X}} F(x) = \|Ax - b\|_2^2 + \lambda \|x\|_0$ with $\lambda > 0$. This is a special case of the sparse learning problem in Section 3.2, with $q = 0$. We use the same DS decomposition therein.

We set $\mathcal{X} = \{-1, 0, 1\}^n$, $n = 256$, $s = 26 = \lceil 0.1n \rceil$, and draw $A$ i.i.d from $\mathcal{N}(0, 1/m)$, with $m$ varied from 26 to $n$. We choose a random support for $x^\natural$, then sample its non-zero

entries i.i.d uniformly from $\{-1, 1\}$. The noise variance $\sigma^2$ is set to achieve an $\text{SNR}_{\text{dB}}$ of 8 dB. We consider as baselines orthogonal matching pursuit (OMP) (Pati et al., 1993) and a variant of LASSO (Tibshirani, 1996) with the additional box constraint $x \in [-1, 1]^n$, solved using FISTA (Beck & Teboulle, 2009). Since both methods return solutions outside of $\mathcal{X}$, we round the solutions to the nearest vector in $\mathcal{X}$. As a non-convex method, the performance of DCA-LS is highly dependent on its initialization. We thus consider two different intialization, one with $x^0 = 0$ (DCA-LS) and one with the solution from the box LASSO variant (DCA-LS-LASSO). We evaluate methods on three metrics; recovery probability, support error $|\text{supp}(\hat{x}) \Delta \text{supp}(x^\natural)|$, and estimation error $\|\hat{x} - x^\natural\|_2 / \|x^\natural\|_2$, where $\Delta$ is the symmetric difference. Figure 1 (bottom) reports the best values, as $\lambda$ is varied from 1 to $10^{-5}$ in DCA-LS and FISTA, and the sparsity of the OMP solution is varied from 1 to $\lceil 1.5s \rceil$.

DCA-LS and DCA-LS-LASSO significantly outperform baselines in recovery probability. While DCA-LS outperforms OMP in all metrics, it performs worse than LASSO in estimation and support errors for $m/n < 0.6$, but overtakes it after. DCA-LS-LASSO outperforms baselines on all metrics, except for $m/n \approx (0.25, 0.4)$, where it slightly lags behind LASSO in estimation and support errors. These correspond to cases where $x^\natural \neq x^*$. Indeed, recall that DCA-LS is a descent method (up to $\epsilon$) so DCA-LS-LASSO is guaranteed to obtain an objective value at least as good as LASSO. In Appendix H.3, we include results for $s = 13 = \lceil 0.05n \rceil$, and for another setup where we fix $m/n = 0.5$ and vary the $\text{SNR}_{\text{dB}}$. We observe similar trends in these settings too.

## Acknowledgements

We thank Xiao-Wen Chang for helpful discussions. George Orfanides was partially supported by NSERC CREATE INTER-MATH-AI and a Fonds de recherche du Québec Doctoral Research Scholarship B2X-326710.

## Impact Statement

This paper presents work whose goal is to advance the field of Machine Learning. There are many potential societal consequences of our work, none which we feel must be specifically highlighted here.

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

## A. Effect of looser $\alpha$ and $\beta$ bounds on performance

In this section, we discuss how the bounds $\alpha$ and $\beta$ in the generic decomposition given in Proposition 3.1 affect the runtime of Algorithm 1.

As discussed in Section 4.2, the runtime of Algorithm 1, particularly for solving the submodular minimization subproblem at each iteration $t$, depends on the Lipschitz constant $L_{f_\downarrow^t}$ of $f_\downarrow^t$. Recall that $f_\downarrow^t$ is the continuous extension of $F^t = G - H^t$, where $H^t$ is the modular function whose continuous extension is $\langle Y^t, X \rangle_F$. The decomposition in Proposition 3.1 defines $G = F + \frac{\alpha}{\beta}\tilde{H}$ and $H = \frac{\alpha}{\beta}\tilde{H}$, for some normalized strictly submodular function $\tilde{H}$. Let $\tilde{Y}^t = \frac{\beta}{\alpha}Y^t$, we can thus write

$$f_\downarrow^t(X) = f_\downarrow(X) + \tfrac{\alpha}{\beta}(\tilde{h}_\downarrow(X) - \langle \tilde{Y}^t, X \rangle_F).$$

Looser bounds $\alpha$ and $\beta$ result in a larger Lipschitz constant $L_{f_\downarrow^t}$, and thus a longer runtime. To show that, we provide upper and lower bounds on $L_{f_\downarrow^t}$ which grow with $\frac{|\alpha|}{\beta}$. For any $X, X' \in \mathbb{R}_\downarrow^{n \times (k-1)}$, we have

$$|f_\downarrow^t(X) - f_\downarrow^t(X')| \leq |f_\downarrow(X) - f_\downarrow(X')| + \tfrac{|\alpha|}{\beta}|\tilde{h}_\downarrow(X) - \tilde{h}_\downarrow(X')| + \tfrac{|\alpha|}{\beta}|\langle \tilde{Y}^t, X - X' \rangle_F|$$
$$\leq (L_{f_\downarrow} + 2\tfrac{|\alpha|}{\beta}L_{\tilde{h}_\downarrow^t})\|X - X'\|_F$$

Hence, $L_{f_\downarrow^t} \leq L_{f_\downarrow} + 2\frac{|\alpha|}{\beta}L_{\tilde{h}_\downarrow^t}$. Note that $L_{f_\downarrow}$ and $L_{\tilde{h}_\downarrow^t}$ are independent of $\alpha$ and $\beta$. This upper bound indeed grows with $\frac{|\alpha|}{\beta}$.

Since $Y^t \in \partial h_\downarrow(X^t)$, we know from Proposition 7-b Bach (2019) that $h_\downarrow(X) \geq \langle Y^t, X \rangle$ for any $X \in \mathbb{R}_\downarrow^{n \times (k-1)}$ with equality if and only if $Y^t \in \partial h_\downarrow(X)$. We can thus write $h_\downarrow(X^t) = \langle Y^t, X^t \rangle_F$, or equivalently $\tilde{h}_\downarrow(X^t) = \langle \tilde{Y}^t, X^t \rangle_F$. Taking $X' = X^t$, we get

$$|f_\downarrow^t(X) - f_\downarrow^t(X^t)| = |f_\downarrow(X) - f_\downarrow(X^t) + \tfrac{\alpha}{\beta}(\tilde{h}_\downarrow(X) - \langle \tilde{Y}^t, X \rangle_F)|$$
$$\geq \tfrac{|\alpha|}{\beta}|\tilde{h}_\downarrow(X) - \langle \tilde{Y}^t, X \rangle_F| - |f_\downarrow(X) - f_\downarrow(X^t)|.$$

Hence, for any $X \in \mathbb{R}_\downarrow^{n \times (k-1)}$, we have

$$L_{f_\downarrow^t} \geq \frac{|f_\downarrow^t(X) - f_\downarrow^t(X^t)|}{\|X - X^t\|_F}$$
$$\geq \frac{|\alpha|}{\beta}\frac{|\tilde{h}_\downarrow(X) - \langle \tilde{Y}^t, X \rangle_F|}{\|X - X^t\|_F} - \frac{|f_\downarrow(X) - f_\downarrow(X^t)|}{\|X - X^t\|_F}.$$

Choosing any $X \neq X^t$ such that $Y^t \notin \partial h_\downarrow(X)$, we get $\frac{|\tilde{h}_\downarrow(X) - \langle \tilde{Y}^t, X \rangle_F|}{\|X - X^t\|_F} > 0$. Such $X$ must exists, since otherwise $h_\downarrow(X) = \langle Y^t, X \rangle_F$ for all $X \in \mathbb{R}_\downarrow^{n \times (k-1)}$, and $\tilde{H}(X) = \sum_{i=1}^n \sum_{j=1}^{x_i} \tilde{Y}_{ij}^t$ by Proposition F.4. Since $\tilde{H}$ is strictly submodular, it can't be modular unless $n = 1$, otherwise Ineq (3) will not hold strictly. Then, this lower bound indeed also grows with $\frac{|\alpha|}{\beta}$. We can thus conclude that $L_{f_\downarrow^t}$ itself grows with $\frac{|\alpha|}{\beta}$.

## B. Connection to Related Non-convex Classes

In this section, we discuss the relation of DS functions to other related classes of non-convex functions, namely DC and discrete DC functions.

### B.1. Relation to DC Functions

We remark that the class of DS functions is not a subclass of DC functions. Since DC functions have continuous domains, it is evident that DS functions defined on discrete domains are not DC. For continuous domains, any univariate function which is not DC can serve as a counter-example, since as mentioned earlier when $n = 1$ any function is modular. The function $F : [0, 1] \to \mathbb{R}$ given by $F(x) = 1 - \sqrt{|x - 1/2|}$ is one such example, given in de Oliveira (2020, Example 7). We provide below another counter-example for any $n$.

**Proposition B.1.** *The function $F : \mathcal{X} \to \mathbb{R}$ defined as $F(x) = \|x\|_q^q$ with $q \in [0, 1)$ ($F(x) = \|x\|_0$ if $q = 0$) and where each $\mathcal{X}_i$ is a closed interval, is a modular function but not a DC function, whenever $0 \in \operatorname{int} \mathcal{X}$.*

*Proof.* We first note that $F$ is a separable function, hence it is modular. Next, we recall that convex functions are locally Lipschitz on the interior of their domain (Mordukhovich & Nam, 2023, Corollary 2.27), i.e., for all $x \in \operatorname{int} \mathcal{X}$ there exists a neighborhood $U$ of $x$ such that $F$ is Lipschitz continuous on $U \cap \operatorname{int} \mathcal{X}$. Since the difference of locally Lipschitz functions is also locally Lipschitz, DC functions are then also locally Lipschitz on the interior of their domain.

We show that $F$ is not locally Lipschitz at $0 \in \operatorname{int} \mathcal{X}$, and thus it cannot be DC. We show that there exists $\{x^k\} \subset \mathcal{X}$ such that $\{x^k\} \to 0$ and $\frac{|F(x^k)|}{\|x^k\|_2}$ is unbounded. In particular, we take $x^k = \frac{1}{k} e_1$. For any $q \in [0, 1)$, we have

$$\frac{|F(x^k)|}{\|x^k\|_2} = \frac{(1/k)^q}{1/k} = k^{1-q}.$$

Taking the limit as $k \to \infty$, we see $\frac{|F(x^k) - F(0)|}{\|x^k - 0\|_2} = \frac{|F(x)|}{\|x\|_2} \to \infty$. $\qquad\square$

The function in Proposition B.1 is used as a sparsity regularizer in sparse learning problems, which we showed in Section 3.2 are instances of Problem (1).

## B.2. Relation to Discrete DC Functions

Next, we discuss the relation between DS functions on discrete domains and discrete DC functions considered in Maehara & Murota (2015). Therein, a function $F : \mathbb{Z}^n \to \overline{\mathbb{Z}}$ is called *discrete DC* if it can be written as $F = G - H$, where $G, H : \mathbb{Z}^n \to \overline{\mathbb{Z}}$ are $L^\natural$-convex or $M^\natural$-convex functions. We recall the definitions of both types of functions, see e.g., Murota (1998, Chap. 1, Eq. (1.33) and (1.45)).

**Definition B.2** ($L^\natural$-convex). A function $F : \mathbb{Z}^n \to \overline{\mathbb{R}}$ is called $L^\natural$-*convex* if it satisfies the discrete midpoint convexity; that is, for all $x, y \in \mathbb{Z}^n$, we have

$$F(x) + F(y) \geq F\left(\left\lfloor \frac{x+y}{2} \right\rfloor\right) + F\left(\left\lceil \frac{x+y}{2} \right\rceil\right). \tag{12}$$

**Definition B.3** ($M^\natural$-convex). A function $F : \mathbb{Z}^n \to \overline{\mathbb{R}}$ is called $M^\natural$-*convex* if, for any $x, y \in \mathbb{Z}^n$ and $i \in [n]$ such that $x_i > y_i$, we have

$$F(x) + F(y) \geq \min\{F(x - e_i) + F(y + e_i), \min_{j \in [n], y_i > x_i} F(x - e_i + e_j) + F(y + e_i - e_j)\} \tag{13}$$

$L^\natural$-convex functions are a proper subclass of submodular functions on $\mathbb{Z}^n$ (Maehara & Murota, 2015, Section 2.1), while $M^\natural$-convex functions are a proper subclass of supermodular functions on $\mathbb{Z}^n$ (Murota & Shioura, 2001, Theorem 3.8, Example 3.10). It follows then that any discrete DC function is a DS function on $\mathbb{Z}^n$. This is of course not surprising in light of Proposition 3.1 (which still holds for functions defined on $\mathbb{Z}^n$). Corollary B.4 is the analogue of Proposition 3.1 for discrete DC functions, which was stated in Maehara & Murota (2015, Corollary 3.9-1) for functions defined on finite subsets of $\mathbb{Z}^n$.

**Corollary B.4.** *Given $F : \mathcal{X} \to \mathbb{Z}$, where each $\mathcal{X}_i \subseteq \mathbb{R}$ is a finite set, there exists an $L^\natural$ - $L^\natural$ discrete DC function $F' : \mathbb{Z}^n \to \mathbb{Z}$ such that $F(x) = F'(m^{-1}(x))$ for all $x \in \mathcal{X}$, where $m : \prod_{i=1}^n [0 : k_i - 1] \to \mathcal{X}$ is a bijection with $k_i = |\mathcal{X}_i|$.*

*Proof.* Given $F : \mathcal{X} \to \mathbb{R}$, where each $\mathcal{X}_i \subseteq \mathbb{R}$ is a finite set, we define $\bar{F} : \prod_{i=1}^n [0 : k_i - 1] \to \mathbb{R}$ as $\bar{F}(x) = F(m(x))$ where $m : \prod_{i=1}^n [0 : k_i - 1] \to \mathcal{X}$ is the bijection defined in Corollary 4.2. Since $\bar{F}$ is defined on a finite subset of $\mathbb{Z}^n$, by Corollary 3.9-1 in Maehara & Murota (2015), there exists $F' : \mathbb{Z}^n \to \mathbb{Z}$ such that $F' = G' - H'$, with $L^\natural$-convex functions $G', H' : \mathbb{Z}^n \to \mathbb{Z}$, and $F'(z) = \bar{F}(z)$ for all $z \in \prod_{i=1}^n [0 : k_i - 1]$ and $F'(z) = 0$ otherwise. We thus have for all $x \in \mathcal{X}$, $F(x) = \bar{F}(m^{-1}(x)) = F'(m^{-1}(x))$. $\qquad\square$

Since $L^\natural$-convex functions are submodular on $\mathbb{Z}^n$ and submodularity is preserved by restriction, then the restrictions $G'_{|\mathcal{X}}, H'_{|\mathcal{X}} : \mathcal{X} \to \mathbb{Z}$ of $G', H'$ to $\mathcal{X}$ are submodular, and the restriction $F'_{|\mathcal{X}} : \mathcal{X} \to \mathbb{Z}$ of $F'$ to $\mathcal{X}$ is a DS function with $F = F'_{|\mathcal{X}} = G'_{|\mathcal{X}} - H'_{|\mathcal{X}}$. Note however that $F$ itself is not necessarily an $L^\natural$ - $L^\natural$ discrete DC function, since $L^\natural$-convexity is

not preserved by restriction. Nevertheless, minimizing $F$ on $\mathcal{X}$ is equivalent to minimizing $F'$ on $\mathbb{Z}^n$ if $\min_{x \in \mathcal{X}} F(x) \leq 0$, which can be assumed wlog. The class of discrete DC functions, in particular $L^\natural$ - $L^\natural$ functions, is thus essentially equivalent to the class of *integer valued* DS functions on discrete domains.

**Applications which are DS but not discrete DC:** In Section 3.2, we outlined two classes of applications which have natural DS objectives. We now show that both types of applications do not have a natural discrete DC decomposition, for general discrete domains, even if we ignore the assumption in Maehara & Murota (2015) that $F$ is integer valued.

For QPs of the form $\min_{x \in \mathcal{X}} F(x) = \frac{1}{2} x^\top Q x + c^\top x$, $F$ admits a natural discrete DC decomposition, when $\mathcal{X}_i$'s are *uniform* finite sets. However, this is not the case if $\mathcal{X}_i$ is a non-uniform finite set for some $i$, as in the integer least squares experiments we consider in Section 5.1. Such cases also arise for example in non-uniform quantization of neural networks (Kim et al., 2025). To see this, note that any $L^\natural$-convex function $F : \mathbb{Z} \to \mathbb{Z} \cup \{+\infty\}$ should satisfy for all $x, y \in \operatorname{dom} F$, $\lceil \frac{x+y}{2} \rceil, \lfloor \frac{x+y}{2} \rfloor \in \operatorname{dom} F$. If we're minimizing a quadratic over a uniform grid, we can map from the uniform grid to $\prod_i [0, k_i - 1]$ and the resulting function will still be a quadratic. We can then easily decompose the objective into $L^\natural$ - $L^\natural$ DC function, as we show in Proposition B.5. However, if the domain is a non-uniform grid, the resulting function is no longer a quadratic.

**Proposition B.5.** *Any quadratic objective $F : \mathbb{Z}^n \to \mathbb{Z}$ defined as $F(x) = x^\top Q x$ can be written as the difference of two $L^\natural$ functions $G(x) = x^\top (Q^- + D) x$ and $H(x) = -x^\top (-Q^+ + D) x$, where $Q^- = \min\{Q, 0\}$, $Q^+ = \max\{Q, 0\}$, and $D$ is a diagonal matrix with $D_{ii} = \max\{-\sum_j Q_{ij}^-, -\sum_j Q_{ji}^-, \sum_j Q_{ij}^+, \sum_j Q_{ji}^+\}$.*

*Proof.* Let $\nabla^2 F$ be the $L^\natural$ Hessian of $F$ (Maehara & Murota, 2015, Section 3.2), then $\nabla^2 F = Q + Q^\top$. To see this note that for any $x \in \mathbb{Z}^n, i \in [n]$, we have:

$$F(x + e_i) - F(x) = e_i^\top Q x + x^\top Q e_i + Q_{ii}. \tag{14}$$

Then for any $j \neq i$, we get:

$$\nabla_{ij}^2 F(x) = F(x + e_j + e_i) - F(x + e_j) - (F(x + e_i) - F(x)) \tag{15}$$

$$= e_i^\top Q(x + e_j) + (x + e_j)^\top Q e_i + Q_{ii} - (e_i^\top Q x + x^\top Q e_i + Q_{ii}) \tag{16}$$

$$= Q_{ij} + Q_{ji} \tag{17}$$

and thus

$$\nabla_{ii}^2 F(x) = F(x + \mathbb{1} + e_i) - F(x + \mathbb{1}) - (F(x + e_i) - F(x)) - \sum_{j \neq i} \nabla_{ij}^2 F(x) \tag{18}$$

$$= e_i^\top Q(x + \mathbb{1}) + (x + \mathbb{1})^\top Q e_i + Q_{ii} - (e_i^\top Q x + x^\top Q e_i + Q_{ii}) - \sum_{j \neq i} \nabla_{ij}^2 F(x) \tag{19}$$

$$= e_i^\top Q \mathbb{1} + \mathbb{1}^\top Q e_i - \sum_{j \neq i} (Q_{ij} + Q_{ji}) \tag{20}$$

$$= 2 Q_{ii}. \tag{21}$$

Hence, $\nabla^2 F = Q + Q^\top$.

Similarly the $L^\natural$ Hessian of $G$ is $\nabla^2 G(x) = Q^- + (Q^-)^\top + 2D$ and of $H$ os $\nabla^2 H(x) = -Q^+ - (Q^+)^\top + 2D$. Note that for any $j \neq i$ we have $\nabla_{ij}^2 G(x) \leq 0$ and $\nabla_{ij}^2 H(x) \leq 0$. We also have

$$\sum_{j=1}^{n} \nabla_{ij}^2 G(x) = \sum_{j=1}^{n} (Q_{ij}^- + Q_{ji}^-) + 2D_{ii} \geq 0.$$

Similarly,

$$\sum_{j=1}^{n} \nabla_{ij}^2 H(x) = -\sum_{j=1}^{n} (Q_{ij}^+ + Q_{ji}^+) + 2D_{ii} \geq 0.$$

Hence, by the Hessian characterization of $L^\natural$-convexity (Maehara & Murota, 2015, Theorem 3.7), $G$ and $H$ are $L^\natural$-convex. □

For sparse learning problems $\min_{x \in \mathcal{X}} \ell(x) + \lambda \|x\|_q^q$, where $q \in [0, 1)$, $\lambda \geq 0$, and $\ell$ is a smooth function, we show below that $\|x\|_q^q$ is not a discrete convex function.

**Proposition B.6.** *The function $f(x) = \|x\|_q^q$, $q \in [0, 1)$ (when $q = 0$, we let $f = \| \cdot \|_0$), is not $L^\natural$-convex nor $M^\natural$-convex over $\mathbb{Z}^n$*

*Proof.* As a simple counterexample for the case $q \in (0, 1)$, take $n = 1$ so that $f : \mathbb{Z} \to \mathbb{R}_+$ is given as $f(x) = |x|^q$. Taking $x = 0$ and $y = 2$, we have

$$f(x) + f(y) \geq f\left(\left\lfloor \frac{x + y}{2} \right\rfloor\right) + f\left(\left\lceil \frac{x + y}{2} \right\rceil\right) \tag{22}$$

$$\iff 2^q + 0 \geq 2 \tag{23}$$

which, for any $q \in [0, 1)$, is a contradiction.

For $n = 1$, $M^\natural$-convexity definition simplifies to (the minimum over empty set is $+\infty$)

$$f(x) + f(y) \geq f(x - 1) + f(x + 1),$$

for all $x > y$. We again consider the counterexample $x = 2, y = 0$. When $q = 0$, we have $f(x) + f(y) = 1 + 0 < 2 = f(x - 1) + f(x + 1)$ which contradicts the $M^\natural$-convexity definition.

Similarly, for $q \in (0, 1)$, we have: $f(x) + f(y) = 2^q + 0 < 1 + 3^q = f(x - 1) + f(x + 1)$, where the inequality follows from the fact that $|x + y|^q < |x|^q + |y|^q$, which implies that $2^q = |3 - 1|^q < 3^q + 1^q$. $\qquad \square$

Of course, in both cases, we can still use the generic decomposition given in Maehara & Murota (2015, Corollary 3.9-1), this would yield $G$ and $H$ with large Lipschitz constants, which would lead to slow optimization.

## C. Reduction to set function case

In this section, we discuss the reduction of a submodular minimization problem $\min_{x \in \mathcal{X}} F(x)$ over the integer lattice $\mathcal{X} = [0 : k - 1]^n$ to the minimization of a submodular set function over $\{0, 1\}^{n \times (k-1)}$, given in Bach (2019, Section 4.4). As mentioned in Section 4.2, the same reduction can be used to reduce Problem (1) to DS set function minimization.

We define the map $\pi : \{0, 1\}^{n \times (k-1)} \to \{0, 1\}_\downarrow^{n \times (k-1)}$ as

$$\pi(X)_{i,j} = \begin{cases} 1 & \text{if there exists } j' \geq j, \ X_{i,j'} = 1, \\ 0 & \text{otherwise,} \end{cases} \tag{24}$$

for all $i \in [n], j \in [k - 1]$, i.e., $\pi(X)$ is the smallest row non-increasing matrix such that $X \leq \pi(X)$. Given $B_i \geq 0$ for all $i \in [n]$, define the set-function $\tilde{F} : \{0, 1\}^{n \times (k-1)} \to \mathbb{R}$ as

$$\tilde{F}(X) = F(\Theta(\pi(X))) + \sum_{i=1}^{n} B_i \|\pi(X)_i - X_i\|_1 \tag{25}$$

where $\pi(X)_i, X_i$ are the $i^{\text{th}}$ rows of $\pi(X), X$, respectively. Then minimizing $\tilde{F}$ is equivalent to minimizing $F$; $\min_{X \in \{0,1\}^{n \times (k-1)}} \tilde{F}(X) = \min_{x \in [0:k-1]^n} F(x)$. To ensure $\tilde{F}$ is submodular, we choose $B_i > 0$ such that

$$|F(y) - F(x)| \leq \sum_{i=1}^{n} B_i |x_i - y_i|, \tag{26}$$

for all $x \leq y \in \mathcal{X}$ (Bach, 2019, Section 4.4). We thus reduced minimizing $F$ to minimizing a submodular set function. Similarly, we can reduce Problem (1) with $\mathcal{X} = [0 : k - 1]^n$ to the following DS set function minimization:

$$\min_{X \in \{0,1\}^{n \times (k-1)}} \tilde{G}(X) - \tilde{H}(X),$$

where $\tilde{G}, \tilde{H}$ are submodular set functions defined as in (25).

However, as discussed in Bach (2019, Section 4.4), this strategy adds extra parameters $B_i$ which are often unkown, and leads to slower optimization due to the larger Lipschitz constant $L_{\tilde{f}_\downarrow}$ of the continuous extension $\tilde{f}_\downarrow$ of $\tilde{F}$ (which reduces to Lovász extension in this case). In particular, while we can bound $L_{f_\downarrow} \leq \sqrt{(k-1)n} \max_i B_i$ (Proposition 2.3-g), the bound for $\tilde{f}_\downarrow$ is $(2k-3)$ times larger, i.e., $L_{\tilde{f}_\downarrow} \leq (2k-3)\sqrt{(k-1)n} \max_i B_i$. The time complexity of both submodular minimization algorithms in Bach (2019, Section 5) and the one in Axelrod et al. (2020) (the fastest known inexact submodular set function minimization algorithm) scales quadratically with the Lipschitz constant of the continuous extension: $\tilde{O}((\frac{nkL_{f_\downarrow}}{\epsilon})^2 \mathrm{EO}_F)$ and $\tilde{O}(n(kL_{f_\downarrow}/\epsilon)^2 \, \mathrm{EO}_F)$, respectively. Using the reduction $\tilde{F}$ increases their complexity by at least a factor of $O(k^2)$. Moreover, the cost $\mathrm{EO}_{\tilde{F}}$ of evaluating $\tilde{F}$ can also be larger; if computed via Eq. (25) it incurs an additional $O(nk)$ time, i.e., $\mathrm{EO}_{\tilde{F}} = \mathrm{EO}_F + O(nk)$. This also applies in our setting, where $F$ is a DS function, since DCA-LS requires solving a submodular minimization at each iteration, which will be similarly slower using the reduction.

To validate this empirically, we compare the minimization of a submodular quadratic function $F$ with that of its reduction $\tilde{F}$:

$$\min_{x \in [-1,1]^n} F(x) = \frac{1}{2} x^\top Q x, \tag{27}$$

We set $n = 50$ and construct a symmetric matrix $Q \in \mathbb{R}^{n \times n}$ as follows: for $i < j$, draw $Q_{i,j} \sim \mathcal{U}_{[-1/n,0]}$ where $\mathcal{U}$ denotes the uniform distribution; and draw diagonal elements as $Q_{i,i} \sim \mathcal{U}_{[0,1]}$. The resulting $Q$ has non-positive off-diagonal entries, making $F$ submodular by Proposition 2.1-a. This construction also helps ensure the minimizer is randomly located within the lattice $\mathcal{D}$.

As discussed in Section 4.1, we can convert Problem (27) to a submodular minimization problem over $\mathcal{X} = [0 : k-1]^n$ by discretization. In particular, we define $F' : \mathcal{X} \to \mathbb{R}$ as $F'(x) = F(m(x))$ with $m : \mathcal{X} \to [-1,1]^n$, $m_i(x) = \frac{2}{k-1} \cdot x_i - 1$. Then $F'$ is submodular by Proposition 4.1.

To apply the reduction, we must find positive $B_i$'s which satisfy (26) for the function $F'$. One valid choice is to set $B_i = B$ for all $i \in [n]$, where $B$ is the Lipschitz constant of $F'$ with respect to the $\ell_1$-norm. We start by bounding the Lipschitz constant of $F$ over $[-1,1]^n$ with respect to the $\ell_1$-norm. For all $x \in [-1,1]^n$, we have

$$\|\nabla F(x)\|_\infty = \|Qx\|_\infty \leq \|Q\|_{\infty \to \infty} \|x\|_\infty \leq \|Q\|_{\infty \to \infty}, \tag{28}$$

Hence, $F$ is $L$-Lipschitz continuous with respect to the $\ell_1$-norm with $L = \|Q\|_{\infty \to \infty}$, which is equal to the maximum $\ell_1$-norm of the rows of $Q$. For any $x, y \in \mathcal{X}$, we have

$$|F'(x) - F'(y)| = |F(m(x)) - F(m(y))| \tag{29}$$
$$\leq L\|m(x) - m(y)\|_1 \tag{30}$$
$$= \frac{2L}{k-1} \cdot \|x - y\|_1. \tag{31}$$

Hence, $F'$ is Lipschitz continuous in the $\ell_1$-norm over $[-1,1]^n$ with constant $B = \frac{2L}{k-1}$. We can now define the reduction $\tilde{F}$ of $F'$ as in (25). Then Problem (27) reduces to a submodular set-function minimization problem:

$$\min_{X \in \{0,1\}^{n \times (k-1)}} \tilde{F}(X). \tag{32}$$

A minimizer $X^*$ of (32) yields a minimizer $x^* \in \mathcal{X}$ of $F'$ by setting $x^* = \Theta(X^*)$.

We numerically compare minimizing $F'$ and $\tilde{F}$ over their respective domains to get a minimizer of Problem (27). We minimize $F'$ and $\tilde{F}$ using pairwise FW, which we run for 200 iterations. Let $\hat{x}, \tilde{x} \in \mathcal{X}$ denote the resulting solutions, where $\tilde{x}$ is obtained by applying $\Theta$ to the output of pairwise FW with $\tilde{F}$. We test different discretization levels $k = 100, 200, \ldots, 500$ and initialize both methods at zero (i.e. $0 \in \mathbb{R}^n$ for $F'$ and $0 \in \mathbb{R}^{n \times (k-1)}$ for $\tilde{F}$). For each value of $k$, we run 50 random trials. We evaluate the two approaches on three metrics: the relative objective gap between $\hat{x}$ and $\tilde{x}$, the duality gap given in Bach (2019, Section 5.2), and the running time. Average results are shown in Figure 2. We observe that $\tilde{x}$ has a worse duality gap and objective value than $\hat{x}$ and a longer running time, after the same number of iterations. This confirms that minimizing $\tilde{F}$ is significantly slower than minimizing $F'$, both in terms of convergence speed (due to the larger Lipschitz constant $L_{\tilde{f}_\downarrow}$) and per iteration runtime (due to the higher evaluation cost of $\tilde{F}$).

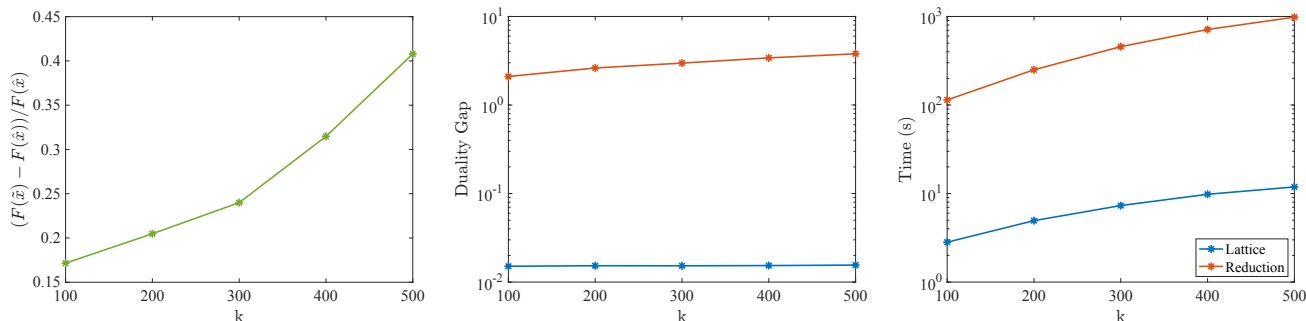

Figure 2: Results averaged over 50 runs comparing the direct minimization of $F$ on the lattice $\mathcal{X}$ (Lattice) and the minimization of its reduction $\tilde{F}$ on $\{0,1\}^{n \times (k-1)}$ (Reduction) on Problem (27).

## D. Proofs of Section 2

### D.1. Proof of Proposition 2.3-g

We prove here the bounds on the Lipschitz constant of the continuous extension of a submodular function given in Proposition 2.3-g. The general bound was stated in Bach (2019, Section 5.1), and follows directly from the definition of the subgradients (see Proposition 2.3-e).

**Lemma D.1.** *Given a normalized submodular function $F : [0 : k-1]^n \to \mathbb{R}$, its continuous extension $f_\downarrow$ is $L_{f_\downarrow}$-Lipschitz continuous with respect to the Frobenius norm, with $L_{f_\downarrow} \leq \sqrt{(k-1)n}B$ where $B = \max_{x, x+e_i \in [0:k-1]^n} |F(x+e_i) - F(x)|$. If $F$ is also non-decreasing, then $L_{f_\downarrow} \leq F((k-1)\mathbb{1})$.*

*Proof.* For any $X \in \mathbb{R}_\downarrow^{n \times (k-1)}$, let $(p, q)$ be a non-increasing permutation of $X$ and $Y$ the corresponding subgradient of $f_\downarrow$ at $X$, defined as in Proposition 2.3-e. The bound for the general case follows directly by bounding the Frobenius norm of $Y$.

$$
\begin{aligned}
\|Y\|_F^2 &= \sum_{i=1}^{n(k-1)} Y_{p_i, q_i}^2 \\
&= \sum_{i=1}^{n(k-1)} \left( F(y^i) - F(y^{i-1}) \right)^2 \\
&\leq n(k-1)B^2
\end{aligned}
$$

Moreover, if $F$ is non-decreasing, then $Y_{i,j} \geq 0$ for all $i, j$. By telescoping sums, we get:

$$
\begin{aligned}
\|Y\|_F \leq \|Y\|_{1,1} &= \sum_{i=1}^{n(k-1)} Y_{p_i, q_i} \\
&= \sum_{i=1}^{n(k-1)} F(y^i) - F(y^{i-1}) \\
&= F(y^{n(k-1)}) - F(y^0) = F((k-1)\mathbb{1}) - F(0).
\end{aligned}
$$

Since $F$ is normalized, then we have $\|Y\|_F \leq F((k-1)\mathbb{1})$. □

## E. Proofs of Section 3

### E.1. Proofs of Proposition 3.1 and Example 3.2

**Proposition 3.1.** *Given any normalized $F : \mathcal{X} \to \mathbb{R}$ where each $\mathcal{X}_i \subset \mathbb{R}$ is finite, there exists a decomposition $F = G - H$, where $G, H : \mathcal{X} \to \mathbb{R}$ are normalized (strictly) submodular functions.*

*Proof.* We give a constructive proof, similar to the one provided in Iyer & Bilmes (2012, Lemma 3.1) for set functions.

Let $\alpha \leq 0$ be any lower bound on how much $F$ violates Eq. (3), i.e. for any $x \in \mathcal{X}$, $i \neq j \in [n]$, and $a_i, a_j > 0$ such that $x + a_i e_i, x + a_j e_j \in \mathcal{X}$, we have

$$F(x + a_i e_i) - F(x) - F(x + a_i e_i + a_j e_j) + F(x + a_j e_j) \geq \alpha. \tag{33}$$

Since $F$ is finitely valued and defined on finitely many points, such a lower bound must exist. For example, we can use $\alpha = -4 \max_{x \in \mathcal{X}} |F(x)|$. Note that $\alpha < 0$ unless $F$ is already submodular, in which case we can take $G = F$ and $H = 0$.

We now choose any normalized *strictly* submodular function $\tilde{H} : \mathcal{X} \to \mathbb{R}$. We provide an example in Example 3.2. $\tilde{H}$ satisfies Eq. (3) with strict inequality. As each $\mathcal{X}_i$ is finite, there exists $\beta > 0$ such that

$$\tilde{H}(x + a_i e_i) - \tilde{H}(x) - \tilde{H}(x + a_i e_i + a_j e_j) + \tilde{H}(x + a_j e_j) \geq \beta. \tag{34}$$

whenever $x \in \mathcal{X}$, $i \neq j \in [n]$, and $a_i, a_j > 0$ are such that $x + a_i e_i, x + a_j e_j \in \mathcal{X}$.

We next construct $G$ and $H$ based on $\tilde{H}$. In particular, we define $G = F + \frac{|\alpha|}{\beta} \tilde{H}$ and $H = \frac{|\alpha|}{\beta} \tilde{H}$. Since $F$ and $\tilde{H}$ are normalized, $G$ and $H$ are also normalized. Also, $H$ is strictly submodular, since multiplying by a positive scalar preserves strict submodularity. We check that $G$ is also submodular. For any $x \in \mathcal{X}$, $i \neq j \in [n]$, and $a_i, a_j > 0$ such that $x + a_i e_i, x + a_j e_j \in \mathcal{X}$, we have

$$\begin{aligned}
G(x + a_i e_i) - G(x) - G(x + a_i e_i + a_j e_j) + G(x + a_j e_j) &= F(x + a_i e_i) - F(x) - F(x + a_i e_i + a_j e_j) \\
&\quad + F(x + a_j e_j) + \frac{|\alpha|}{\beta} \big( \tilde{H}(x + a_i e_i) - \tilde{H}(x) \\
&\quad - \tilde{H}(x + a_i e_i + a_j e_j) + \tilde{H}(x + a_j e_j) \big) \\
&\geq \alpha + \frac{|\alpha|}{\beta} \cdot \beta = 0.
\end{aligned} \tag{35}$$

Hence, $G$ is submodular by Equation (3). This yields the desired decomposition $F = G - H$, with $G$ and $H$ normalized submodular functions. Choosing $\alpha < 0$ which strictly satisfies the inequality (33) (which is again always possible), we can further guarantee that $G$ is strictly submodular, since the inequality (35) will become strict too. $\qquad\square$

**Example 3.2.** Let $H : \mathbb{R}^n \to \mathbb{R}$ be the quadratic function $H(x) = -\frac{1}{2} x^\top J x$, where $J$ is the matrix of all ones, and define $\tilde{H} : \mathcal{X} \to \mathbb{R}$ as $\tilde{H}(x) = H(x) - H(x^{\min})$. Then, $\tilde{H}$ is a normalized strictly submodular function.

*Proof.* Since each $\mathcal{X}_i$ is a compact subset of $\mathbb{R}$, there exists an interval $[a_i, b_i]$ such that $\mathcal{X}_i \subseteq [a_i, b_i]$. The function $H$ is twice-differentiable on $\prod_{i=1}^{n} [a_i, b_i]$, and the Hessian of $H$ has negative entries. Hence, it is strictly submodular on $\prod_{i=1}^{n} [a_i, b_i]$ by Proposition 2.1-b. Since strict submodularity is preserved by restricting $H$ to $\mathcal{X}$, $\tilde{H}$ is also strictly submodular. It is also normalized by definition. $\qquad\square$

### E.2. Proof of Proposition 3.3

**Proposition 3.3.** *Given any normalized $F : \mathcal{X} \to \mathbb{R}$ where each $\mathcal{X}_i$ is a closed interval, if $F$ is differentiable and there exist $L_F \geq 0$ such that $\frac{\partial F}{\partial x_i}(x) - \frac{\partial F}{\partial x_i}(x + a e_j) \geq -L_F a$, for all $x \in X$, $i \neq j$, $a > 0$, $x_j + a \in \mathcal{X}_j$, then there exists a decomposition $F = G - H$, where $G, H : \mathcal{X} \to \mathbb{R}$ are normalized (strictly) submodular functions.*

*Proof.* The proof is again constructive. Note that $L_F > 0$ unless $F$ is already submodular, in which case we can take $G = F$ and $H = 0$.

We now choose any normalized *strictly* submodular function $\tilde{H} : \mathcal{X} \to \mathbb{R}$, which is differentiable and there exists $L_{\tilde{H}} > 0$ such that

$$\frac{\partial \tilde{H}}{\partial x_i}(x) - \frac{\partial \tilde{H}}{\partial x_i}(x + a e_j) \geq L_{\tilde{H}} a,$$

for all $x \in \mathcal{X}$, $i \neq j$, $a > 0$ such that $x_j + a \in \mathcal{X}_j$. We provide an example in Example 3.2.

We next construct $G$ and $H$ based on $\tilde{H}$. In particular, we define $G = F + \frac{L_F}{L_{\tilde{H}}} \tilde{H}$ and $H = \frac{L_F}{L_{\tilde{H}}} \tilde{H}$. Since $F$ and $\tilde{H}$ are normalized, $G$ and $H$ are also normalized. Also, $H$ is strictly submodular, since multiplying by a positive scalar preserves

strict submodularity. We check that $G$ is also submodular. For any $x \in \mathcal{X}$, $j \in [n]$, $a > 0$ such that $i \neq j$ and $x_j + a \in \mathcal{X}_j$, we have

$$\frac{\partial G}{\partial x_i}(x) - \frac{\partial G}{\partial x_i}(x + ae_j) = \frac{\partial F}{\partial x_i}(x) - \frac{\partial F}{\partial x_i}(x + ae_j) + \frac{L_F}{L_{\tilde{H}}}\left(\frac{\partial \tilde{H}}{\partial x_i}(x) - \frac{\partial \tilde{H}}{\partial x_i}(x + ae_j)\right)$$

$$\geq -L_F a + \frac{L_F}{L_{\tilde{H}}} L_{\tilde{H}} a = 0. \tag{36}$$

Hence, $G$ is submodular by Proposition 2.1-a. This yields the desired decomposition $F = G - H$, with $G$ and $H$ normalized submodular functions. Choosing $L_F > 0$ which satisfies the strict inequality $\frac{\partial F}{\partial x_i}(x) - \frac{\partial F}{\partial x_i}(x + ae_j) > -L_F a$, we can further guarantee that $G$ is strictly submodular, since the inequality (36) will become strict too. $\qquad \square$

## F. Proofs of Section 4

### F.1. Proof of Proposition 4.1

**Proposition 4.1.** *Given $\mathcal{X} = \prod_{i=1}^n \mathcal{X}_i$, $\mathcal{X}' = \prod_i \mathcal{X}_i'$, with compact sets $\mathcal{X}_i, \mathcal{X}_i' \subset \mathbb{R}$, let $F : \mathcal{X} \to \mathbb{R}$ and $m : \mathcal{X}' \to \mathcal{X}$ be a monotone function such that $[m(x)]_i = m_i(x_i)$. If $F$ is submodular, then the function $F' : \mathcal{X}' \to \mathbb{R}$ given by $F'(x) = F(m(x))$ is submodular. Moreover, if $m$ is strictly monotone, then $F$ is submodular iff $F'$ is submodular.*

*Proof.* For any monotone $m$, if $F$ is submodular, then for any $x, y \in \mathcal{X}$ we have

$$\begin{aligned} F'(\min\{x, y\}) + F'(\max\{x, y\}) &= F\big(m(\min\{x, y\})\big) + F\big(m(\min\{x, y\})\big) \\ &= F\big(\min\{m(x), m(y)\}\big) + F\big(\max\{m(x), m(y)\}\big) \\ &\leq F\big(m(x)\big) + F\big(m(y)\big) \qquad\qquad (F \text{ is submodular}) \\ &= F'(x) + F'(y). \end{aligned}$$

Hence, $F'$ is submodular. If $F$ is modular, the above inequality becomes an equality, implying that $F'$ is modular too.

Moreover if $m$ is strictly monotone, $m$ is invertible, and we may write $F(y) = F'(m^{-1}(y))$. We show that the converse claim is also true in this case. Indeed, since the inverse of a strictly monotone function is again strictly monotone, we have by the first claim that if $F'$ is (sub)modular then $F$ is also (sub)modular. $\qquad \square$

### F.2. Discretization of a non-Lipschitz function example

In Section 4.1, we assumed that the function $F : [0, 1]^n \to \mathbb{R}$ is Lipschitz continuous to bound its discretization error. However, Lipschitz continuity is not always necessary. We show below how to handle an example of a non-Lipschitz continuous function. In particular, we consider a function $F : [-B, B]^n \to \mathbb{R}$ given by $F(x) = \ell(x) + \|x\|_q^q$, with $q \in [0, 1)$, where $\ell' : [0, 1]^n \to \mathbb{R}$ is a DS function $L$-Lipschitz continuous w.r.t the $\ell_\infty$-norm. The modular function $\|x\|_q^q$ is not Lipschitz continuous on the domain $[-B, B]^n$.

We first observe that the function $|x|^q$ satisfies a notion of Lipschitz continuity, for $x \neq 0$.

**Lemma F.1.** *The function $F : \mathbb{R} \to \mathbb{R}$ defined as $F(x) = |x|^q$ with $q \in (0, 1)$ satisfies*

$$\big||y|^q - |x|^q\big| \leq 2 \max\{|y|, |x|\}^{q-1}|y - x|,$$

*for every two scalars $x, y \neq 0$ of same sign.*

*Proof.* We assume without loss of generality that $|y| \geq |x| > 0$. Let $t > 0$ be the smallest integer such that $2^t q \geq 1$. Note that

$$|y|^q - |x|^q = \frac{|y|^{2q} - |x|^{2q}}{|y|^q + |x|^q} < \frac{|y|^{2q} - |x|^{2q}}{|y|^q}$$

Applying this relation recursively yields

$$|y|^q - |x|^q = \frac{|y|^{2^t q} - |x|^{2^t q}}{|y|^{q \sum_{i=0}^{t-1} 2^i}} \leq \frac{2^t q |z|^{2^t q - 1}|y - x|}{|y|^{q(2^t - 1)}} \leq 2|y|^{q-1}|y - x|, \tag{37}$$

where the first inequality follows by the mean value theorem with $z$ a scalar between $x$ and $y$, and the 2nd inequality holds since by definition of $t$, we have $2^{t-1}q < 1 \leq 2^t q$ and thus $|z|^{2^t q - 1} \leq |y|^{2^t q - 1}$. $\qquad\square$

We shift and scale the function $F$ to obtain a function $F' : [0,1]^n \to \mathbb{R}$ defined as $F'(x) = F(B(2x - \mathbb{1}))$, which remains DS by Proposition 4.1.

**Proposition F.2.** *Given $q \in (0,1), B > 0$ and $F : [0,1]^n \to \mathbb{R}$ defined as $F(x) = \ell'(x) + \|B(2x - \mathbb{1})\|_q^q$, where $\ell' : [0,1]^n \to \mathbb{R}$ is a DS function $L$-Lipschitz continuous w.r.t the $\ell_\infty$-norm, let $k = 2\max\{\lceil 2L/\epsilon \rceil, \lceil B(8n/\epsilon)^{1/q} \rceil\} + 1$ and define the function $F' : [0:k-1]^n \to \mathbb{R}$ as $F'(x) = F(x/(k-1))$. Then $F'$ is DS and we have*

$$\min_{x' \in [0:k-1]^n} F'(x') - \epsilon/2 \leq \min_{x \in [0,1]^n} F(x) \leq \min_{x \in [0:k-1]^n} F'(x). \tag{38}$$

*The same also holds if $q = 0$, with $k = 2\lceil L/\epsilon \rceil + 1$.*

*Proof.* Let $\phi : [0:k-1]^n \to [0,1]^n$ denote the map $\phi(x) = x/(k-1)$, so $F'(x) = F(\phi(x))$. Since $F$ is DS and $\phi$ is monotone, then $F'$ is also DS by Proposition 4.1.

Let $\pi : [0,1]^n \to [0:k-1]^n$ be defined for all $x \in [0,1]^n$ such that for each $i \in [n]$, $[\pi(x)]_i = \frac{k-1}{2}$ if $x_i = 1/2$, and $[\pi(x)]_i = t$ where $\phi(t)$ is the nearest point to $x_i$ for $t$ in $[0:k-1] \setminus \frac{k-1}{2}$, if $x_i \neq 1/2$. Note that $k-1$ is an even number hence $\frac{k-1}{2} \in [0:k-1]$.

For any $x \in [0,1]^n$, let $x' = \pi(x)$ be the vector obtained from the above map. Note that for any $i$, if $x_i = \frac{1}{2}$ then $\phi(x'_i) = \frac{1}{2}$, otherwise we have $|x_i - \phi(x'_i)| \leq \frac{1}{k-1}$ and $|\phi(x'_i) - \frac{1}{2}| \geq \frac{1}{k-1}$. Let $G_i(x_i) = |B(2x_i - 1)|^q$, we show that for all $i$ where $x_i \neq 1/2$, we have

$$|G_i(x_i) - G_i(\phi(x'_i))| \leq 4B(\tfrac{2B}{k-1})^{q-1}|x_i - \phi(x'_i)|.$$

Note that the nearest points $\phi(t)$ to $x_i = 1/2$, for $t$ in $[0:k-1] \setminus \frac{k-1}{2}$, are $\phi(t) = 1/2 \pm 1/(k-1)$ which are equidistant from $1/2$. Hence $B(2x_i - 1)$ and $B(2\phi(x'_i) - 1)$ will always have the same sign, and $\max\{B(2x_i - 1), B(2\phi(x'_i) - 1)\} \geq \frac{2B}{k-1}$. By Lemma F.1, we thus have

$$|G_i(x_i) - G_i(\phi(x'_i))| \leq 2(\tfrac{2B}{k-1})^{q-1}|B(2x_i - 1) - B(2\phi(x'_i) - 1)| \leq 4B(\tfrac{2B}{k-1})^{q-1}|x_i - \phi(x'_i)|.$$

Putting everything together we get:

$$
\begin{aligned}
|F(x) - F'(x')| &= |F(x) - F(\phi(x'))| \\
&= |\ell'(x) - \ell'(\phi(x'))| + \sum_{x_i \neq 1/2} |G_i(x_i) - G_i(\phi(x'_i))| \\
&\leq L\|x - \phi(x')\|_\infty + 4B(\tfrac{2B}{k-1})^{q-1} \sum_{x_i \neq 1/2} |x_i - \phi(x'_i)| \\
&\leq \frac{L}{k-1} + \frac{4B(\tfrac{2B}{k-1})^{q-1}n}{k-1} \\
&\leq \frac{L}{k-1} + 2n(\tfrac{2B}{k-1})^q \\
&\leq \frac{\epsilon}{4} + \frac{\epsilon}{4} = \frac{\epsilon}{2}.
\end{aligned}
$$

The same bound holds for $q = 0$ by noting that $B(2x_i - 1)$ and $B(2\phi(x'_i) - 1)$ have the same support, hence $\|B(2x_i - 1)\|_0 = \|B(2\phi(x'_i) - 1)\|_0$ (no discretization error). $\qquad\square$

### F.3. Proof of Proposition 4.4

**Proposition 4.4.** *For any normalized function $F : [0:k-1]^n \to \mathbb{R}$, we have*

$$\min_{x \in [0:k-1]^n} F(x) = \min_{X \in [0,1]^{n \times (k-1)}_\downarrow} f_\downarrow(X). \tag{10}$$

*Moreover, if $x^*$ is a minimizer of $F$ then $\Theta^{-1}(x^*)$ is a minimizer of $f_\downarrow$, and if $X^*$ is a minimizer of $f_\downarrow$ then $\mathrm{Round}_F(X^*)$ is a minimizer of $F$.*

*Proof.* Let $x^*$ be a minimizer of $F$. By Proposition 2.3-a, we have $F(x^*) = f_\downarrow(\Theta^{-1}(x^*))$. Since $\Theta^{-1}(x^*) \in [0,1]_\downarrow^{n \times (k-1)}$, then

$$\min_{x \in [0:k-1]^n} F(x) \geq \min_{X \in [0,1]_\downarrow^{n \times (k-1)}} f_\downarrow(X).$$

Let $X^*$ is a minimizer of $f_\downarrow$. By Proposition 2.3-c, we have $F(\mathrm{Round}_F(X^*)) \leq f_\downarrow(X^*)$. Since $\mathrm{Round}_F(X^*) \in [0 : k-1]^n$, we have

$$\min_{x \in [0:k-1]^n} F(x) \leq \min_{X \in [0,1]_\downarrow^{n \times (k-1)}} f_\downarrow(X).$$

$\square$

### F.4. Continuous extension of a normalized modular function

In this section, we derive the form of a normalized modular function and its continuous extension.

**Lemma F.3.** *A function $F : [0 : k-1]^n \to \mathbb{R}$ is modular and normalized iff there exists $W \in \mathbb{R}^{n \times (k-1)}$ such that*

$$F(x) = \sum_{i=1}^{n} \sum_{j=1}^{x_i} W_{i,j} \text{ for all } x \in [0 : k-1]^n. \tag{39}$$

*Proof.* We first recall that a function $F$ is modular iff it is separable under addition (Topkis, 1978, Theorem 3.3).

( $\implies$ ) If $F$ is modular, then it is separable under addition, i.e. there exist $F_i : [0 : k-1] \to \mathbb{R}$, $i \in [n]$, such that $F(x) = \sum_{i=1}^{n} F_i(x_i)$ for any $x \in [0 : k-1]^n$. As $F$ is normalized, we also have that $F(0) = \sum_{i=1}^{n} F_i(0) = 0$.

Now, define $W \in \mathbb{R}^{n \times (k-1)}$ such that $W_{i,j} = F_i(j) - F_i(j-1)$, for $i \in [n]$, $j \in [k-1]$. We thus obtain

$$\begin{aligned}
F(x) &= \sum_{i=1}^{n} F_i(x_i) - F(0) \\
&= \sum_{i=1}^{n} F_i(x_i) - F_i(0) \\
&= \sum_{i=1}^{n} \sum_{j=1}^{x_i} F_i(j) - F_i(j-1) && \text{(by telescoping sums)} \\
&= \sum_{i=1}^{n} \sum_{j=1}^{x_i} W_{i,j}.
\end{aligned}$$

( $\impliedby$ ) If there exists a matrix $W \in \mathbb{R}^{n \times (k-1)}$ such that $F(x) = \sum_{i=1}^{n} \sum_{j=1}^{x_i} W_{ij}$, then $F$ is separable under addition, and hence modular. Indeed, we can define the functions $F_i : [0 : k-1] \to \mathbb{R}$, $i \in [n]$, by $F_i(x) = \sum_{j=1}^{x_i} W_{i,j}$. Then $F(x) = \sum_{i=1}^{n} F_i(x_i)$ for any $x \in [0 : k-1]^n$. $F$ is also normalized, since $F(0) = 0$. $\square$

**Proposition F.4.** *A function $F : [0 : k-1]^n \to \mathbb{R}$ is modular and normalized, with $F(x) = \sum_{i=1}^{n} \sum_{j=1}^{x_i} W_{ij}$ for all $x \in [0 : k-1]^n$ iff its continuous extension $f_\downarrow$ is linear on its domain, with $f_\downarrow(X) = \langle W, X \rangle_F$ for all $X \in \mathbb{R}_\downarrow^{n \times (k-1)}$.*

*Proof.* Recall from Lemma F.3, that $F$ is modular and normalized iff it can be written as $F(x) = \sum_{i=1}^{n} \sum_{j=1}^{x_i} W_{ij}$ for some $W \in \mathbb{R}^{n \times (k-1)}$.

( $\impliedby$ ) If $F$ has a linear continuous extension such that $f_\downarrow(X) = \langle W, X \rangle_F$ for all $X \in \mathbb{R}_\downarrow^{n \times (k-1)}$ for some $W \in \mathbb{R}^{n \times (k-1)}$. Then, for any $x \in [0 : k-1]^n$, let $X = \Theta^{-1}(x)$. By Proposition 2.3-a and the definition of the $\Theta^{-1}$ (7b), we have that

$$F(x) = f_\downarrow(X) = \langle W, X \rangle_F = \sum_{i=1}^{n} \sum_{j=1}^{k-1} W_{i,j} X_{i,j} = \sum_{i=1}^{n} \sum_{j=1}^{x_i} W_{ij}. \tag{40}$$

( $\implies$ ) Suppose $F$ is modular and normalized. By Definition 2.2, the continuous extension of $-F$ is equal to $-f_\downarrow$ on its domain. Since $F$ and $-F$ are submodular, then both $f_\downarrow$ and $-f_\downarrow$ are convex by Proposition 2.3-d, which implies that $f_\downarrow$ is affine on its domain. Since $F$ is normalized, we have $f_\downarrow(0) = F(0) = 0$ by Proposition 2.3-a, hence $f_\downarrow$ is linear on its domain, i.e,. $f_\downarrow(X) = \langle W', X \rangle_F$ for all $X \in \mathbb{R}_\downarrow^{n \times (k-1)}$ for some $W' \in \mathbb{R}^{n \times (k-1)}$. By the above argument for the reverse direction, we must also have that $W' = W$.

$\square$

### F.5. Proof of Theorem 4.5

Before proving Theorem 4.5, we recall some properties of DCA that apply to iterates of Algorithm 1.

**Proposition F.5.** *Let $\epsilon, \epsilon_x \geq 0$, and $\{X^t\}$, $\{\tilde{X}^t\}$, $\{Y^t\}$ be generated by Algorithm 1, where the subproblem at line 6 is solved up to accuracy $\epsilon_x$. Then for all $t \in \mathbb{N}$, we have:*

*a) $f(\tilde{X}^{t+1}) \leq f(X^t) + \epsilon_x$.*

*b) If $f(X^t) - f(\tilde{X}^{t+1}) \leq \epsilon$, then $X^t$ is an $\epsilon + \epsilon_x$-critical point of $g - h$ with $Y^t \in \partial_{\epsilon+\epsilon_x} g(X^t) \cap \partial h(X^t)$.*

*Proof.* Note that the iterates $\tilde{X}^{t+1}$, $Y^t$ are generated by a DCA step from $X^t$, with subproblem (9b) solved up to accuracy $\epsilon_x$. The proposition follows then directly from Proposition 2.5-a,b. $\square$

**Theorem 4.5.** *Let $\{x^t\}$ be generated by Algorithm 1, where the subproblem on line 6 is solved up to accuracy $\epsilon_x \geq 0$. For all $t \in [T], \epsilon \geq 0$, we have:*

*a) $F(x^{t+1}) \leq F(x^t) + \epsilon_x$.*

*b) Let $\{y^i\}_{i=0}^{(k-1)n}$ be the vectors corresponding to the permutation $(p, q)$ from line 4, defined as in Definition 2.2. If $(p, q)$ is row-stable and $F(x^t) - F(x^{t+1}) \leq \epsilon$, then*

$$F(x^t) \leq F(y^i) + \epsilon + \epsilon_x \text{ for all } i \in [0 : (k-1)n].$$

*c) Algorithm 1 converges to an $(\epsilon + \epsilon_x)$-local minimum of $F$ after at most $(F(x^0) - F^*)/\epsilon$ iterations.*

*Proof.* We first observe that for all $t \in [T]$, we have:

$$F(x^t) - F(x^{t+1}) = f(X^t) - f(X^{t+1}) \geq f(X^t) - f(\tilde{X}^{t+1}). \tag{41}$$

Since by the extension property (Proposition 2.3-a), we have $f(X^{t+1}) = F(x^{t+1})$ and $f(X^t) = F(x^t)$. And by rounding (Proposition 2.3-c), we have $f(X^{t+1}) \leq f(\tilde{X}^{t+1})$. If $\tilde{X}^{t+1}$ is integral and rounding is skipped, this still holds trivially since $X^{t+1} = \tilde{X}^{t+1}$.

(a) This follows from Proposition F.5-a and Eq. (41)

(b) If $F(x^t) - F(x^{t+1}) \leq \epsilon$, then $f(X^t) - f(\tilde{X}^{t+1}) \leq \epsilon$ too, by Eq. (41). By Proposition F.5-b, we thus have $Y^t \in \partial_{\epsilon+\epsilon_x} g(X^t) \cap \partial h(X^t)$. We observe that by definition of $y^i$, if $(p, q)$ is a row-stable non-increasing permutation of $X^t$, then it is also a non-increasing permutation of $\Theta^{-1}(y^i)$, for all $i \in [0 : (k-1)n]$. Hence, $Y^t \in \partial h(\Theta^{-1}(y^i))$ by Proposition 2.3-e, and $Y^t \in \partial_{\epsilon+\epsilon_x} g(X^t) \cap \partial h(\Theta^{-1}(y^i))$. Therefore, by Proposition 2.4 and Proposition 2.3-a,

$$F(x^t) = f(X^t) \leq f(\Theta^{-1}(y^i)) + \epsilon + \epsilon_x = F(y^i) + \epsilon + \epsilon_x, \tag{42}$$

for any $i \in [(k-1)n]$.

(c) We first argue that Algorithm 1 converges ($F(x^t) - F(x^{t+1}) \leq \epsilon$) after at most $(F(x^0) - F^*)/\epsilon$ iterations. By telescoping sums, we have $\sum_{t=0}^{T-1} F(x^t) - F(x^{t+1}) = F(x^0) - F(x^T) \leq F(x^0) - F^*$. Hence,

$$\min_{t \in [T-1]} F(x^t) - F(x^{t+1}) \leq \frac{F(x^0) - F^*}{T}. \tag{43}$$

Taking $T = (F(x^0) - F^*)/\epsilon$, we get that there exists some $t \in [T-1]$ such that

$$F(x^t) - F(x^{t+1}) \leq (F(x^0) - F^*)/T = \epsilon.$$

Since $(p, q)$ is chosen on line 4 to be a common permutation of $X^t$ and $\Theta^{-1}(\bar{x}^t)$, then by the same argument as in Item b, we have at convergence $F(x^t) \leq F(\bar{x}^t) + \epsilon + \epsilon_x$. Since $\bar{x}^t$ is the best neighboring point of $x^t$, this implies that $x^t$ is an $(\epsilon + \epsilon_x)$-local minimum of $F$.

$\square$

## G. Experimental Setup Additional Details

We provide here additional details on the experimental setups used in Section 5. All methods were implemented in MATLAB. To solve the submodular minimization subproblem in DCA-LS (Algorithm 1-line 6), we use the pairwise FW algorithm from Bach (2019), available at `http://www.di.ens.fr/~fbach/submodular_multi_online.zip`. In both experiments, during each DCA iteration, we run pairwise FW for a maximum of 400 iterations, stopping earlier if the duality gap reaches $10^{-4}$. We warm start pairwise FW with the subproblem's solution from the previous DCA iteration.

To achieve a desired target $(\text{SNR}_{\text{dB}})$ of $\alpha \geq 0$, we choose the noise variance $\sigma^2$ using the following formula,

$$\sigma = \sqrt{10^{-\frac{\alpha}{10}} \frac{\|b\|_2^2}{\|\xi'\|_2^2}}, \tag{44}$$

where $\xi' \sim \mathcal{N}(\mathbf{0}, I_m)$, then set the noise vector to $\xi = \sigma \xi'$.

### G.1. Additional Details for Section 5.1

The RAR solution is obtained by first solving the relaxed least squares problem

$$x^{LS} \in \underset{x \in [-1,3]^n}{\operatorname{argmin}} \; F(x) = \|Ax - b\|_2^2 \tag{45}$$

then rounding $x^{LS}$ to the nearest point in $\mathcal{X}$. We implement OBQ according to the pseudocode provided in Frantar & Alistarh (2022, Algorithm 3). But since in our setting we assume no access to $x^\natural$, we initialize OBQ with $x^{LS}$. For the ADMM approach of Takapoui et al. (2020), we use the code from `https://github.com/cvxgrp/miqp_admmm?tab=readme-ov-file`. In DCA-LS, we use a maximum of $T = 50$ outer iterations and set $\epsilon = 10^{-5}$.

For $n = 100$, recall that we obtain an optimal solution using Gurobi 10.0.1 (Gurobi Optimization, LLC, 2024). But since the domain $\mathcal{X} = \{-1, 0, 2, 3\}^n$ has unevenly spaced integers, Gurobi cannot solve it directly. Instead, we reformulate the problem into an equivalent unconstrained binary quadratic program (UBQP)

$$\min_{x \in \{0,1\}^{2n}} x^\top U x + d^\top x \tag{46}$$

where $U = M^\top A^\top A M$, $d^\top = -2(\mathbb{1}^\top A^\top A M + b^\top A M)$ and $M \in \mathbb{R}^{n \times 2n}$ is defined as $M = I_n \otimes [3, 1]$, with $\otimes$ denoting the Kronecker product. To see that the two problems are equivalent, the map $Mx - \mathbb{1}$ defines a bijection between $\{0, 1\}^{2n}$ and $\mathcal{X}$, and that the objective in (46) is equal to $F(Mx - \mathbb{1})$, without the constant terms.

### G.2. Additional Details for Section 5.2

We solve the box constrained variant of Lasso

$$\min_{x \in [-1,1]^n} \|Ax - b\|_2^2 + \lambda \|x\|_1, \tag{47}$$

using the FISTA algorithm from Beck & Guttmann-Beck (2019) with default parameter settings, i.e., 1000 maximum iterations and $\|x^{k+1} - x^k\| \leq 10^{-5}$ stopping criterion (for other parameters see `https://www.tau.ac.il/~becka/solvers/fista`). In DCA-LS, we use a maximum of $T = 25$ outer iterations, and set $\epsilon = 10^{-5}$. In DCA-LS, DCA-LS-LASSO, and FISTA, we use $\lambda = 10^{-i}$ for $i \in [0:5]$, and warm start with the solution of the previous $\lambda$ value (in decreasing order). Our implementation of OMP is adapted from the one provided in Becker. We run OMP for a maximum of $\lceil 1.5s \rceil$ iterations, stopping earlier if the residual reaches $\|Ax - b\|_2 \leq \|\xi\|_2$

## H. Additional Experiments

We present here additional experimental results for the applications presented in Section 5.

### H.1. Additional Integer Least Squares Experiments

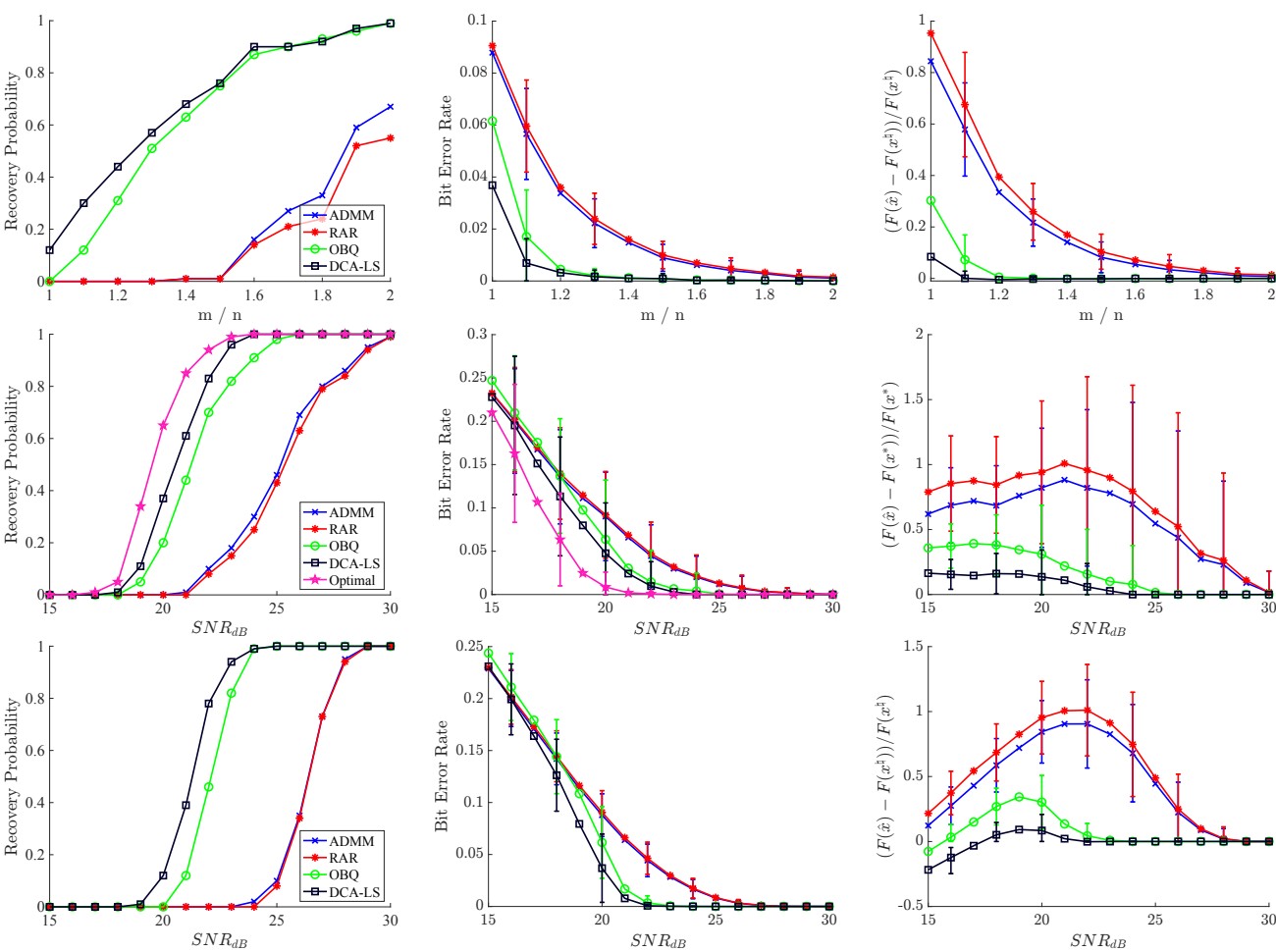

Figure 3: Performance results, averaged over 100 runs, for integer least squares experiments: Varying $m$ with $n = 400$ and $\mathrm{SNR}_{\mathrm{dB}} = 20$ (top), varying $\mathrm{SNR}_{\mathrm{dB}}$ with $m = n = 100$ (middle), and varying $\mathrm{SNR}_{\mathrm{dB}}$ with $m = n = 400$ (bottom).

We repeat the integer least squares experiment from Section 5.1 with $n = 400$. We also consider another setup where we fix $m = n$ and vary the $\mathrm{SNR}_{\mathrm{dB}}$. Performance results for both are reported in Figure 3. When using a signal of length $n = 400$, we compute the relative objective gap with respect to $x^{\natural}$, as it is very time consuming to obtain $x^*$ using Gurobi. We again observe that DCA-LS outperforms baselines on all metrics. For $n = 100$, DCA-LS performs on par with the optimal solution $x^*$ starting from $\mathrm{SNR}_{\mathrm{dB}} = 24$ dB. OBQ matches the performance of DCA-LS from $\mathrm{SNR}_{\mathrm{dB}} = 26$ dB onward. For $n = 400$ OBQ matches the performance of DCA-LS from $\mathrm{SNR}_{\mathrm{dB}} = 24$ dB. Note that the relative objective gap in this case can be negative, since the true signal $x^{\natural}$ is not necessarily the optimal solution for the integer least squares problem. Indeed, we see that this is the case when the noise is high; DCA-LS and OBQ obtain a better solution for $\mathrm{SNR}_{\mathrm{dB}} \leq 18$ dB and 15 dB, respectively.

### H.2. Times for Integer Least Squares Experiments

We report the average running time of the compared methods on the integer least squares experiments from Section 5.1 and Appendix H.1 in Figure 4. The reported times for DCA-LS and ADMM do not include the time for the RAR initialization, and for OBQ do not include the time for the relaxed solution initialization (which has similar time as RAR). We observe that

DCA-LS is significantly faster that the optimal Gurobi solver, even for the small problem instance ($n = 100$). For the larger instance ($n = 400$), Gurobi is already too slow to be practical. While our algorithm is slower than heuristic baselines, it remains efficient, with a maximum runtime of 100 seconds for $m = n \approx 400$.

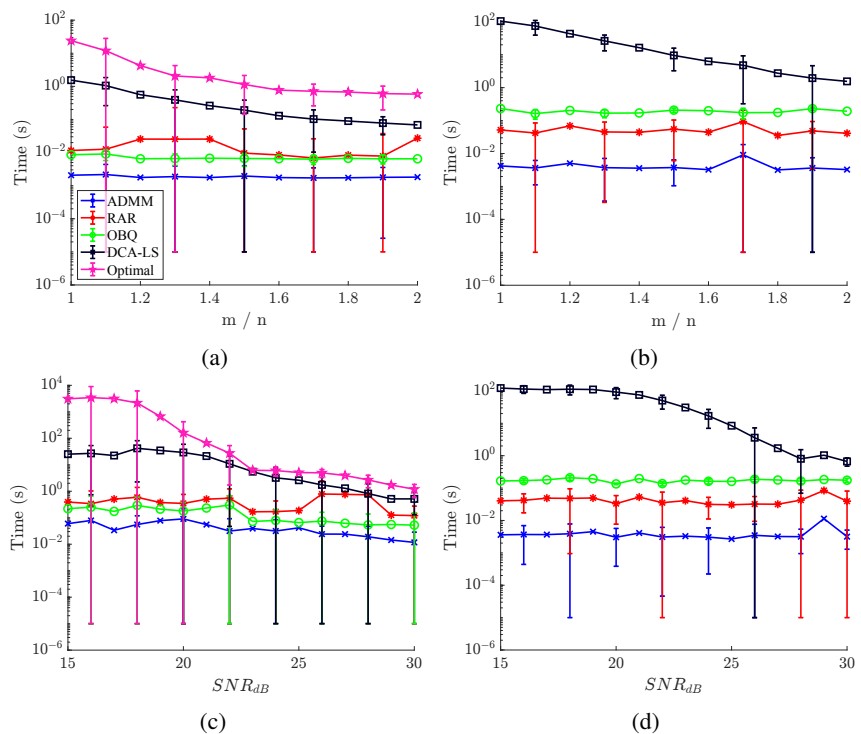

(a)

(b)

(c)

(d)

Figure 4: Running times (log-scale), averaged over 100 runs, for integer least squares experiments: (a) Varying $m$ with $n = 100$ and $\mathrm{SNR}_{\mathrm{dB}} = 20$, (b) varying $m$ with $n = 400$ and $\mathrm{SNR}_{\mathrm{dB}} = 20$, (c) varying $\mathrm{SNR}_{\mathrm{dB}}$ and $m = n = 100$, and (d) varying $\mathrm{SNR}_{\mathrm{dB}}$ and $m = n = 400$. Optimal solution is computed using Gurobi.

### H.3. Additional Integer Compressed Sensing experiments

We repeat the integer compressed sensing experiment from Section 5.2, with another sparsity level $s = 13 = \lceil 0.05n \rceil$. We also consider another setup where we fix $m/n = 0.5$ and vary the $\mathrm{SNR}_{\mathrm{dB}}$. Performance results for both are reported in Figure 5. When varying the number of measurements for $s = 13$, we again see that DCA-LS-LASSO outperforms all baselines and DCA-LS across all metrics. DCA-LS also outperforms the baselines in terms of recovery probability, but lags slightly behind in estimation error and support error when $m/n \approx 0.4$. Similar trends emerge in the experiments where $s = 13$ and the amount of additive noise is varied. Again, we see that DCA-LS-LASSO and DCA-LS outperform in terms of recovery probability, and when $\mathrm{SNR}_{\mathrm{dB}} \geq 3$, outperform in estimation error and support error. Interestingly, when we decrease the sparsity to $s = 26$, we see that DCA-LS recovers signals earlier than the baselines, but does not reach a 100% recovery rate once $\mathrm{SNR}_{\mathrm{dB}} = 20$. Similar to the experiment in Section 5.2, we see that DCA-LS-LASSO significantly outperforms all other methods, especially in terms of recovery probability.

### H.4. Times for Integer Compressed Sensing Experiments

We report the average running time of the compared methods on the integer compressed sensing experiments from Section 5.2 and Appendix H.3 in Figure 6. For LASSO and DCA-LS, the reported times correspond to the sum of running times for the 6 $\lambda$ values tried, $\lambda = 1, 0.1, \ldots, 10^{-5}$. The reported time for DCA-LS-LASSO include the time for the LASSO initialization. We also report the running time of LASSO and DCA-LS for each $\lambda$ separately at $m/n \approx 0.2$ and $0.5$ ($m = 50$ and $m = 123$, respectively) with $\mathrm{SNR}_{\mathrm{dB}} = 8$ in Figure 7.

We observe that DCA-LS and DCA-LS-LASSO are slower than baselines, but they remain efficient, with a maximum total runtime of $\sim 37$ seconds for DCA-LS and $\sim 71$ seconds for DCA-LS-LASSO, for solving the problem for all 6 $\lambda$

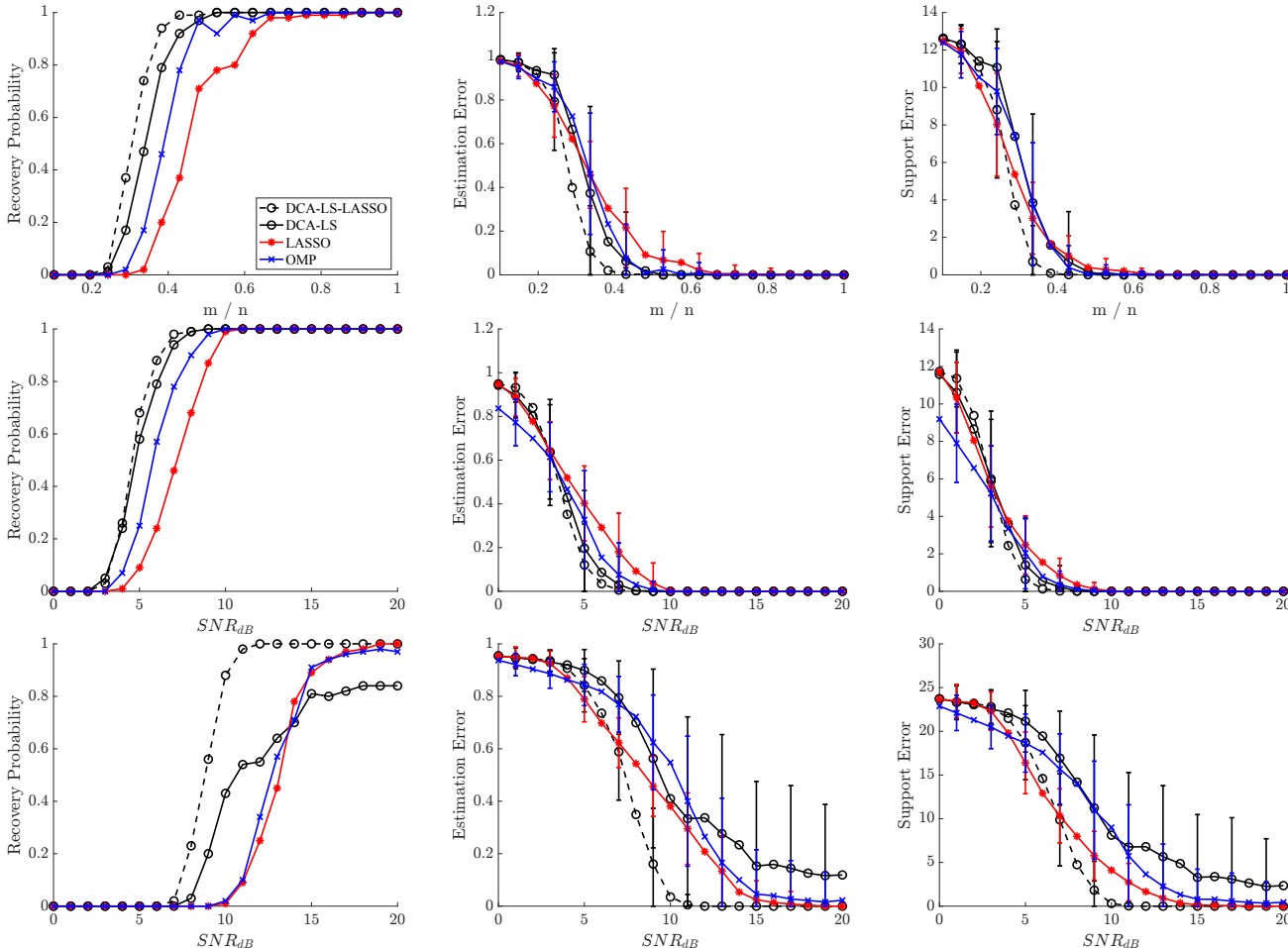

Figure 5: Performance results, averaged over 100 runs, for integer compressed sensing experiments with $n = 256$: Varying $m$ with $s = 13 = \lceil 0.05n \rceil$ and $\text{SNR}_{\text{dB}} = 8$ dB (top), varying $\text{SNR}_{\text{dB}}$ with $s = 13 = \lceil 0.05n \rceil$ and $m/n = 0.5$ (middle), and varying $\text{SNR}_{\text{dB}}$ with $s = 26 = \lceil 0.1n \rceil$ and $m/n = 0.5$ (bottom).

values. In terms of total runtime (Figure 6), DCA-LS-LASSO is faster than DCA-LS in some regimes: at high $\text{SNR}_{\text{dB}}$ with $m/n = 0.5$ (for $\text{SNR}_{\text{dB}} \geq 6$ with $s = 13$, and $\text{SNR}_{\text{dB}} \geq 9$ with $s = 26$), and for $m/n \in (0.35, 0.75)$ with $\text{SNR}_{\text{dB}} = 8$, but slower in others. In terms of individual runtime per $\lambda$ (Figure 7), DCA-LS-LASSO is faster than DCA-LS for $\lambda \geq 0.01$.

We also note that DCA-LS-LASSO's total runtime increases significantly when $m/n \gtrsim 0.75$. This is likely because for small $\lambda$ values, the influence of the $\ell_1$-norm regularizer in LASSO is minimal, so DCA-LS-LASSO is effectively initialized with a least-squares solution, which can be worse than the $x^0 = 0$ initialization used in DCA-LS. Inspecting the $\lambda$ values that yielded the best performance for both DCA-LS-LASSO and DCA-LS, we found that they were always $\lambda \geq 0.01$. So restricting $\lambda$ to this range would not impact performance. In Figure 8, we plot the sum of running times for only $\lambda = 1, 0.1, 0.01$ with $\text{SNR}_{\text{dB}} = 8$. With this restriction, DCA-LS-LASSO's total runtime no longer increases at $m/n \gtrsim 0.75$, and is actually lower than DCA-LS for all $m/n$.

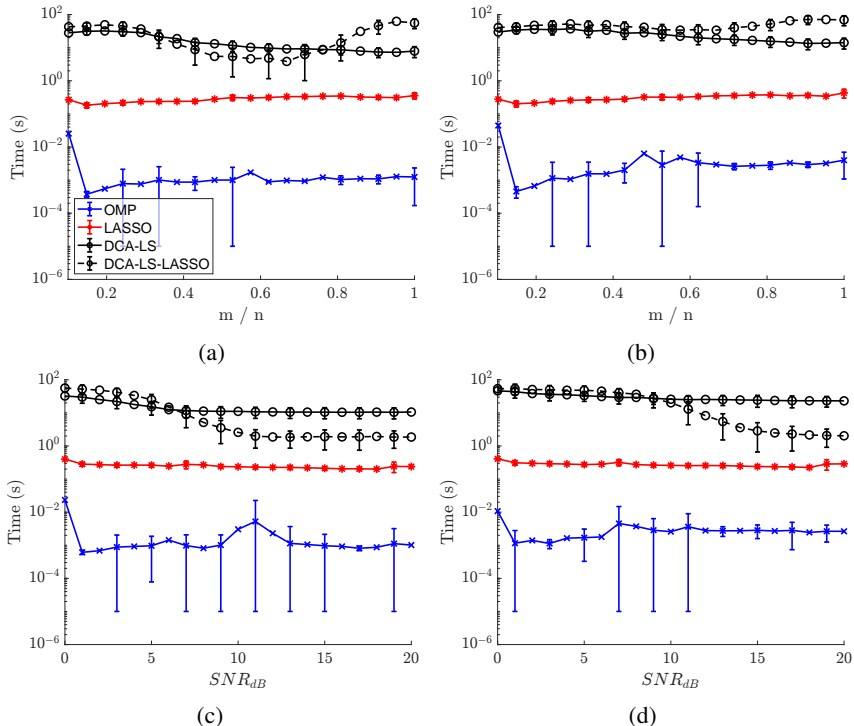

Figure 6: Running times (log-scale), summed over all $\lambda$ values and averaged over 100 runs, for integer compressed sensing experiments with $n = 256$: (a) Varying $m$ with $s = 13$ and $\text{SNR}_{\text{dB}} = 8$, (b) varying $m$ with $s = 26$ and $\text{SNR}_{\text{dB}} = 8$, (c) varying $\text{SNR}_{\text{dB}}$ with $s = 13$ and $m/n = 0.5$, and (d) varying $\text{SNR}_{\text{dB}}$ with $s = 26$ and $m/n = 0.5$.

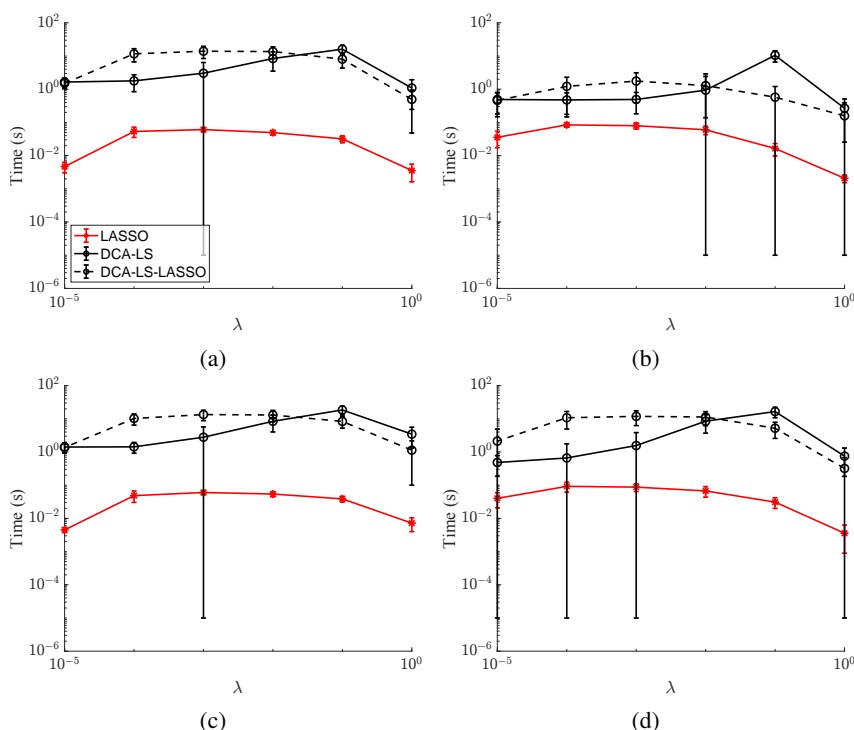

Figure 7: Running times (log-scale) for each $\lambda$, averaged over 100 runs, for integer compressed sensing experiments with $n = 256$ and $\text{SNR}_{\text{dB}} = 8$: (a) $m/n \approx 0.2$ and $s = 13$, (b) $m/n \approx 0.5$ and $s = 13$, (c) $m/n \approx 0.2$ and $s = 26$, and (d) $m/n \approx 0.5$ and $s = 26$.

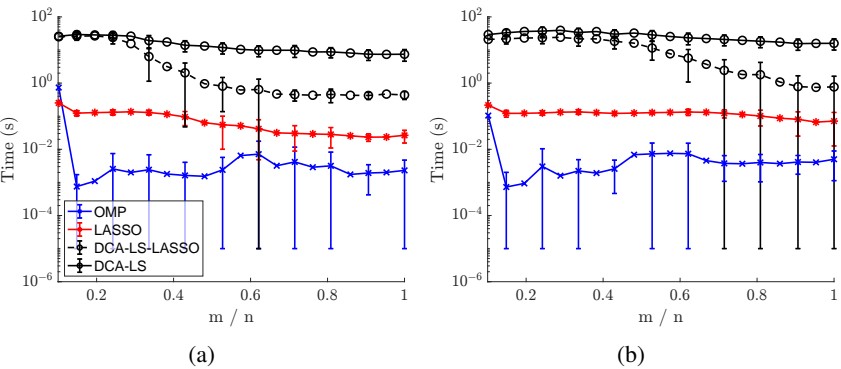

Figure 8: Running times (log-scale), summed over $\lambda = 1, 0.1, 0.01$ and averaged over 100 runs, for integer compressed sensing experiments with $n = 256$: (a) Varying $m$ with $s = 13$ and $\text{SNR}_{\text{dB}} = 8$, and (b) varying $m$ with $s = 26$ and $\text{SNR}_{\text{dB}} = 8$.

# I. DCA Variants Comparison

As discussed in Section 4.2, El Halabi et al. (2023) proposed two variants of standard DCA (9) for the special case of Problem (1), where $F$ is a set function ($\mathcal{X} = \{0, 1\}^n$). Both monotonically decrease the objective $F$ (up to $\epsilon_x$) and converge to a local minimum. In this section, we extend the more efficient of the two variants (Algorithm 2 therein) to general discrete domains and compare it with our proposed DCA variant (DCA-LS; Algorithm 1). We refer to this extension as DCA-Restart, and present it in Algorithm 2.

---

**Algorithm 2** DCA with restart

---

1: $\epsilon \geq 0, \epsilon' \geq \epsilon, T \in \mathbb{N}, X^0 \in [0, 1]_{\downarrow}^{n \times (k-1)}$
2: **for** $t = 1, \ldots, T$ **do**
3:     Choose $Y^t \in \partial h(X^t)$ (preferably corresponding to a row-stable permutation)
4:     $X^{t+1} \in \mathrm{argmin}_{X \in [0,1]_{\downarrow}^{n \times (k-1)}} g_{\downarrow}(X) - \langle Y^t, X \rangle_F$
5:     **if** $f(X^t) - f(X^{t+1}) \leq \epsilon$ **then**
6:       **if** $X^t \in \{0, 1\}_{\downarrow}^{n \times (k-1)}$ **then**
7:         $x^t = \Theta(X^t)$
8:       **else**
9:         $x^t = \mathrm{Round}_F(X^t)$
10:       **end if**
11:       $\bar{x}^t \in \mathrm{argmin}_{x \in N_{\mathcal{X}}(x^t)} F(x)$
12:       **if** $F(x^t) \leq F(\bar{x}^t) + \epsilon'$ **then**
13:         Stop.
14:       **else**
15:         $X^{t+1} = \Theta^{-1}(\bar{x}^t)$
16:       **end if**
17:     **end if**
18: **end for**

---

**Theoretical comparison** Similar to the set function variant, DCA-Restart runs standard DCA (9) and, at convergence, checks if rounding the current iterate yields an approximate local minimum of $F$. If not, it restarts from the best neighboring point. We show in Theorem I.1 that DCA-Restart satisfies the same theoretical guarantees as DCA-LS (see Theorem 4.5). In particular, it also recovers the guarantees of El Halabi et al. (2023) in the special case of set functions.

**Theorem I.1.** *Let $\{X^t\}, \{x^t\}$ be generated by Algorithm 2, where the subproblem on line 4 is solved up to accuracy $\epsilon_x \geq 0$. For all $t \in [T], \epsilon \geq 0, \epsilon' \geq \epsilon$, we have:*

*a) $f(X^{t+1}) \leq f(X^t) + \epsilon_x$.*

*b) Let $(p, q)$ be the permutation used to compute $Y^t$ in Proposition 2.3-e, and $\{y^i\}_{i=0}^{(k-1)n}$ the vectors corresponding to $(p, q)$, defined as in Definition 2.2. If $(p, q)$ is row-stable and $f(X^t) - f(X^{t+1}) \leq \epsilon$, then*

$$F(x^t) \leq F(y^i) + \epsilon + \epsilon_x \text{ for all } i \in [0 : (k - 1)n].$$

*c) Algorithm 2 converges to an $\epsilon'$-local minimum of $F$ after at most $(f(X^0) - f^*)/\epsilon$ iterations.*

*Proof.* Note that between each restart (line 15), Algorithm 2 is simply running standard DCA (9), so Proposition 2.5 applies.

a) This holds between each restart by Proposition 2.5-a. Whenever the algorithm restarts, we have $F(\bar{x}^t) < F(x^t) - \epsilon'$, i.e., $x^t$ is not an $\epsilon'$-local minimum. In this case, we have

$$\begin{aligned} f(X^{t+1}) &= F(\bar{x}^t) &&\text{(by Proposition 2.3-a)} \\ &< F(x^t) - \epsilon' \\ &\leq f(X^t) - \epsilon' &&\text{(by Proposition 2.3-a,c)} \end{aligned} \tag{48}$$

b) If $f(X^t) - f(X^{t+1}) \leq \epsilon$, then by Proposition 2.5-b, we have $Y^t \in \partial_{\epsilon+\epsilon_x} g(X^t) \cap \partial h(X^t)$. If $(p, q)$ is row-stable, then by definition of $y^i$, $(p, q)$ is a common non-increasing permutation for $X^t$ and $\Theta^{-1}(y^i)$, for all $i \in [0 : (k-1)n]$. Hence, $Y^t \in \partial h(\Theta^{-1}(y^i))$ by Proposition 2.3-e, and thus $Y^t \in \partial_{\epsilon+\epsilon_x} g(X^t) \cap \partial h(\Theta^{-1}(y^i))$. Therefore,

$$
\begin{aligned}
F(x^t) &\leq f(X^t) && \text{(by Proposition 2.3-a,c)} \\
&\leq f(\Theta^{-1}(y^i)) + \epsilon + \epsilon_x && \text{(by Proposition 2.4)} \\
&= F(y^i) + \epsilon + \epsilon_x, && \text{(by Proposition 2.3-a)}
\end{aligned}
$$

for any $i \in [(k-1)n]$.

c) For any iteration $t \in [T]$, if the algorithm did not terminate, then either $f(X^t) - f(X^{t+1}) > \epsilon$ or $F(\bar{x}^t) < F(x^t) - \epsilon'$. In the second case, the algorithm restarts, then by Equation (48) we have $f(X^t) - f(X^{t+1}) \geq \epsilon$, since $\epsilon' \geq \epsilon$. We therefore have $F(x^0) - F(x^t) \geq t\epsilon$, hence $t < (f(X^0) - f^*)/\epsilon$. If the algorithm did terminate, then $x^t$ must be an $\epsilon'$-local minimum of $F$.

$\square$

DCA-Restart has a similar computational cost per iteration as DCA-LS. The only difference is that DCA-LS has an additional cost per iteration of $O(n\text{EO}_F)$ for the local search step (line 3) and $O(nk)$ to map $\tilde{X}^{t+1}$ to $x^{t+1}$ (line 8). In DCA-Restart, these operations are only done at convergence $(f(X^t) - f(X^{t+1}) \leq \epsilon)$, which in the worst case can occur at every iteration. However, these differences have little impact on the overall per-iteration cost, which is dominated by the cost of solving the subproblem. Indeed, the total cost per iteration in DCA-Restart is $\tilde{O}(n(kL_{f^t_\downarrow}/\epsilon)^2 \text{EO}_{F^t} + nk\,\text{EO}_H)$; the same as in DCA-LS. Moreover, the number of iterations for both variants is at most $(f(X^0) - f^*)/\epsilon$. So, theoretically, both variants have very similar theoretical guarantees and runtime.

**Empirical comparison**  We empirically compare DCA-LS to DCA-Restart on all experiments included in the paper. We use the same parameters $T$ and $\epsilon$, and the same subproblem solver (pairwise-FW) in DCA-Restart as in DCA-LS; see Appendix G for how these are set in each experiment. We choose $\epsilon' = 0$, i.e., $x^t$ should be an exact local minimum if DCA-Restart stops before reaching the maximum number of iterations. We also choose a row-stable permutation $(p, q)$ when computing $Y^t$.

We report their performance on integer least squares (ILS) in Figure 9 and corresponding running times in Figure 10. Similarly, Figure 13 and Figure 14 show their performance and running times on integer compressed sensing (ICS). We also plot the number of DCA outer and inner iterations for ILS in Figures 11 and 12 and for ICS in Figures 16 and 17. The reported numbers of DCA inner iterations are the total number of inner iterations, i.e., iterations of pairwise-FW, summed over all outer iterations $t$. For ICS, the reported numbers of iterations and running times correspond to the sum over the 6 values of $\lambda$ tried, $\lambda = 1, 0.1, \ldots, 10^{-5}$. We also report the running time for each $\lambda$ separately at $m/n \approx 0.2$ and $0.5$ ($m = 50$ and $m = 123$, respectively) with $\text{SNR}_{\text{dB}} = 8$ in Figure 15. All results are again averaged over 100 runs, with error bars for standard deviations.

We observe that DCA-LS matches or outperforms DCA-Restart on all experiments. The two variants perform similarly when initialized with a good solution (LASSO in ICS, RAR in ILS); otherwise, DCA-LS performs better, sometimes by a large margin (see 4th row in Figure 13). In terms of runtime, DCA-Restart is generally faster on both ILS and ICS. For both applications, DCA-Restart takes slightly more outer iterations to converge, but has a lower per-iteration runtime, as evidenced by its smaller number of inner iterations. Intuitively, we expected each step of DCA-LS to decrease the objective more than in DCA-Restart, because of its more careful permutation choice, and thus to converge faster. In practice, this choice seems to lead to better solutions in some cases, but not significantly faster convergence, and comes at the cost of increased subproblem complexity. The slower convergence of pairwise-FW in DCA-LS may be due to the subgradients $Y^t$ changing more between consecutive iterations, causing the subproblem's solution to vary more, and thus take longer to solve, given that we warm-start pairwise-FW with the previous iteration's solution. Overall, the choice between the two variants is problem dependent; for example, on whether a good initialization is available.

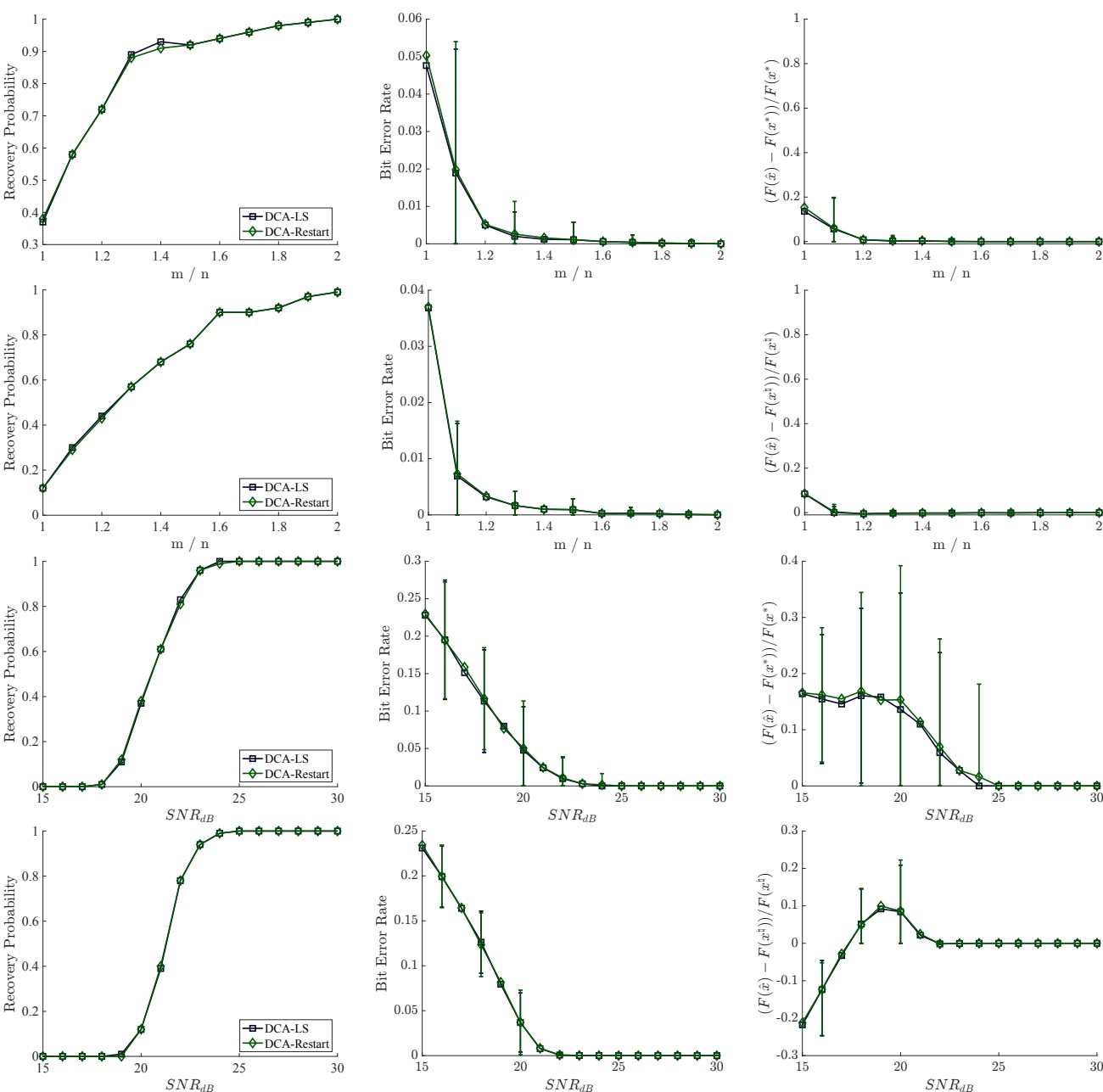

Figure 9: Performance results, averaged over 100 runs, for integer least squares experiments comparing two DCA variants: Varying $m$ with $n = 100$ and $SNR_{dB} = 8$ (first row), varying $m$ with $n = 400$ and $SNR_{dB} = 8$ (second row), varying $SNR_{dB}$ with $m = n = 100$ (third row), and varying $SNR_{dB}$ with $m = n = 400$ (fourth row).

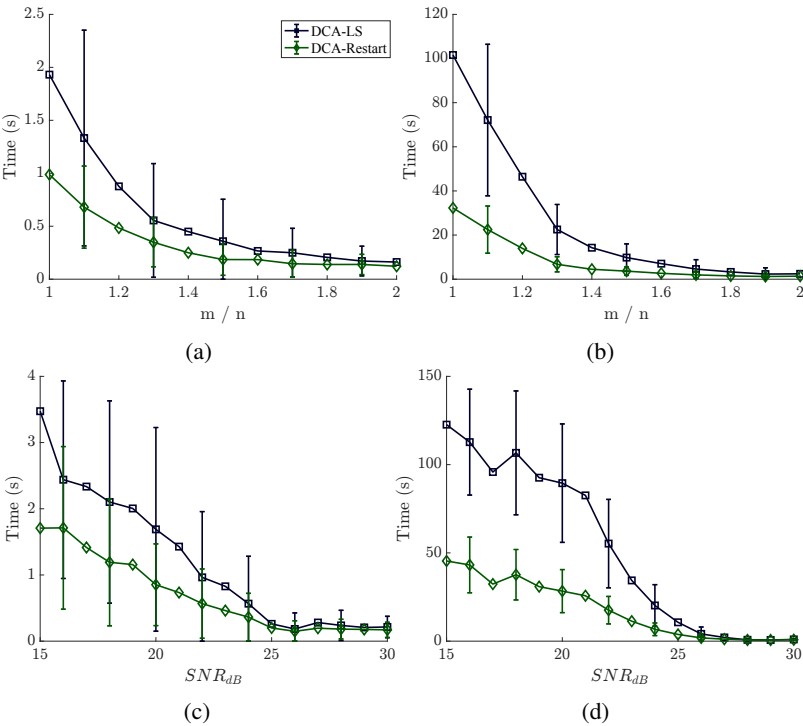

Figure 10: Running times, averaged over 100 runs, for integer least squares experiments comparing two DCA variants: (a) Varying $m$ with $n = 100$ and $SNR_{dB} = 8$, (b) varying $m$ with $n = 400$ and $SNR_{dB} = 8$, (c) varying $SNR_{dB}$ with $m = n = 100$, and (d) varying $SNR_{dB}$ with $m = n = 400$.

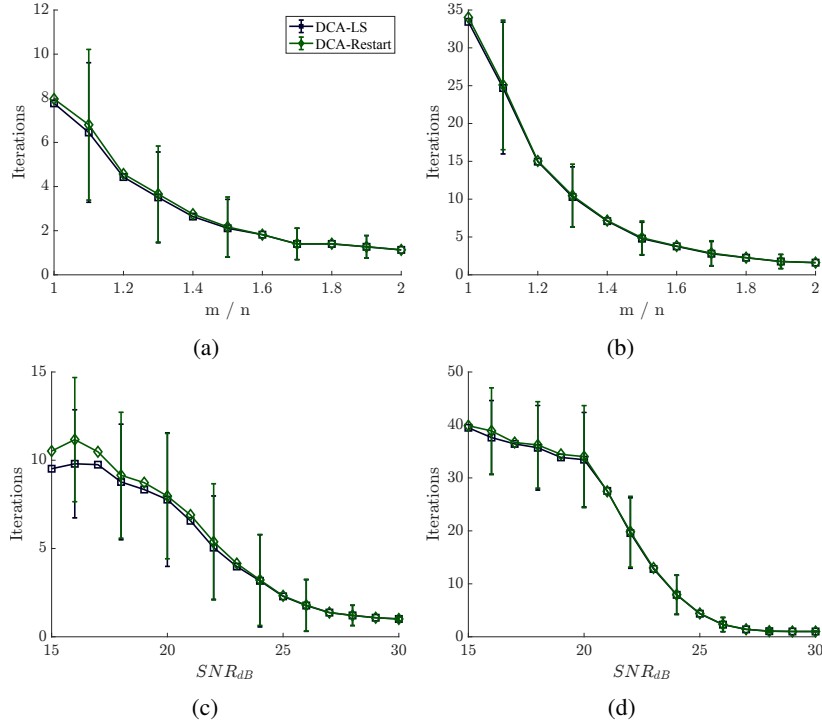

Figure 11: Number of DCA iterations, averaged over 100 runs, for integer least squares experiments: (a) Varying $m$ with $n = 100$ and $SNR_{dB} = 8$, (b) varying $m$ with $n = 400$ and $SNR_{dB} = 8$, (c) varying $SNR_{dB}$ with $m = n = 100$, and (d) varying $SNR_{dB}$ with $m = n = 400$.

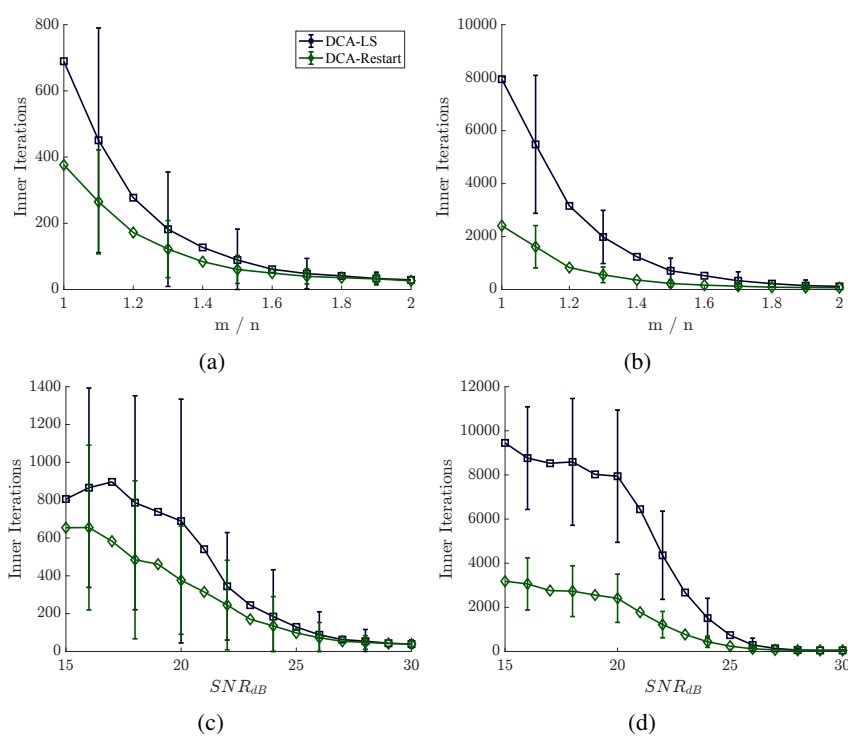

Figure 12: Number of DCA inner iterations, summed over all outer iterations $t$, averaged over 100 runs, for integer least squares experiments: (a) Varying $m$ with $n = 100$ and $\text{SNR}_{\text{dB}} = 8$, (b) varying $m$ with $n = 400$ and $\text{SNR}_{\text{dB}} = 8$, (c) varying $\text{SNR}_{\text{dB}}$ with $m = n = 100$, and (d) varying $\text{SNR}_{\text{dB}}$ with $m = n = 400$.

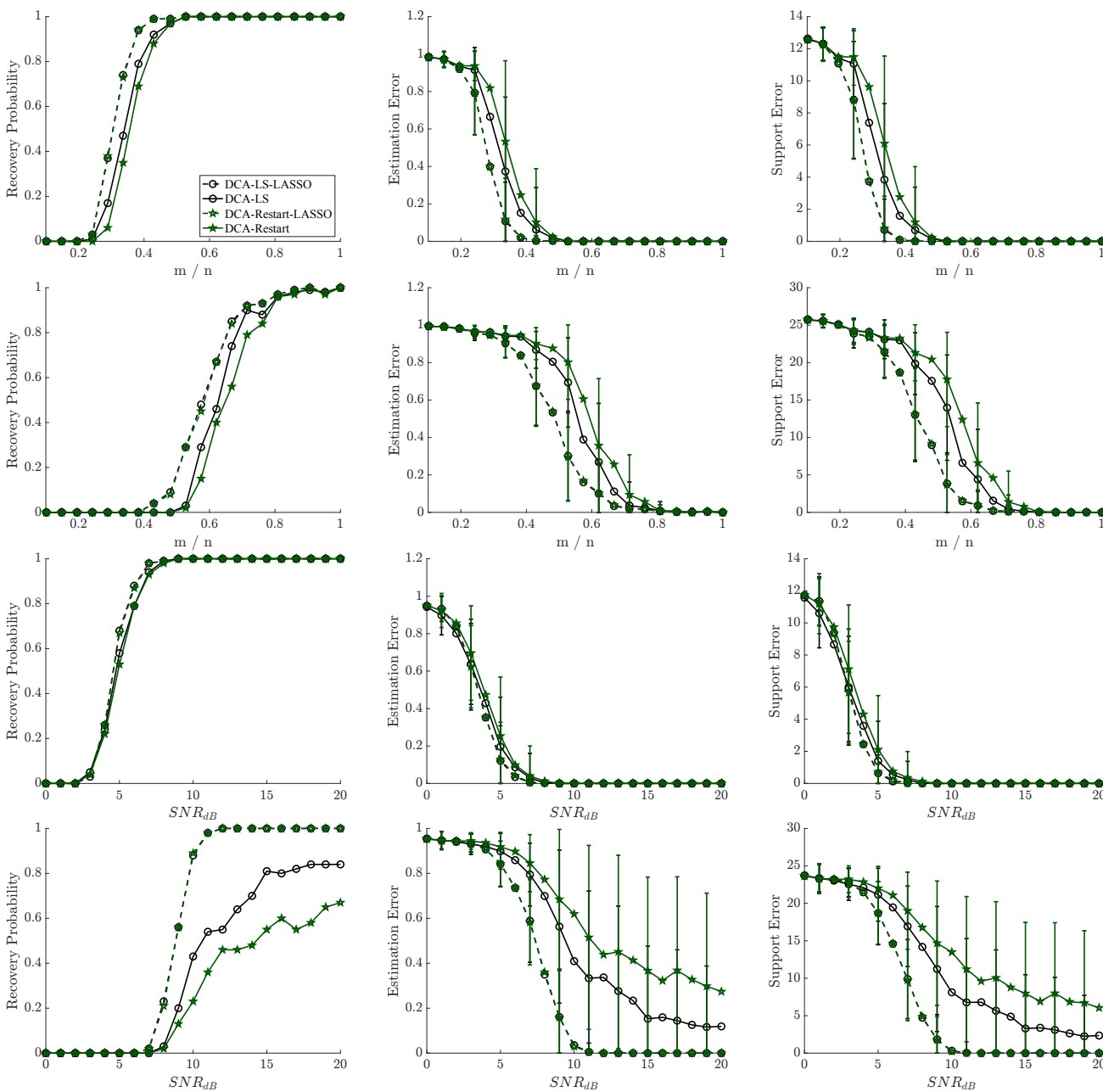

Figure 13: Performance results, averaged over 100 runs, for integer compressed sensing experiments comparing two DCA variants with $n = 256$: Varying $m$ with $s = 13$ and $SNR_{dB} = 8$ (first row), varying $m$ with $s = 26$ and $SNR_{dB} = 8$ (second row), varying $SNR_{dB}$ with $m/n = 0.5$ and $s = 13$ (third row), and varying $SNR_{dB}$ with $m/n = 0.5$ and $s = 26$ (fourth row).

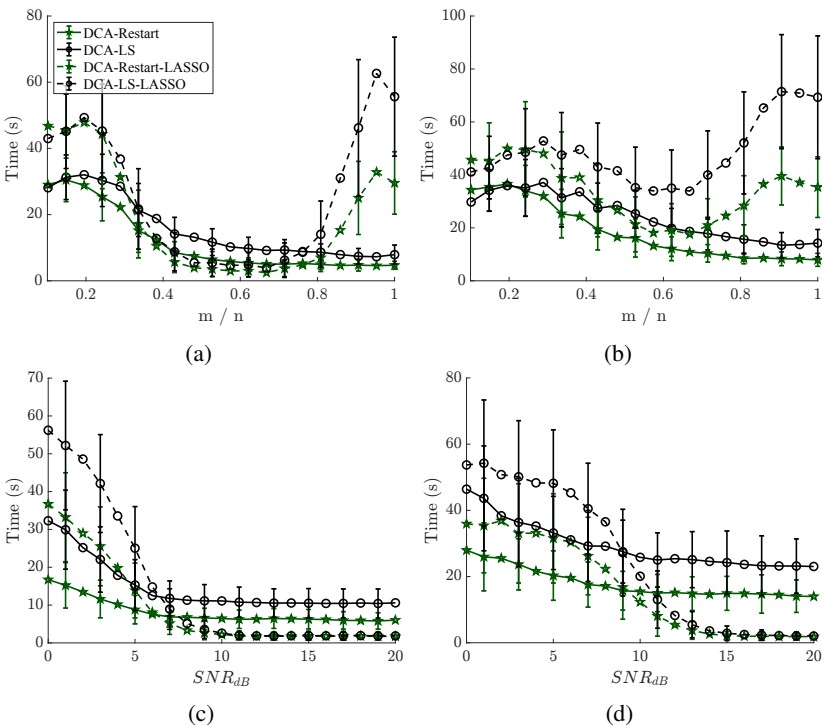

Figure 14: Running times, summed over all $\lambda$ values and averaged over 100 runs, for integer compressed sensing experiments comparing two DCA variants with $n = 256$: (a) Varying $m$ with $s = 13$ and $\text{SNR}_{\text{dB}} = 8$, (b) varying $m$ with $s = 26$ and $\text{SNR}_{\text{dB}} = 8$, (c) varying $\text{SNR}_{\text{dB}}$ with $s = 13$ and $m/n = 0.5$, and (d) varying $\text{SNR}_{\text{dB}}$ with $s = 26$ and $m/n = 0.5$.

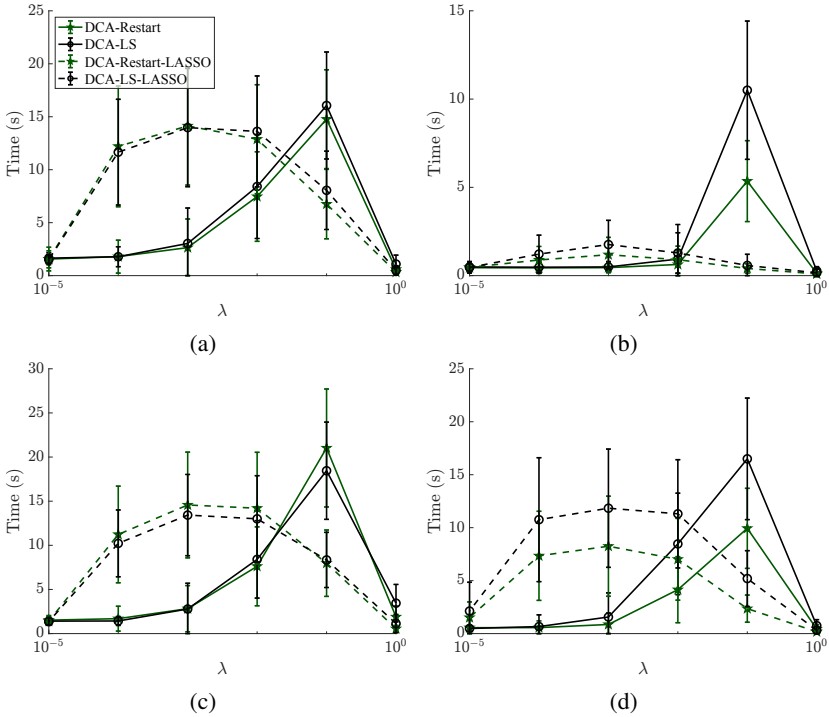

Figure 15: Running times for each $\lambda$, averaged over 100 runs, for integer compressed sensing experiments comparing two DCA variants with $n = 256$ and $\text{SNR}_{\text{dB}} = 8$: (a) $m/n \approx 0.2$ and $s = 13$, (b) $m/n \approx 0.5$ and $s = 13$, (c) $m/n \approx 0.2$ and $s = 26$, and (d) $m/n \approx 0.5$ and $s = 26$.

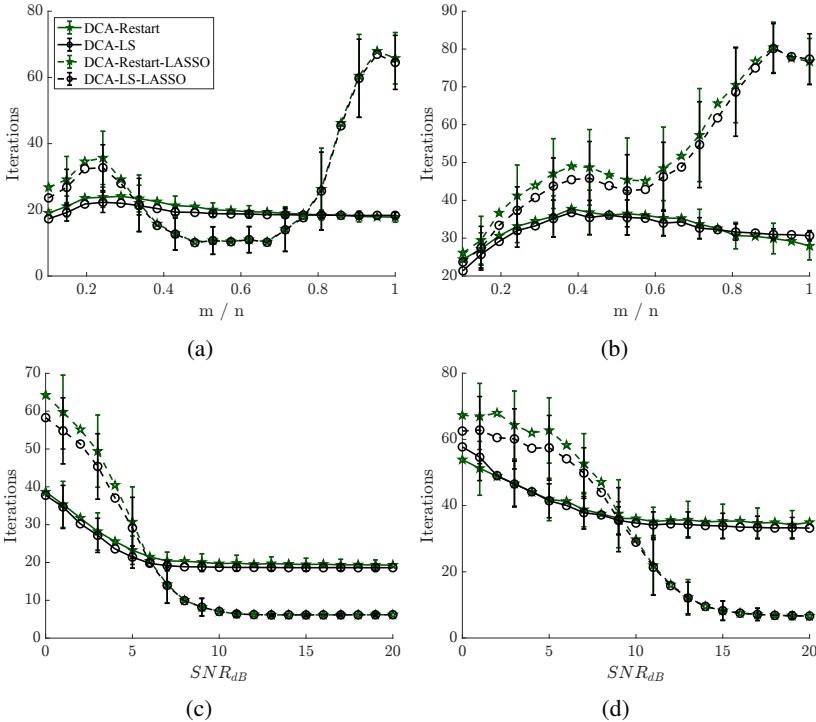

Figure 16: Number of DCA iterations, summed over all $\lambda$ values and averaged over 100 runs, for integer compressed sensing experiments with $n = 256$: (a) Varying $m$ with $s = 13$ and $\text{SNR}_{\text{dB}} = 8$, and (b) varying $m$ with $s = 26$ and $\text{SNR}_{\text{dB}} = 8$, (c) varying $\text{SNR}_{\text{dB}}$ with $s = 13$ and $m/n = 0.5$, and (d) varying $\text{SNR}_{\text{dB}}$ with $s = 26$ and $m/n = 0.5$

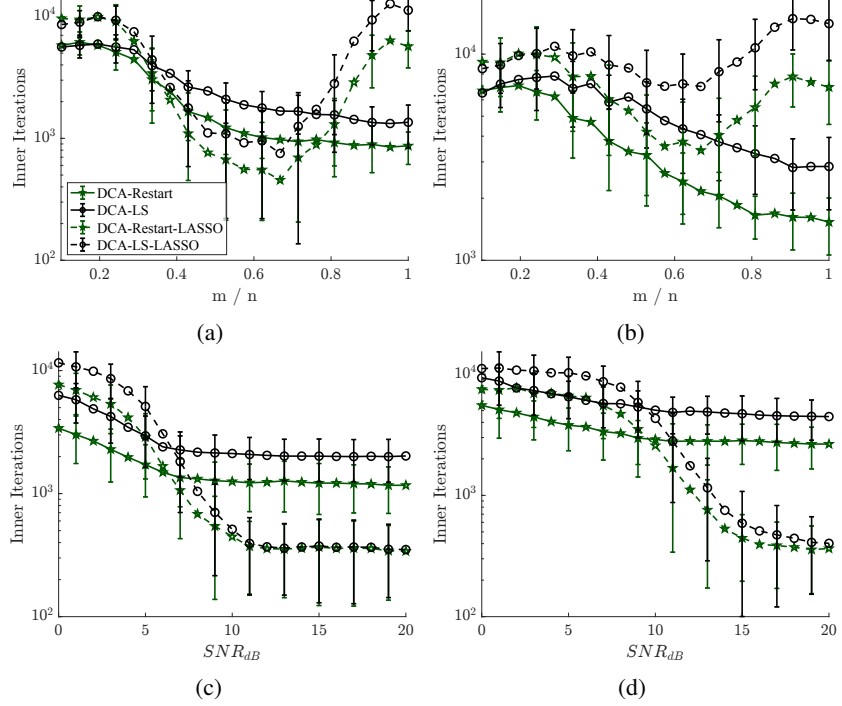

Figure 17: Number of DCA inner iterations (log-scale), summed over all outer iterations $t$ and all $\lambda$ values, averaged over 100 runs, for integer compressed sensing experiments with $n = 256$: (a) Varying $m$ with $s = 13$ and $\text{SNR}_{\text{dB}} = 8$, and (b) varying $m$ with $s = 26$ and $\text{SNR}_{\text{dB}} = 8$, (c) varying $\text{SNR}_{\text{dB}}$ with $s = 13$ and $m/n = 0.5$, and (d) varying $\text{SNR}_{\text{dB}}$ with $s = 26$ and $m/n = 0.5$.

