# OpenReview forum: "Discrete and Continuous Difference of Submodular Minimization"
_ICML.cc/2025/Conference — ICML 2025 poster_

### Official Review · Reviewer_o1eG · 2025-03-07

**Overall Recommendation:** 3

**Summary:**

This paper explores the minimization of the difference of submodular (DS) functions over both discrete and continuous domains, extending prior work that was restricted to set functions. The authors introduce a new variant of the DC Algorithm (DCA) to minimize DS functions, providing theoretical guarantees comparable to previous work in the set function case. The study demonstrates that DS functions naturally arise in applications such as quadratic programming and sparse learning. The proposed method outperforms existing baselines in integer compressive sensing and integer least squares tasks. Experimental results show significant improvements in recovery probability and error rates.

**Claims And Evidence:**

The claims in the paper appear well-supported both theoretically and experimentally, with no evident problematic claims.

**Essential References Not Discussed:**

Please refer to the above.

**Experimental Designs Or Analyses:**

The authors should include more results regarding the running time of their algorithm to better demonstrate its efficiency.

**Methods And Evaluation Criteria:**

I think the proposed methods and evaluation criteria make sense for the problem and application at hand. The problem of minimizing the difference of submodular functions (DS functions) in both discrete and continuous domains is a well-defined and important one in optimization, with practical applications in areas like quadratic programming, sparse learning, and compressive sensing.

**Other Comments Or Suggestions:**

N.A.

**Other Strengths And Weaknesses:**

N.A.

**Questions For Authors:**

Please refer to the above.

**Relation To Broader Scientific Literature:**

The problem addressed in this paper may be related to regularized submodular optimization, which involves optimizing a submodular function that is regularized by subtracting a linear or convex function [1][2].

[1] Kazemi, Ehsan, et al. "Regularized submodular maximization at scale." International Conference on Machine Learning. PMLR, 2021.

[2] Cui, Shuang, et al. "Constrained subset selection from data streams for profit maximization." Proceedings of the ACM Web Conference 2023. 2023.

**Theoretical Claims:**

I have conducted a preliminary review of the theoretical proofs in the paper, and they appear to be correct. However, I should note that my examination was not exhaustive, and there remains a possibility that some details may have been overlooked.

---

> ### Author Rebuttal · Authors · 2025-04-01
>
> Thank you for your positive review and helpful feedback. We address below your comments and questions.
>
> ---
>
> **1- Results on the proposed algorithm's running time**
>
> We report the average running time of the compared methods on the integer least squares experiment (Section 5.1) in  [Figure 4](https://drive.google.com/file/d/1qG4rhu3aBCSAeWfOMx_QJna69vjvDxxM/view). The reported times for DCA and ADMM do not include the time for the RAR initialization, and for OBQ do not include the time for the relaxed solution initialization (which has similar time as RAR).
> We observe that our algorithm is significantly faster that the optimal Gurobi solver, even for the small problem instance ($n=100$). For the larger instance ($n=400$), Gurobi is already too slow to be practical. While our algorithm is slower than heuristic baselines, it remains efficient, with a maximum runtime of $100$ seconds for $m=n=400$. We will include these plots and this discussion in the revision.
>
> The primary focus of our experiments was to demonstrate that our algorithm outperforms state-of-the-art baselines on challenging problems, rather than optimizing its computational efficiency. For example, the most computationally intensive part of our algorithm is computing subgradients of the continuous extensions, which requires evaluating successive marginals $F^t(y^{i-1} + e_{p_i}) - F^t(y^{i-1})$. In our current implementation, we compute these marginals by doing separate calls to $F^t$. This can be significantly sped up by reusing computation from evalutating $F^t(y^{i-1})$ when computing $F^t(y^{i-1} + e_{p_i}).
>
> ---
>
> **2- Relation to regularized submodular optimization [1][2]**
>
> The problem studied in [1][2] is that of maximizing the difference between a submodular set function and a modular set function over constraints. Our problem setup does not consider constraints. But the unconstrained version of that problem is indeed a special case of DS **set** function minimization. We will cite works on unconstrained regularized submodular set function optimization in the revision.

---

### Official Review · Reviewer_eVr5 · 2025-03-14

**Overall Recommendation:** 2

**Summary:**

Submodular functions are commonly studied as set functions, which can be viewed as functions defined on the vertices of the hypercube $ \\{ 0,1 \\}^n$.   This paper, however, similar to some prior literature, considers an extension of submodularity, where functions are defined over cartesian products of compact subsets of $\mathbb{R}$. They study the problem of minimizing the difference of these generalized submodular functions. For discrete domains, they propose an algorithm to solve DS via a reduction to integer lattice domains followed by their variant of DCA (difference of convex functions minimization algorithm), which converges to a local minimum at a rate of $O(1/k)$. They claim that with discretization, the same method applies to continuous domains as well.

**Claims And Evidence:**

Claims like "The results can be easily extended to unequal $k_i$s" need some explanation.

**Essential References Not Discussed:**

.

**Experimental Designs Or Analyses:**

The experiments compare their proposed algorithm for integer least squares problem (a special case of quadratic programs) and integer compressive sensing problem (a special case of Sparse Learning) with appropriate baselines for the respective problem.

**Methods And Evaluation Criteria:**

Overall, Methods and Evaluation criteria seems fine.

This paper builds upon "Difference of Submodular Minimization via DC Programming" by El Halabi et al., so it should include a clear comparison between its DCA variant and those of Halabi et al. in this setting.

**Other Comments Or Suggestions:**

.

**Other Strengths And Weaknesses:**

The paper is hard to follow.

This work appears to build on established ideas without introducing a significant departure from prior research.

**Questions For Authors:**

To my understanding, the paragraph Computational Complexity in Section 4 discusses the complexity of your DCA algorithm.
Could you please specify the theoretical guarantees of your DS algorithm as a whole?

**Relation To Broader Scientific Literature:**

This paper extends the result of the paper "Difference of Submodular Minimization via DC Programming" by El Halabi et al. (2023) to the setting proposed in the paper "Submodular functions: from discrete to continuous domains" by Bach.

**Theoretical Claims:**

The main body of the paper includes only proof sketches and high-level ideas. They seemed sound, but I was not able to verify their correctness.

---

> ### Author Rebuttal · Authors · 2025-04-01
>
> Thank you for your valuable feedback. We address below your comments and questions.
>
> ---
>
> **1- Explain the claim "The results can be easily extended to unequal $k_i$'s"**
>
> The extension to unequal k_i’s follows directly, though the notation becomes more cumbersome. The key modifications are:\
> 	- The relaxed domain (i.e., the domain of the DC program in Eq. (11)) changes to $\prod_{i=1}^n [0,1]\_{\downarrow}^{k_i -1}$.\
> 	- The map $\Theta$ is adjusted to map from $\prod_{i=1}^n \\{0,1\\}\_{\downarrow}^{k_i -1}$ to $\prod_{i=1}^n [0:k_i-1]$, with a similar definition.\
> 	- The continuous extension is now defined on $\prod_{i=1}^n \mathbb{R}^{k_i -1}$, and the summation in Eq. (6) runs from $1$ to $\sum_{i=1}^n (k_i - 1)$.
>
> ---
>
> **2- Comparison with the DCA variants of El Halabi et al. (2023)**
>
> Please refer to our response to **Reviewer 3rHC (3rd item)**, where we include a clear comparison between the DCA variants.
>
> ---
>
> **3- The paper is hard to follow.**
>
> We understand that the technical overhead may be challenging for readers, and we are happy to revise the paper to improve accessibility. Could you clarify which parts were hard to follow and suggest areas where clarity could be improved?
>
> ---
>
> **4- This work appears to build on established ideas without introducing a significant departure from prior research.**
>
> All reviewers seem to agree that the problem we address is interesting and well motivated, and that our results are valuable. We believe that achieving our results by extending existing results in a relatively simple way should not be seen as a weakness.
>
> That said, we would like to clarify that while our results build on existing work, they did require novel ideas and proofs. In particular, the main challenges are the following:
> 1. One factor that makes our results appear “straightforward” is the use of simpler and cleaner notation.
> For example, we represent the input of the continuous extension as a matrix $X\in [0,1]_{\downarrow}^{n \times (k-1)}$, instead of a list of vectors $X = (x_1, \cdots, x_n) \in [0: k-1]^n$ as in Bach and Axelrod. This change significantly simplified the presentation of the results.
> 2. While the proof that any function on a discrete domain is a DS function (Proposition 3.1) is a straighforward extension of the set function result in (Iyer & Bilmes, 2012), the analogous result for continuous domains (Proposition 3.3) required leveraging an alternative definition of submodularity and relating it to function smoothness.
> 3. Extending DCA (whether our variant or those in (El Halabi et al., 2023)) from set functions to general discrete functions is non-trivial. It required restricting the non-increasing permutation $(p,q)$ of $X^t$ to be row-stable, to ensure that it's a non-increasing permutation of $\{\Theta^{-1}(y^i)\}\_{i=1}^{(k-1)n}$ too. This is essential to guarantee local minimality (see proof of Theorem 4.5-b). This is not needed in the set function case ($k=2$), where any non-increasing permutation of $X \in \mathbb{R}_\downarrow^{n \times (k-1)}$ is row-stable.
> 4. Our proposed DCA variant (Algorithm 1) introduces a novel approach for selecting the subgradient using the local search step (lines 3-4), which ensures direct convergence to an approximate local minimum. This improves performance in some settings compared to the variant (extended to general discrete domains) proposed in (El Halabi et al., 2023). For further details, please see our response to **Reviewer 3rHC (3rd item)**.
> 5. Identifying important applications (Section 3.2) that have natural DS decompositions but are not DC, and do not have easy discrete DC decompositions, was also non-trivial. It required exploiting properties of DC functions (e.g., in Proposition A.1) and discrete DC functions (see Section A.2).
> 6. Showing that Lipschitz continuity is not necessary for bounding the discretization error for continuous domains, as in the case of the $\ell_q$-norm (Proposition G.2), is novel and its proof is non-trivial. We  believe that this result could be generalized to other similar functions.
>
> ---
>
> **5- Clarification on Theoretical Guarantees and Computational Complexity of DS Algorithm**
>
> Our full DS algorithm is presented in Algorithm 1. Its theoretical guarantees are given in Theorem 4.5, and its computational complexity is discussed in the corresponding paragraph in Section 4.2.
> The algorithm applies DCA to the DC Problem (11) using the DC decomposition given at the beginning of paragraph "Algorithm", while also maintaining iterates $x^t$ as the solution to the original DS Problem (1).
> The iterates $X^t \in [0,1]_{\downarrow}^{n \times (k-1)}$ are updated with DCA updates (see Eq. (9)), while iterates $x^t \in [0:k-1]^n$ are obtained by rounding (line 7). As explained, the rounding step can be skipped if the solution of the subproblem $\tilde{X}^t$ is integral.

---

### Official Review · Reviewer_yXuG · 2025-03-18

**Overall Recommendation:** 4

**Summary:**

This paper considers the minimization of a difference of submodular functions (DS) in both the continuous and discrete domains. Unlike the submodular minimization problem, which can be solved in polynomial time, this problem cannot even be approximated efficiently. This paper accomplishes two main things: (i) It shows how broad the class of DS functions are by proving that many existing functions can be represented this way, although it is computationally hard to find a representation, and (ii) It provides an algorithm and proves that the algorithm returns an approximate local minimum (even finding a true local minimum is computationally hard. This algorithms builds upon the fact that the continuous extension of DS functions via their Lovasz extension is a difference of convex functions for which the DCA algorithm exists. This paper proposes a modified variant of the DCA algorithm. Finally, they include an experimental section where they compare their algorithm to existing alternatives on two different applications.

**Claims And Evidence:**

Yes

**Essential References Not Discussed:**

I did not notice any missing essential references.

**Experimental Designs Or Analyses:**

Yes, the experiments looked reasonable to me.

**Methods And Evaluation Criteria:**

Yes

**Other Comments Or Suggestions:**

None

**Other Strengths And Weaknesses:**

Strengths
- I think their problem statement is interesting, and even though it's minimization I think it is related to submodular maximization applications as well. In particular, I wonder if summarization objectives commonly found in submodular maximization papers where there is a diversity penalty can also be viewed in this sort of problem statement.
- The problem is computationally challenging, but they were able to develop an algorithm that finds an approximate local optimum efficiently. This is an interesting type of algorithm, and maybe could be applied more broadly.
- Their result is connected with convex optimization, and they build upon the existing DCA algorithm.
- They provided an experimental section.

Weaknesses
- The algorithm may not be extremely novel, but this is fine in my opinion.
- Some of the writing, in particular in the introduction, was a bit hard to interpret. E.g. "developed algorithms for this special case, that monotonically decrease the objective value" I did not understand when reading the introduction.

**Questions For Authors:**

None

**Relation To Broader Scientific Literature:**

This paper is related to several existing related works that have studied special cases of this problem, e.g. Narasimhan and Bilmes [2005]. This work is of interest to the submodular optimization community.

**Theoretical Claims:**

I did not thoroughly check correctness, but I did not notice any issues.

---

> ### Author Rebuttal · Authors · 2025-04-01
>
> Thank you for your positive review. We will improve the writing of the introduction.

---

### Official Review · Reviewer_3rHC · 2025-03-19

**Overall Recommendation:** 2

**Summary:**

The minimization of a difference of submodular functions is studied, in a discrete and continuous setting. The discrete setting is more like a lattice that generalizes set optimization. A variant of the DC algorithm is developed that uses local search ideas. Experiments are performed to validate the algorithm.

**Claims And Evidence:**

Yes.

**Essential References Not Discussed:**

No.

**Experimental Designs Or Analyses:**

Yes. See strengths and weaknesses.

**Methods And Evaluation Criteria:**

Yes.

**Other Comments Or Suggestions:**

- Line 208: funcion -> function
- Line 167: us -> use
- The writing style is generally fine but it is a little informal. I kind of like that, because it means it wasn't written by an AI. But still, there were multiple examples of incomplete sentences (often starting with "Though"). Though is an informal shortening of although, and starts a subordinate clause that needs to be paired with a main clause. As in: "Although I like cats, I like dogs better."

**Other Strengths And Weaknesses:**

Strengths:
- Exposition is fairly easy to follow, even for someone unfamiliar with the area. Background and preliminary information goes until page 4. However, see weaknesses.
- The problem setup is very general, which makes it difficult. Some natural applications are given, which motivate the work well.
- The experimental results are convincing, although perhaps a natural baseline was left out. See weaknesses.

Weaknesses:
- Although much background information relevant to the problem studied is presented, I found it difficult to get a good idea of what is novel to this paper, and what is challenging about the variant that is studied. Such a good exposition of context is given that it becomes difficult to separate the contribution from the context.
I think the algorithm developed (a DCA variant) is interesting. But I don't know enough to really judge its distance from standard DCA. The authors did not explain clearly was is different.
- I didn't understand why the reduction discussed on line 092 for DR submodular functions, could not be extended to the case of DS functions on discrete domains. And then existing techniques from DS set optimization applied. A related issue is the reduction mentioned on line 317. The authors state that this reduction (line 317) is more expensive, but no justification is given in the main text. Also, I don't think the authors compared with this approach experimentally.

**Questions For Authors:**

- See strengths and weaknesses.
- Is the algorithm of El Halabi et al. (2023) compared, either theoretically or empirically? As this is also a variant of DCA, shouldn't it be a natural baseline?

**Relation To Broader Scientific Literature:**

Generalizes DS set function optimization to discrete domains of R^n. For continuous domains, there is a work of El Halabi et al. (2023). I'm unsure how the results in this paper are related to the former, although they do cite and discuss.

**Theoretical Claims:**

I did not check. All proofs are relegated to appendix or results from other works.

---

> ### Author Rebuttal · Authors · 2025-04-01
>
> Thank you for your valuable feedback. We address below your questions.
>
> ---
>
> **1- Distinguishing novel contributions from background**
>
> Our main contributions are outlined in the introduction (lines 61-74, 1st col). The DS minimization problem over general discrete and continuous domains $\mathcal{X}$ (Problem 1) has not been studied before. Prior work only considered special cases, where $F$ is a set function ($\mathcal{X} = \{0,1\}^n$), or is submodular ($H=0$). DS Minimization is significantly harder than submodular minimization, which is solvable in polynomial time, whereas even finding a local minimum or a polynomial approximation factor for DS minimization is provably hard.
>
> Standard DCA (Eq. 9) is not guaranteed to converge to an approximate local minimum, even for set functions (Section 4.2, lines 324- 333). Our DCA variant (Algorithm 1) differs by maintaining iterates for the original problem $x^t \in [0:k-1]^n$, obtained by rounding the iterates of the relaxed problem $X^t \in [0,1]_{\downarrow}^{n \times (k-1)}$, and by carefully choosing a subgradient $Y^t$ which ensures convergence to an approximate local minimum.
> For further discussion on the technical novelty of our results, see our response to **Reviewer eVr5 (4th item)**.
>
> ---
>
> **2-  Extension of the DR submodular reduction to DS functions & Cost of the submodular set function reduction and empirical comparison**
>
> The reduction by Ene & Nguyen (2016) (line 092) applies only to DR-submodular functions, a subclass of submodular functions that are concave along nonnegative directions. For a DS function $F = G - H$, the submodular components $G$ and $H$ are not necessarily DR-submodular.
> This reduction cannot be readily extended to general submodular functions. It relies on a carefully chosen map $M: \{0,1\}^t \to [0:k-1]^n$, where $t \leq 2 \log k + 1 $, to construct an equivalent submodular set function $\tilde{F}(X) = F(M(X))$ whose domain scales with $O(\log k)$. when $F$ is not DR-submodular, $M$ does not preserve submodularity.
>
> We discuss the general reduction for submodular functions (line 317) in Appendix B and compare its empirical performance. Our results show that the reduction approach is indeed slower than the direct approach used by Bach and us.
> See also our response to **Reviewer Ecvy (6th item)** for a discussion of the theoretical complexity of the two approaches.
> We will add a brief explanation in the main text.
>
> ---
>
> **3- Relation to (El Halabi et al., 2023) & Theoretical and empirical comparison with their DCA variants**
>
> El Halabi et al. (2023) study a special case of our DS minimization problem where $F$ is a **set** function ($\mathcal{X} = \{0,1\}^n$); they do not consider continuous domains.
>
> They proposed two DCA variants: One is infeasible, as it requires trying $O(n)$ subgradients per iteration, while the other (Algorithm 2 therein) restarts from the best neighboring point at convergence if the solution is not an approximate local minimum.
> Both can be extended to general discrete domains similarly to our DCA variant. The main non-trivial change is the restriction of the permutation $(p,q)$ to be row-stable.
> Our DCA variant (Algorithm 1) instead selects a single subgradient using a local search step (lines 3-4), ensuring direct convergence to an approximate local minimum without restarts.
>
> Our theoretical guarantees (Theorem 4.5) generalize those of (El Halabi et al. 2023) to general discrete domains; recovering the same guarantees in the set function case.
>
> We empirically compared our DCA variant (DCA-LS) to an extension of the more efficient DCA variant from El Halabi et al. (2023) (DCA-Restart) on all  experiments included in the paper.
> We report their performance on integer least squares (ILS) in [Figure 5](https://tinyurl.com/Rebuttal-Fig5) and  running times in [Figure 6](https://tinyurl.com/Rebuttal-Fig6). Similarly, [Figure 7](https://tinyurl.com/Rebuttal-Fig7) and [Figure 8](https://tinyurl.com/Rebuttal-Fig8) show their performance and running times on integer compressive sensing (ICS). We plot running times for all $\lambda$ values at $m/n =0.5$ and $0.2$.
> DCA-LS matches or outperforms DCA-Restart on all experiments. The two variants perform similarly when initialized with a good solution (LASSO in ICS, RAR in ILS), otherwise DCA-LS performs better, sometimes by a large margin.
> In terms of runtime, DCA-Restart is faster on ILS and for some $m/n$ values in ICS, e.g., $m/n=0.5$, but slower for others, e.g., $m/n=0.2$. Thus, the choice between the two variants is problem dependent.
>
> We will revise the paper to include these results and adjust the claim on line 343-344 (1st col), which originally stated that our DCA variant is more efficient than those in (El Halabi et al. 2023). This is only true for one of them. This claim was based on an earlier, less extensive comparison on an ICS experiment, where DCA-LS was both faster and performed better than DCA-Restart.

---

### Official Review · Reviewer_Ecvy · 2025-03-23

**Overall Recommendation:** 3

**Summary:**

The paper investigates the minimization of difference-of-submodular (DS) functions over both discrete (products of finite sets) and continuous (products of intervals) domains. The authors establish that every function on a discrete domain and every smooth function on a continuous domain admits a DS decomposition.

For DS minimization over discrete domains, they show that the problem can be reduced to the case of having an integer lattice as domain. In the continuous setting, they approximate the problem by discretization and then reducing it to the integer lattice case approximatively. The key result is that DS minimization on an integer lattice is equivalent to minimizing a continuous extension (generalizing the Lovász extension), enabling the use of a difference-of-convex (DC) algorithm. This algorithm monotonically decreases function values and converges to a local minimum.

The authors validate their approach with experiments on integer least squares and integer compressive sensing, demonstrating improved performance over state-of-the-art methods.

**Claims And Evidence:**

The main claims of the paper are:
- Any function on a discrete domain and any smooth function on a continuous domain can be expressed as a DS function.
- One can (approximately, in the continuous case) reduce the problem to an integer lattice domain.
- Minimizing a DS decomposition over an integer lattice domain is equivalent to minimizing a continuous extension.
- The algorithm computes an approximate local minimum.
- Applying a DC algorithm to integer least squares and integer compressive sensing improves performance over existing methods.

The first three claims are theoretical and well-supported. The paper also discusses the computational complexity and the theoretical guarantees of the algorithm’s output, which appear to be correct. The experiments further support the empirical claims.

Below are some minor issues:

- It is not entirely clear whether the theoretical guarantees extend to a DS function over a continuous domain after discretization. While the single results put together suggest this should be the case, the paper would benefit from a formal theorem stating: The algorithm finds a local optimum with guarantee G in time T for the class of DS functions C.

- The paper asserts that finding the best DS decomposition is generally infeasible, but it does not define what constitutes the "best" decomposition, nor does it provide strong support for this claim. The authors could strengthen this point by referencing literature.

- After Proposition 3.1, the paper states: "Obtaining tight lower bounds α and β in the above proof requires exponential time in general" and "Note that loose bounds on α and β would degrade performance." Both claims lack supporting argumentation. A brief explanation or reference would help clarify these statements.

- On line 241 (right), the sentence "We show in Appendix A.2 that both applications do not have a natural discrete DC decomposition." seems overly strong, as the authors only prove that a specific decomposition is not DC. A more precise formulation would improve accuracy.

**Essential References Not Discussed:**

As discussed in the Claims and Evidence section, the paper asserts that finding the best DS decomposition is generally infeasible, but it does not connect this claim to existing literature. The authors could reference works such as Decomposition Polyhedra of Piecewise Linear Functions (https://arxiv.org/abs/2410.04907), which explores the complexity of DS and DC decompositions.

**Experimental Designs Or Analyses:**

I read the experimental setup in the main paper and it appeared sound to me. But I did not validate any technical details and I am not an expert neither of the problems nor the methods.

**Methods And Evaluation Criteria:**

The chosen problems (integer least squares and integer compressive sensing) to evaluate the performance of the proposed algorithm and the selected metrics (recovery probability and estimation error) appear reasonable. The competing algorithms seem to represent the state of the art, but I am not deeply familiar with the field or these specific methods.

**Other Comments Or Suggestions:**

There are statements in the main paper where the proofs are deferred to the appendix, but no links are provided to direct the reader to the relevant proofs. As a result, readers must search through the appendix to find the proofs, even for key results such as Proposition 4.4 and 4.5. Including hyperlinks would make the paper more reader-friendly.

Further suggestions:

- It would be helpful to define what a subgradient is.

- For the definition of round, it could be clarified that the goal is to obtain the vector, not the index. This is only clear from the context, but the definition itself is somewhat ambiguous.

- Providing some intuition and perhaps including a small illustration of an epsilon-subdifferential would be beneficial.

 - A sentence or two following Proposition 2.5, explaining the implications of the statement on the theoretical guarantees of the algorithm, would make it easier to understand its significance.

**Other Strengths And Weaknesses:**

I appreciate the idea of relating general DS functions to a DC function, and it is interesting that this approach seems to work better in practice than reducing the problem to the set function case and minimizing the Lovász extensions via a DC algorithm, as illustrated in the appendix. This together with the experiments showing better performance than exisiting methods for DS functions makes the paper a valuable contribution.

From a theoretical perspective, the results don’t seem particularly profound, as they appear to be a straightforward extension of existing knowledge.

**Questions For Authors:**

- Can you think of any theoretic justification why the attempt of translating the DS problem directly to a DC problem works better than reducing first to the set function case?

**Relation To Broader Scientific Literature:**

The paper is well-grounded in the existing literature and acknowledges prior results and ideas upon which it builds.

**Theoretical Claims:**

The theoretical results appear sound and well-supported, and the proof sketches are reasonable. However, I did not verify all the proofs in detail.

---

> ### Author Rebuttal · Authors · 2025-04-01
>
> Thank you for your positive review and helpful feedback.  We address below your comments and questions.
>
> ---
> **1 - Extension of theoretical guarantees to continuous domains**
>
> Theorem 4.5 extends to continuous domains as follows:\
> Let $F'$ be defined as in Section 4.1, i.e., $F'(x) = F(x/(k-1))$ where $k = \lceil L/\epsilon' \rceil + 1$ for some $\epsilon'>0$. Let $\tilde{x}^t = x^t/(k-1)$, where $x^t$ are Algorithm 1's iterates for $F'$. Then:\
> a) $F(\tilde{x}^{t+1}) \leq F(\tilde{x}^t) + \epsilon_x$\
> b) At convergence: $F(\tilde{x}^t) \leq F(y^i/(k-1)) + \epsilon + \epsilon_x$ for all $i$. In particular, $F(\tilde{x}^t) \leq F(\tilde{x}^t \pm e_i/(k-1)) + \epsilon + \epsilon_x$.\
> c) Number of iterations is at most $(F(\tilde{x}^0) - F^*)/\epsilon$. \
> However, the second guarantee in (b) is meaningful only if $\epsilon' > \epsilon + \epsilon_x$, as Lipschitz continuity of $F$ already implies $F(\tilde{x}^t) \leq F(\tilde{x}^t \pm \tfrac{e_i}{k-1}) + \epsilon'$. We will clarify this in the paper.
>
> ---
> **2- Define "best" DS decomposition and clarify its infeasiblility**
>
> By best DS decomposition, we meant the one with the tightest $\alpha$ or $L_F$ in the proofs of Propositions 3.1 and 3.3. On lines 243-245 (1st col), we cite a reference showing that computing a tight $L_F$ is exponential hard. Computing the tightest $\alpha$ also has exponential complexity, even for set functions. Indeed, we can test if $F$ is submodular, which has exponential complexity (Seshadhri & Vondrák, 2014), by computing $\alpha = \min_{i, j \in V, S \subseteq V \setminus \{i, j\}} F(S \cup i) - F(S) - F(S \cup \{i, j\}) + F(S \cup j)$ and checking if $\alpha \geq 0$. We will clarify this in the revision.
>
> Like DC functions, DS functions have infinitely many DS decompositions. Finding the "best" one is an even more difficult question, as it's unclear how to define "best". Thank you for the suggested reference, we will cite it.
>
> Seshadhri, C., & Vondrák, J. (2014). “Is submodularity testable?” Algorithmica, 69(1), 1-25.
>
> ---
> **3- Impact of loose bounds on $\alpha$ and $\beta$**
>
> Looser bounds on $\alpha$ and $\beta$ lead to a DS decomposition where the continuous extensions of $G$ and $H$ have larger Lipschitz constants,slowing down optimization. As discussed in Section 4.2 ("Computational complexity"), the runtime of Algorithm 1, particularly for solving the submodular minimization at each iteration $t$, depends on the Lipschitz constant* $L_{f^t_\downarrow}$ of $f^t_\downarrow$, the continous extension  of $F^t = G - H^t = F + \frac{\alpha}{\beta} \tilde{H} - \tilde{H}^t$. Here, $\tilde{H}^t(x) = \frac{\beta}{\alpha} H^t(x)$ is a modular approximation of $\tilde{H}(x)$ satisfying $\tilde{H}(x) \geq \tilde{H}(x^t) + \tilde{H}^t(x) - \tilde{H}^t(x^t)$. We can lower bound $L_{f^t\_\downarrow}$ by the Lipschitz constant of $F^t$: $L\_{f^t_\downarrow} \geq \max\_{x, x'} \frac{|F^t(x) - F^t(x')|}{\\|x - x'\\|\_1}$. For example, for a non-decreasing function $F$ and $x  \leq x'$,
> taking $x' = x^t$ gives $|F^t(x) - F^t(x')| = |F(x) - F(x^t)| + \frac{|\alpha|}{\beta} |(\tilde{H}^t(x) - \tilde{H}^t(x^t)) - (\tilde{H}(x) - \tilde{H}(x^t))|.$ Thus, a larger $\frac{|\alpha|}{\beta}$ yields a larger $L_{f^t_\downarrow}$. We will clarify this in our revision.
>
> *Typo on line 373: $L_{F^t_\downarrow} \to L_{f^t_\downarrow}$
>
> ---
> **4- Revise the sentence on line 241 (right) to be more accurate.**
>
> We will revise the sentence to: "We show in Appendix A.2 that the natural DS decomposition in both applications is not a discrete DC decomposition as defined in (Maehara & Murota, 2015) and cannot be easily adapted into one for general discrete domains, even when ignoring the integer-valued restriction."
>
> ---
> **5- Theoretical results seem straightforward extensions of existing work**
>
> See our response to **Reviewer eVr5 (4th item)**, where we clarify the novel aspects of our theoretical results.
>
> ---
> **6- Why is the direct approach better than set function reduction?**
>
> As discussed in (Bach, 2019, section 4.4), reducing a submodular function $F$ to a submodular set function $\tilde{F}$ leads to slower optimization due to the larger Lipschitz constant $L_{\tilde{f}\_\downarrow}$ of the continuous extension $\tilde{f}\_\downarrow$  of $\tilde{F}$. Specifically, while we can bound $L_{f\_\downarrow} \leq \sqrt{n k} \max\_i B_i$, the bound for $\tilde{f}\_\downarrow$ is $k$ times larger, i.e., $L_{\tilde{f}\_\downarrow} \leq k \sqrt{n k} \max\_i B_i$, where $B_i$ is defined in Eq. (26) in Appendix B.
> This affects submodular minimization algorithms in both (Bach, 2019) and (Axelrod et al., 2020), as their complexity scales with the Lipschitz constant. For example, Bach's algorithms have complexity $\tilde{O}((\frac{n k L_{f_\downarrow}}{\epsilon})^2 \textnormal{EO}_F)$ when applied directly to $F$. Using the reduction to $\tilde{F}$, this increases by $O(k^2)$ factor.
>
> ---
> Thank you for your other suggestions. We will incorporate them in our revision.

---

### Decision · Program_Chairs · 2025-05-01

**Decision:**

Accept (poster)

**Comment:**

This paper studies how to minimize a difference of submodular functions on discrete and continuous domains that go beyond the standard set function case. The authors provide a DC-type algorithm, which is natural for this problem, but nevertheless novel. The algorithm is analyzed theoretically and empirically.
Most of the reviewers agree that this is a solid contribution to ICML, with the more knowledgeable reviewers being the more positive ones. I therefore support this sentiment and recommend acceptance.
I urge the authors to take reviewer feedback seriously and revise the paper accordingly.